# Detection of diagnostic and prognostic methylation-based signatures in liquid biopsy specimens from patients with meningiomas

Grayson A. Herrgott[1], James M. Snyder[1], Ruicong She[2], Tathiane M. Malta[1], Thais S. Sabedot[1], Ian Y. Lee[1], Jacob Pawloski[1], Guilherme G. Podolsky-Gondim [3], Karam P. Asmaro[1], Jiaqi Zhang[2], Cara E. Cannella[2], Kevin Nelson[1], Bartow Thomas[1], Ana C. deCarvalho [1], Laura A. Hasselbach[1], Kelly M. Tundo[1], Rehnuma Newaz[1], Andrea Transou[1], Natalia Morosini[1], Victor Francisco[1], Laila M. Poisson [1,2], Dhananjay Chitale [4], Abir Mukherjee[4], Maritza S. Mosella[1], Adam M. Robin[1], Tobias Walbert [1], Mark Rosenblum[1], Tom Mikkelsen[1], Steven Kalkanis[1], Daniela P. C. Tirapelli[3], Daniel J. Weisenberger [5], Carlos G. Carlotti Jr[3], Jack Rock[1], Ana Valeria Castro [1,6] ✉ & Houtan Noushmehr[1,6] ✉

Recurrence of meningiomas is unpredictable by current invasive methods based on surgically removed specimens. Identification of patients likely to recur using noninvasive approaches could inform treatment strategy, whether intervention or monitoring. In this study, we analyze the DNA methylation levels in blood (serum and plasma) and tissue samples from 155 meningioma patients, compared to other central nervous system tumor and non-tumor entities. We discover DNA methylation markers unique to meningiomas and use artificial intelligence to create accurate and universal models for identifying and predicting meningioma recurrence, using either blood or tissue samples. Here we show that liquid biopsy is a potential noninvasive and reliable tool for diagnosing and predicting outcomes in meningioma patients. This approach can improve personalized management strategies for these patients.

Meningiomas are the most common primary tumors of the central nervous system (CNS)[1]. According to the World Health Organization (WHO), meningiomas classified as grades 2 and 3 account for 20–30% of cases. These tumors present an estimated rate of recurrence of 20–75% across grade 2 and an observed universal rate of recurrence across grade 3, within 10 years of patient follow-up[2]. Additionally, some cases have potential for malignization, metastasizing and may even prove life-threatening[1]. An immediate challenge following meningioma identification lies in determining whether temporal surveillance through imaging or a tailored interventional approach, such

[1]Department of Neurosurgery, Omics Laboratory, Hermelin Brain Tumor Center, Henry Ford Health, Detroit, MI, USA. [2]Department of Public Health, Biostatistics, Henry Ford Health, Detroit, MI, USA. [3]Department of Neurosurgery, Ribeirao Preto Medical School, University of Sao Paulo, Ribeirao Preto, SP, Brazil. [4]Department of Pathology, Henry Ford Health, Detroit, MI, USA. [5]Department of Biochemistry and Molecular Medicine, Keck School of Medicine, University of Southern California, Los Angeles, CA 90033, USA. [6]Department of Physiology, Michigan State University, E. Lansing, MI, USA. ✉e-mail: acastro1@hfhs.org; hnoushm1@hfhs.org

as surgery or radiation, is the most appropriate patient management. Off-label investigational therapeutics have been attempted in clinical trials; however, no widely approved systemic therapies for this disease currently exist[3,4].

One of the principal hindrances to meningioma treatment advancement is the paucity of standardized assessment criteria or adequate biomarkers to measure success in clinical trials[5]. In tandem, detection of genomic and epigenomic biomarkers has become standard practice in oncology and proven valuable for classification, prognostication and appropriate management of CNS tumors, including meningiomas[6–14]. Specifically, stratification of meningiomas according to DNA methylation patterns in tumor tissue has proven to be an independent and reliable outcome predictor across all meningioma subtypes, and has outperformed the WHO grading system alone, across retrospective and prospective cohorts[8,10,11,15–18]. Furthermore, integration of DNA methylation-based groups with complementary molecular features (e.g., copy number variations, WHO grading, specific mutations: NF2, TERT, etc.) exhibited marked improvements in predicting the recurrence risk in patients with meningiomas[8,10,11,15–18]. Currently, these histologic and molecular characterizations are contingent on the profiling of meningioma tissue obtained through surgery. However, this approach may be infeasible for surgically inaccessible tumors, for patients with complicative comorbidities, or delayed, when tumors detected by imaging are mistakenly considered benign meningioma based on whether they are small, asymptomatic or discovered incidentally through imaging approaches[19–21]. Additionally, multiple surgeries are impractical and pose inherent cumulative risks for serial assessment of these tumors. Therefore, development of minimally- or noninvasive approaches to detect established or novel molecular markers which reflect real-time tumor biology and behavior is warranted. Imaging techniques are the current noninvasive approach used to guide diagnosis and management of meningiomas; however, its associated prognostic value is still unclear and longitudinal assessment may prove costly, unavailable and cumbersome for some patients[2,22]. Moreover, consensus standard radiographic criteria for inclusion and outcome evaluation for use across interventional trials established by the RANO group in 2018 illustrated that limitations exist in the application of imaging criteria alone to characterize this heterogenous disease[23].

Liquid biopsy (LB) is a non- or minimally invasive approach that allows for detection of material shed by tumors (e.g., circulating tumor cells and cell-free or tumor genomic elements) in biofluids (e.g., blood, cerebrospinal fluid, stool, urine, saliva and others)[24,25]. Several studies have described the feasibility of applying blood-based LB to screen mutations and DNA methylation abnormalities using serum- or plasma-cell free (cf) DNA from patients with CNS tumors[26–29]. However, current methylation-based prognostication models have been reported solely across surgically obtained tissue, but not LB specimens, from patients harboring these tumors[8,9,11,16,18,30].

Herein, we surveyed and identified DNA methylation-based signatures in serum which allowed for the development of machine learning classifiers able to accurately distinguish meningioma from controls and other CNS entities and predict recurrence risk, which may also be applied across tissue specimens. Our findings lay the foundation for the implementation of a presurgical detection of meningioma and assessment of its recurrence risk prediction (and possibly progression surveillance) using a noninvasive approach such as a blood draw, ultimately impacting the management and outcomes of patients harboring these tumors.

## Results

### Meningioma cohort features
Demographic and clinicopathological features of patients with meningiomas (MNG) and other CNS entities (non-MNG) treated at Henry Ford Health (HFH) and the University of Sao Paulo (USP) included in this study are detailed within Table 1. Meningioma and non-meningioma cohorts retrieved from the literature and employed in this investigation are detailed within Table 2.

### Methylation data features across liquid biopsy specimens
The preprocessing and quality assessment of our methylation arrays showed that all liquid biopsy samples, excluding one, met expected quality control standards (Supplementary Fig. S1a–e). No batch effects related to sample collection or extraction dates were observed across the liquid biopsy specimens' methylomes (Supplementary Fig. S1f, g).

Serum circulating cfDNA concentration (ng/µL) from patients with meningiomas were significantly lower than gliomas (Wilcoxon rank sum test; $p \leq 0.001$) and pituitary tumors (Wilcoxon rank sum test; $p \leq 0.01$). No significant differences in cfDNA concentration (ng/µL) across meningioma WHO grades, or recurrence risk predictions were observed (Supplementary Fig. S1j–m). The serum k3 cluster, identified through an unsupervised approach further described in Methods, presented the lowest concentration of serum cfDNA compared to other k-means clusters (k2 and k4).

### Paired serum- and plasma or tissue presented similar DNA methylation profiles
The comparison between paired serum and DNA plasma methylomes ($n = 10$ pairs) demonstrated that genome-wide DNA methylation levels and estimated immune cell profiles were highly correlated (Pearson's $\rho = 0.89$–$0.96$) (Supplementary Table S1). The diagnostic and prognostic classifications results from both serum and plasma were mostly concordant (80 and 70%, respectively) (Supplementary Table S1, Supplementary Data 1). Across the comparison between paired serum and tissue methylomes ($n = 25$), there was a significant and positive correlation in relation to genome-wide DNA methylation levels (Pearson's $\rho = 0.694$–$0.907$) (Supplementary Data 5). The correlation across immune proportions between both sources was positive but non-significant (Pearson's $\rho = 0.132$–$0.695$) (Supplementary Data 5).

### Serum cfDNA methylation levels distinguish meningioma from other CNS entities
We observed that genome-wide cfDNA methylation levels in serum only partly distinguished MNG from non-MNG conditions as depicted in the Principal Component Analysis (PCA) (Fig. 1a). However, through supervised methods, we identified 98 meningioma-specific differentially methylated probes (DMPs; 0. 15<diff.mean < −0.175) which significantly separated both groups (Wilcoxon rank sum test: p-value_FDR ≤ 0.05). Notably, the mean DNA methylation levels in MNGs serum were significantly lower compared to controls and non-MNG samples, such as gliomas (Wilcoxon rank sum test; $p \leq 0.001$) and pituitary tumors (Wilcoxon rank sum test; $p \leq 0.05$) (Fig. 1b; Supplementary Fig. S2a; Supplementary Data 2).

To investigate whether similar supervised methods would allow for translatability between tumor tissue and liquid biopsy methylomes, we compared MNG and non-MNG tissue collections and identified a subset of meningioma-specific probes ($n = 221$ DMPs; |diff.mean| ≥0.55; p_FDR ≤ 0.001) from which some signatures were detectable across the serum methylome and also distinguished MNG and non-MNG across serum specimens ($n = 24$ DMPs; Wilcoxon rank sum test: p_FDR ≤ 1e−04) (Fig. 1c, d, Supplementary Data 2).

### The diagnostic-Meningioma Epigenetic Liquid Biopsy (d-MeLB) classifier accurately classifies samples independently of the specimen source
We identified 256,447 tumor-specific CpG probes, termed meningioma-specific DMPs, which exhibited significant differential methylation between MNG tumor tissue ($n = 31$) and publicly available nontumor control collections (epileptic brain; $n = 21$)[31] and were

**Table 1 | Demographic and clinicopathological information for our serum- and tissue HFH/USP cohorts**

| | Original cohort | | | | | | | | Additional cohort | | | | | | | |
| --- | --- | --- | --- | --- | --- | --- | --- | --- | --- | --- | --- | --- | --- | --- | --- | --- |
| | Liquid biopsy (serum) | | | | Meningioma tissue | | | | Meningioma | | | | | | Non-meningioma | |
| | Meningioma (N=63) | | Non-meningioma (N=141) | | Confirmed recurrence (N=35) | | Confirmed non-recurrence (N=15) | | LB: serum (N=20) | | LB: plasma (N=10) | | Tumor tissue (N=39) | | LB: serum (N=6) | |
| Features | Median | (Q1, Q3) | Median | (Q1, Q3) | Median | (Q1, Q3) | Median | (Q1, Q3) | Median | (Q1, Q3) | Median | (Q1, Q3) | Median | (Q1, Q3) | Median | (Q1, Q3) |
| Age at diagnosis (yrs) | 60 | (45.5, 67.0) | 54.0 | (43.0, 64.0) | 56.0 | (49.5, 60.5) | 55.0 | (45.0, 60.5) | 51.0 | (46.8, 63.5) | 51 | (46.3, 60.3) | 51 | (46.5, 64) | 49.0 | (45, 55.5) |
| Person-time (mos) | 31.6 | (23.7, 49.0) | - | - | 26.8 | (6.1, 57.5) | 102.1 | (75.6, 137.4) | 1.6 | (0.1, 32.6) | 1.6 | (0.2, 3.5) | 21.6 | (3.4, 36.9) | - | - |
| MIB LI (%) | 12.1 | (4.5, 15.0) | - | - | - | - | - | - | - | - | - | - | - | - | - | - |
| **Sex** | n | % | n | % | n | % | n | % | n | % | n | % | n | % | n | % |
| Female | 36 | 57.1 | 40 | 28.4 | 17 | 48.6 | 12 | 80 | 8 | 40 | 5 | 50 | 20 | 51.3 | 2 | 33.3 |
| Male | 27 | 42.9 | 66 | 46.8 | 18 | 51.4 | 3 | 20 | 12 | 60 | 5 | 50 | 19 | 48.7 | 4 | 66.7 |
| Unknown | - | - | 35 | 24.8 | - | - | - | - | - | - | - | - | - | - | - | - |
| **Race/Ethnicity** | n | % | n | % | n | % | n | % | n | % | n | % | n | % | n | % |
| Black or African American | 10 | 15.9 | 17 | 12.1 | 3 | 8.6 | 2 | 13.3 | 5 | 25 | 5 | 50 | 8 | 20.5 | 1 | 16.7 |
| White | 46 | 73 | 85 | 60.3 | 26 | 74.3 | 11 | 73.3 | 13 | 65 | 3 | 30 | 25 | 64.1 | 5 | 83.3 |
| Other | 5 | 7.9 | 4 | 2.8 | - | - | 1 | 6.7 | 1 | 5 | 1 | 10 | 2 | 5.1 | - | - |
| Unknown | 2 | 3.2 | 17 | 12.1 | 6 | 17.1 | 1 | 6.7 | 1 | 5 | 1 | 10 | 4 | 10.3 | - | - |
| **Post-surgical MRI report** | n | % | n | % | n | % | n | % | n | % | n | % | n | % | n | % |
| Stable disease | 19 | 30.2 | - | - | 6 | 17.1 | 5 | 33.3 | 2 | 10 | 1 | 10 | 12 | 30.8 | - | - |
| Progressive disease | 11 | 17.5 | - | - | 24 | 68.6 | - | - | 7 | 35 | 3 | 30 | 13 | 33.3 | - | - |
| Non-enhancing disease | 29 | 46 | - | - | 3 | 8.6 | 10 | 66.7 | 2 | 10 | - | - | 3 | 7.7 | - | - |
| Unknown | 4 | 12.7 | - | - | 2 | 5.7 | - | - | 9 | 45 | 6 | 60 | 11 | 28.2 | - | - |
| **Tumor Classification/Histopathological diagnosis** | n | % | n | % | n | % | n | % | n | % | n | % | n | % | n | % |
| Meningioma | 63 | 100 | - | - | 35 | 100 | 15 | 100 | 20 | 100 | 10 | 100 | 39 | 100 | - | - |
| Not other specified (NOS) | 23 | 36.5 | - | - | 1 | 2.9 | 1 | 6.7 | 8 | 40 | 3 | 30 | 10 | 25.6 | - | - |
| Atypical | 24 | 38.1 | - | - | 25 | 71.4 | 8 | 53.3 | 8 | 40 | 4 | 40 | 16 | 41 | - | - |
| Anaplastic | 8 | 33.3 | - | - | 5 | 14.3 | - | - | - | - | - | - | 5 | 12.8 | - | - |
| Rhabdoid | 1 | 1.6 | - | - | 1 | 2.9 | 3 | 20 | - | - | - | - | - | - | - | - |
| Fibrous | 1 | 1.6 | - | - | - | - | - | - | - | - | - | - | - | - | - | - |
| Meningothelial | 2 | 3.2 | - | - | 2 | 5.7 | - | - | 2 | 10 | 2 | 20 | 5 | 12.8 | - | - |
| Psammomatous | 3 | 4.8 | - | - | - | - | - | - | 1 | 5 | 1 | 10 | 1 | 2.6 | - | - |
| Secretory | 1 | 1.6 | - | - | - | - | 1 | 6.7 | - | - | - | - | - | - | - | - |
| Chordoid | - | - | - | - | - | - | - | - | - | - | - | - | - | - | - | - |
| Transitional | - | - | - | - | 1 | 2.9 | 2 | 13.3 | 1 | 5 | - | - | 2 | 5.2 | - | - |
| Glioma | - | - | 109 | 77.3 | - | - | - | - | - | - | - | - | - | - | - | - |
| Pituitary tumor | - | - | 14 | 9.9 | - | - | - | - | - | - | - | - | - | - | - | - |
| Non-tumor | - | - | 6 | 4.3 | - | - | - | - | - | - | - | - | - | - | 1 | 16.7 |
| Other | - | - | 12 | 8.5 | - | - | - | - | - | - | - | - | - | - | 5 | 83.3 |
| CNS lymphoma | - | - | 4 | 33.3 | - | - | - | - | - | - | - | - | - | - | - | - |

**Table 1 (continued) | Demographic and clinicopathological information for our serum- and tissue HFH/USP cohorts**

| | Original cohort | | | | Additional cohort | | | | | |
|---|---|---|---|---|---|---|---|---|---|---|
| | Liquid biopsy (serum) | | Meningioma tissue | | Meningioma | | | | Non-meningioma | |
| | n | % | n | % | n | % | n | % | n | % |
| Other CNS diseases | 77 | 53.1 | - | - | - | - | - | - | 5 | 100 |
| WHO grade | n | % | n | % | n | % | n | % | n | % |
| 1 | 30 | 47.6 | 4 | 11.4 | 12 | 60 | 6 | 60 | 19 | 48.7 |
| 2 | 24 | 38.1 | 25 | 71.4 | 8 | 40 | 4 | 40 | 16 | 41 |
| 3 | 9 | 14.3 | 6 | 17.1 | - | - | - | - | 4 | 10.3 |
| 4 | - | - | - | - | - | - | - | - | - | - |
| Unknown | 28 | 19.3 | - | - | - | - | - | - | - | - |
| Not applicable | 27 | 18.6 | - | - | - | - | - | - | 6 | 100 |
| Tumor location[a] | n | % | n | % | n | % | n | % | n | % |
| Calvarium | 42 | 66.7 | 28 | 80 | 16 | 80 | 8 | 80 | 26 | 66.6 |
| Anterior fossa | 5 | 7.9 | - | - | 1 | 5 | 1 | 10 | 1 | 2.6 |
| Middle fossa | 7 | 11.1 | 3 | 8.6 | - | - | - | - | 2 | 5.2 |
| Posterior fossa | 2 | 3.2 | 1 | 2.9 | 2 | 10 | - | - | 3 | 7.6 |
| Spinal | 3 | 4.8 | - | - | 1 | 5 | 1 | 10 | 4 | 10.2 |
| Other (orbital, other) | 3 | 4.8 | - | - | - | - | - | - | 2 | 5.2 |
| Unknown | 1 | 1.6 | 3 | 8.6 | - | - | - | - | 1 | 2.6 |
| Time of Collection | n | % | n | % | n | % | n | % | n | % |
| Primary | 44 | 69.8 | 18 | 51.4 | 11 | 55 | 2 | 20 | 23 | 48.9 |
| Recurrent | 19 | 30.2 | 17 | 48.6 | 9 | 45 | 8 | 80 | 16 | 41.1 |
| Follow-up (OT) | 6 | 4.1 | - | - | - | - | - | - | - | - |
| Not applicable | 8 | 5.5 | - | - | - | - | - | - | - | - |
| Unknown | 10 | 6.9 | - | - | - | - | - | - | - | - |
| Extent of resection | n | % | n | % | n | % | n | % | n | % |
| Gross total (GTR) | 40 | 63.5 | 22 | 62.9 | 5 | 25 | 2 | 20 | 14 | - |
| Sub-total (STR) | 21 | 33.3 | 13 | 37.1 | 4 | 20 | 4 | 40 | 10 | - |
| Unknown | 2 | 3.2 | - | - | 11 | 55 | 4 | 40 | 15 | - |
| Presurgical treatment | n | % | n | % | n | % | n | % | n | % |
| Embolization | 12 | 19.1 | 1 | 2.9 | - | - | - | - | 1 | 2.6 |
| Radiotherapy | 25 | 39.7 | 23 | 65.7 | 5 | 25 | 4 | 40 | 17 | 43.6 |
| Tumor recurrence | n | % | n | % | n | % | n | % | n | % |
| Confirmed recurrence | 16 | 25.4 | 35 | 100 | 8 | 40 | 4 | 40 | 25 | 64.1 |
| Confirmed non-recurrence | 9 | 14.3 | - | - | - | - | - | - | 2 | 5.1 |
| Unknown | 38 | 60.3 | - | - | 12 | 60 | 6 | 60 | 12 | 30.8 |
| Last report status | n | % | n | % | n | % | n | % | n | % |
| Alive | 57 | 90.5 | 5 | 14.3 | 20 | 100 | 10 | 100 | 28 | 71.8 |
| Dead | 5 | 7.9 | 30 | 85.7 | - | - | - | - | 11 | 28.2 |
| Unknown | 1 | 1.6 | - | - | - | - | - | - | - | - |

LB liquid biopsy, LI labeling indices, OT off-treatment.
[a]Locations based on Yuzawa et al., 2016.

**Table 2 | Publicly available tumor tissue cohort descriptions**

| Publication | OMICs DNA methylation | RNA expression/ sequencing | Data accession | Analyses | Relevant figures |
|---|---|---|---|---|---|
| **MNG cohorts** | | | | | |
| Gao et al., 2013 | Illumina HumanMethylation450 Beadchip; n = 19 MNG | | GSE42882 | (1) Investigation into MeLB DMP clustering (2) Investigation of d-MeLB signature meningioma specificity. | (1/2) Figure 1c, e |
| Capper et al., 2018 | Illumina HumanMethylation450 Beadchip; n = 149 MNG | | GSE109381 | | |
| Harmanci et al., 2018 | Illumina HumanMethylation450 Beadchip; n = 57 MNG | | GSE85135 | | |
| Bayley et al., 2022 | Illumina Methylation EPIC; n = 110 MNG | | GSE189521 | (1) Independent validation of p-MeLB. (2) Validation of p-MeLB classifications. (3) Independent validation of tissue-based LDA. | (1) Figure 2b (2) Figures 3c and S3e, f (3) Figure S4b |
| Choudhury et al., 2022 | Illumina Methylation EPIC; n = 565 MNG | Illumina HiSeq 4000 (Homo Sapiens): n = 185 MNG | GSE183656 | (1) Validation of p-MeLB classifications. (2) Downstream correlation of gene transcription and DNA methylation across RR PGPs (tissue- and serum-identified). | (1) Figures 3c and S3c, d (2) Figure 3d, f, g |
| **Non-MNG cohorts** | | | | | |
| Capper et al., 2018 | Illumina HumanMethylation450 Beadchip; n = 1,20 Non-MNG tumors (embryonal, ependymal, etc.) | | GSE109381 | (1) Investigation into MeLB DMP clustering. (2) Investigation of d-MeLB signature meningioma specificity. | (1/2) Figure 1c, e |
| Mosella et al., 2021 | Illumina HumanMethylation450 Beadchip; n = 10 Pituitary tumors | | Mendeley Data DOI: 10.17632.5pzd2rg5ys.1 | | |
| Kober et al., 2018 | Illumina HumanMethylation450 Beadchip; n = 34 Pituitary tumors | | GSE115783 | | |
| Ling et al., 2014 | Illumina HumanMethylation450 Beadchip; n = 23 Pituitary tumors | | GSE54415 | | |
| The Cancer Genome Atlas (TCGA) | Illumina HumanMethylation450 Beadchip; n = 516 Gliomas | | N/A | | |
| Sloan et al., 2022 | Illumina Methylation EPIC; n = 20 Intracranial Mesenchymal tumors | | GSE164994 | (1) Independent validation of tissue-derived linear discriminant analyses. | (1) Figure S4b |
| N/A | Illumina Methylation EPIC; n = 16 Gliomas | | GSE147391 | | |
| Braun et al., 2019 | Illumina Methylation EPIC; n = 21 Epileptic Non-tumor brain | | GSE111165 | (1) Tumor-specific dimension reduction (d-MeLB step #1). (2) Independent validation of tissue-derived linear discriminant analysis (LDA). | (1) Figure S3b (2) Figure S4b |

MNG meningioma, LGG low-grade glioma.

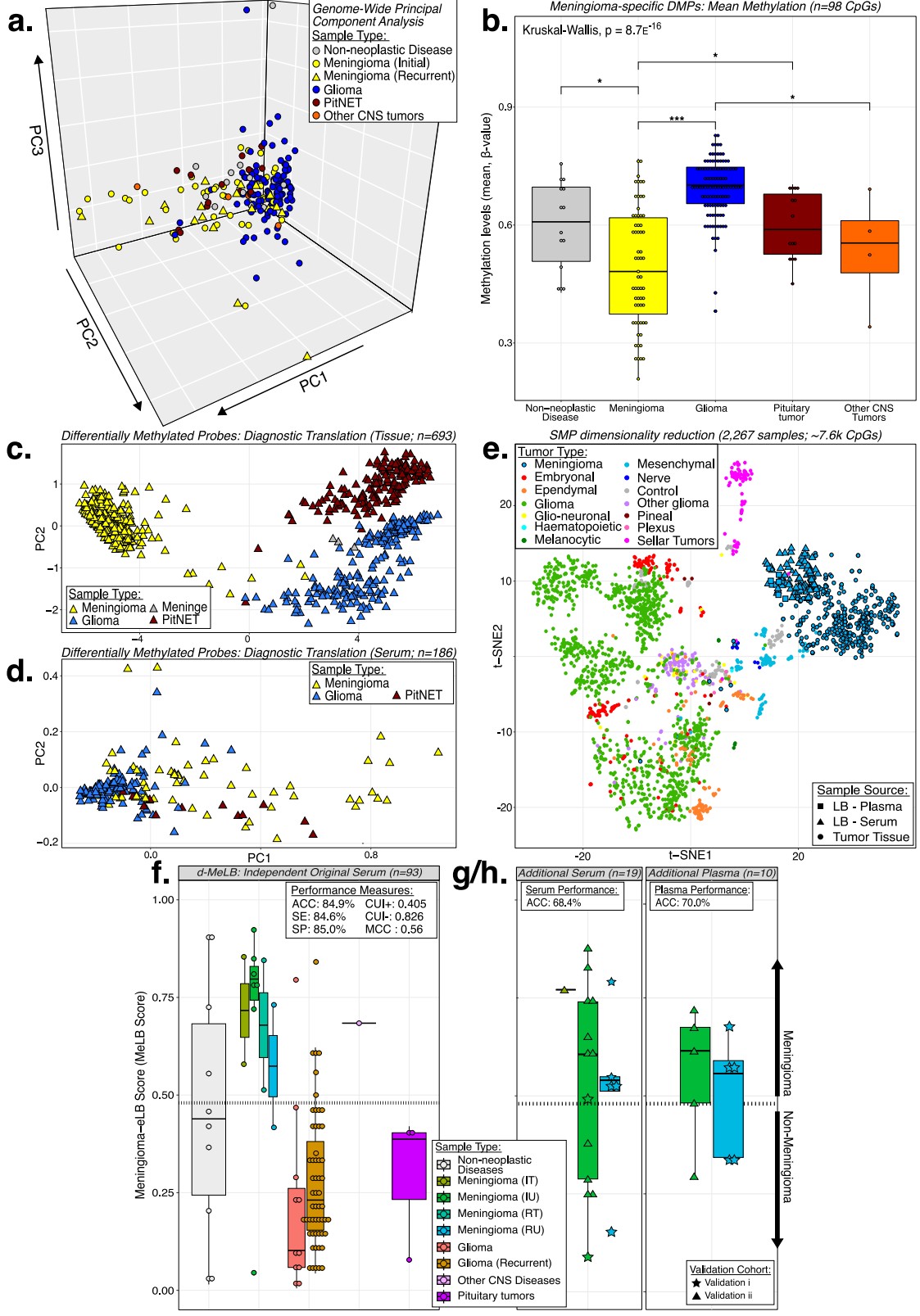

utilized as input into diagnostic classifier construction (Supplementary Fig. S2b, step #1). Within the algorithm, these DMPs were further filtered to those with high DNA methylation level similarities between paired meningioma serum and tissue specimens collected at the time of surgery (origin: Henry Ford Health), namely similarly methylated probes (SMPs: *n* = 7659; Supplementary Data 2). T-distributed stochastic neighbor embedding (t-SNE) was applied to visualize the

behavior of these signatures across internal and external tissue cohorts of meningiomas and other central nervous system (CNS) entities[13,32–36]. Interestingly, we observed that the SMPs clustered meningioma tissue samples together with serum and plasma specimens, and effectively distinguished meningiomas from other CNS entities (Fig. 1e). By filtering these SMPs through a serum-based supervised analysis (untreated MNG vs non-MNG cohorts; Wilcoxon rank-sum test),

**Fig. 1 | Serum circulating cell-free DNA methylation patterns and signatures distinguish meningiomas from other CNS entities. a** Principal component analysis (PCA) depicting the genome-wide mean methylation levels of serum cfDNA derived from patients with meningioma (MNG; $n = 63$) and non-MNG conditions (other CNS entities and non-neoplastic diseases; $n = 141$). Note: MNG: meningiomas; CNS Central Nervous System. **b** Mean methylation levels of the differentially methylated CpG probes (DMP, $n = 98$) across comparisons between MNG and non-MNG (Wilcoxon rank sum test; Kruskal-Wallis; *$p < 0.05$, **$p < 0.01$, ***$p < 0.001$). Box plots - data are presented as median and upper (75%) and lower (25%) quartiles. Whiskers represent minimum to maximum values, excluding outliers. Exact $p$-values: Meningioma vs Non-neoplastic Disease: $p = 0.018$; Meningioma vs Glioma: $p = 4.4e{-}16$; Meningioma vs pituitary neuroendocrine tumors: $p = 0.017$; Glioma vs Other CNS tumors: $p = 0.012$. Note: DMP: differentially methylated probes. t-distributed stochastic neighbor embedding (t-SNE) plots displaying clustering of

meningioma-specific DMPs across MNG and non-MNG tissue specimens (**c**). A subset of these DMPs is detected in the serum and also distinguish equivalent groups (**d**). **e** t-SNE plot displaying dimension-reduced diagnostic-Meningioma Epigenetic Liquid Biopsy (d-MeLB) probes (SMP: $n = 18k$ CpGs) across CNS tumor tissue, liquid biopsy (serum and plasma) and tumor tissue from patients with MNG. SMP similarly methylated probes, LB liquid biopsy. Distribution of the d-MeLB scores across independent cohorts (**f**: original liquid biopsy serum, $n = 93$; **g**: additional MNG serum, $n = 19$; **h**: additional MNG plasma, $n = 10$) (Dashed line: MeLB cutoff score). Box plots - data are presented as median and upper (75%) and lower (25%) quartiles. Whiskers represent minimum to maximum values, excluding outliers. Upper left corner: performance measures. ACC Accuracy, SE Sensitivity, SP Specificity, CUI Clinical Utility Index, MCC Matthew's Correlation Coefficient, IT initial treated, IU initial untreated, RT recurrent treated, RU recurrence untreated.

we derived a signature set applicable to liquid biopsy samples (Supplementary Fig. S2b).

The application of the d-MeLB classifier across the model selection serum cohort showed that a score threshold ≥0.48 had the highest classification accuracy (AUC:1.00; Supplementary Data 1). Validation of the diagnostic classifier was conducted across an independent cohort of MNG and non-MNG including the samples randomized ($n = 30$; Supplementary Fig. S2b, step #2) and excluded ($n = 63$; Supplementary Fig. S2b, step #9.5) during model construction, namely the independent 'original' serum cohort ($n = 93$). Across this cohort, we observed an 84.9% accuracy in identification of meningioma/non-meningioma, with satisfactory performance measures. i.e., Matthew correlation coefficient (MCC: 0.56) and clinical utility index (CUI + : 0.405) (Fig. 1f). Two rounds of additional MNG collections (validation i & ii) were profiled and incorporated as validation independent cohorts after the initial model derivation (serum: $n = 19$; plasma: $n = 10$) in which d-MeLB displayed classification accuracies of 68.4% and 70.0%, respectively (Fig. 1g). For comparisons, performance across the entire liquid biopsy independent cohort ($n = 122$) was considered (Supplementary Fig. S2b, step #10).

In effort to investigate whether our diagnostic MeLB signature presented a spurious immune-related bias due to potential contamination of serum with immune-cell signature released by white blood cells during the clotting process, we applied our generated classifier across an independent cohort of fluorescence activated cell sorting (FACS) purified immune cell and whole blood profiles ($n = 59$)[37]. The d-MeLB had an overall accuracy of 93.2% to classify these samples as non-meningiomas, including neutrophils and whole blood (Supplementary Fig. S2c).

Of note, formulation of the d-MeLB classifier was not conducted with tissue classification in mind and did not include tissue specimens within discovery or independent validation sets; so, it was expected that tissue application would be limited (ACC: <10%). To address this limitation, we used d-MeLB signatures as coefficients for a simple linear-based discriminant algorithm to classify tissue-based collections composed of meningioma and non-meningioma. Summarily, we observed an accuracy of 94.3% to classify an independent cohort into their correct memberships ($n = 176$; Supplementary Fig. S4a).

Confirmation of the detection of our diagnostic signatures (d-MeLB: $n = 25$ CpGs) across 10 cfDNA samples profiled through whole genome bisulfite sequencing (WGBS) was conducted. Their methylation levels determined by β-values (EPIC Array), or percentage values (WGBS) were significantly correlated across these samples (Pearson's ρ = 0.6, $p \leq 2.2e{-}16$) (Supplementary Fig. S4b, c).

### The d-MeLB classifier outperforms other classifier approaches

We compared the performance of the random forest (RF) approach used to develop the d-MeLB with other classification methods using our internal methylome cohort data ($n = 239$), with identical discovery ($n = 117$ MNG & Non-MNG; Supplementary Fig. S2b) and validation

($N = 122$; randomized=30; excluded=63; additional serum=19; additional plasma=10) cohorts as those used throughout the d-MeLB. Compared to the results obtained with the d-MeLB classifier based on random forest algorithm, other approaches including dimension reduction RF, linear discriminant analysis (LDA), extreme gradient boosting [package: XGBoost v1.7.4][38] and logistic regressions (univariate and multivariate analyses of mean methylation values) presented lower accuracies in classification of independent liquid biopsy MNG and non-MNG (Supplementary Table S3 and Supplementary Methods).

### Serum cfDNA methylation clusters are associated with distinct clinicopathological features, outcomes, and immune composition across meningioma specimens

Unsupervised consensus clustering analysis revealed four main k-clusters with distinct cfDNA methylation profiles across MNG serum specimens (Fig. 2a, Supplementary Data 3).

The annotation of these serum-derived molecular groups with clinicopathological and molecular features showed that the clustering occurred independently of sex, age, and race (Supplementary Data 1) and were enriched with features associated with MNG outcomes and prognosis. For instance, compared to k1-and k2 clusters, k3- and k4-clusters presented an enrichment of WHO grades 2 and 3, and confirmed recurrence during patient follow-up (Fig. 2a, Supplementary Fig. S5a).

Through cfDNA methylation-based deconvolution analysis[39], we discovered that the k4 cluster was enriched with neutrophil cell signatures and possessed the highest neutrophil-lymphocyte ratio (NLR) and depleted of the majority of immune cell types included in the analysis. In contrast, the k1-cluster was depleted in neutrophils while enriched with almost all immune cell type proportion estimates (B- and T-cells, natural killer [NK] and monocytes) compared to other clusters (Supplementary Fig. S5a).

### The prognostic-Meningioma Epigenetic Liquid Biopsy (p-MeLB) classifier predicts risk of recurrence (RR) of meningiomas using serum or tissue specimens

The p-MeLB classifier presented an overall 87.7% accuracy and satisfactory performance measurements (CUI + : 86.4%; MCC = 0.577) in predicting true recurrence in an independent validation tissue- and liquid biopsy-based cohort, as confirmed during established follow-up (original cohort: ACC = 82.6%; additional validations: ACC = 90%; Fig. 2b, c).

The application of the p-MeLB classifier to primary meningioma tissue collections ($n = 69$), derived from all three separate profiling's (original, validation i & ii), demonstrated significant agreement with the classifications obtained from a previously published nomogram (Cohen's unweighted kappa; κ = 0.269, $p_κ = 0.01$)[10]. Interestingly, across a subset of primary MNG independent from p-MeLB classifier derivation, p-MeLB demonstrated higher sensitivity (SE) to predict

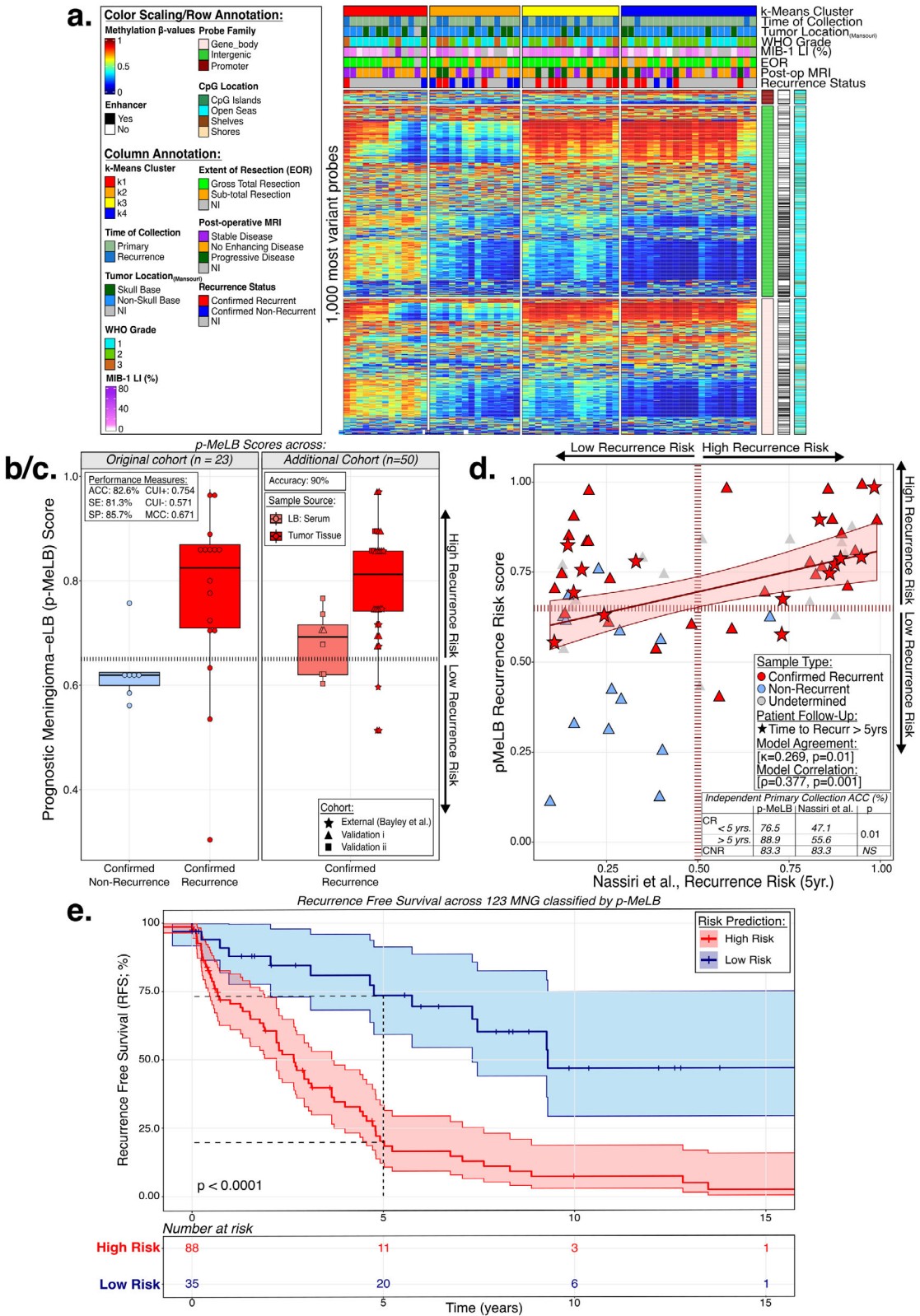

true recurrence confirmed during follow-up, compared to the nomogram-derived results (<5 yrs: SE = 76.5 vs. 47.1%; ≥5 yrs: SE = 88.9 vs. 55.6%) (Fig. 2d).

Across our total cohorts of MNG-tissue ($n = 123$) and serum ($n = 80$) specimens which possessed attributed person-time (mean: 3.7 and 2.7 years, respectively), the 5-year recurrence-free survival probability was significantly lower in MNG classified as having a high risk

than those classified as low risk of recurrence (tissue: 20% vs 73%; serum: 35% vs 75%; log-rank $p \le 1.0e^{-4}$) (Fig. 2e, Supplementary Data 1).

We also confirmed the detection of our prognostic signatures (p-MeLB: $n = 13$ CpGs, 70 high-risk related DMPs) across 10 cfDNA samples profiled through whole genome bisulfite sequencing (WGBS). Similar to the observed for d-MeLB signatures, the DNA methylation levels of the p-MeLB signatures as determined by

**Fig. 2 | Serum circulating cell-free DNA methylation patterns and signatures differentiate meningiomas with different behaviors. a** Methylation heatmap displaying the 1000 most variable methylated probes (β-values) across serum meningioma unsupervised k-clusters (n = 63). Samples are sorted into methylation-based clusters and annotated with clinicopathological/molecular features. Vertical tracks (right) genomic annotations. LI Labeling Index, EOR Extent of Resection, NI non-informed, MRI magnetic resonance imaging **b/c.** Distribution of the prognostic-Meningioma Epigenetic Liquid Biopsy (p-MeLB) scores across (**b**) the original independent cohort (n = 23) and (**c**) additional validations (n = 50) from patients with meningiomas presenting different outcomes (confirmed recurrence or no recurrence; dashed lines: p-MeLB score cutoff). Box plots - data are presented as median and upper (75%) and lower (25%) quartiles. Whiskers represent minimum to maximum values, excluding outliers. Upper left corner: performance measures. ACC Accuracy, SE Sensitivity, SP Specificity, CUI Clinical Utility Index, MCC Matthew's Correlation Coefficient. **d** Scatterplot displaying the relationship between p-MeLB and the nomogram recurrence risk prediction across the primary meningioma tissue subset. Linear relationship is depicted with 95% confidence interval (lower and upper limits). Measurements of concordance are displayed (Cohen's unweighted kappa/Spearman's ρ, p < 0.05). Table comparing the accuracies of p-MeLB and the nomogram-based classifier across an independent subset of primary meningioma tissue (n = 69). ACC accuracy, CR Confirmed Recurrence, CNR Confirmed No Recurrence. **e** Kaplan-Meier survival curves displaying meningioma tumor tissue samples stratified by their predicted recurrence risk (n = 127, vertical ticks: censorship). Survival curves are depicted with 95% confidence intervals (lower and upper limits) for point estimates; comparisons of median survival time in both recurrence risk groups were conducted using log-rank tests (p < 0.0001). MNG meningioma, RR recurrence risk.

β-values (EPIC array) or percentage values (WGBS) were highly correlated across these samples (p-MeLB: Pearson's ρ = 0.73, p = 3.1e−16; risk-related DMPs: Pearson's ρ = 0.68, p = 2.2e−16) (Supplementary Fig. S4b, c). These findings indicate that our EPIC-based results are further supported by WGBS, which serves as a secondary benchmark profiling method.

## High and low recurrence risk meningioma groups present differential clinicopathological features and estimated immune landscapes in serum and tissue specimens

To further characterize our predicted risk groups, we estimated differences in the distribution of relevant clinical features associated with prognosis between samples classified as high and low risk for recurrence (Fig. 3a, b). Summarily, in serum specimens, we observed a significantly higher odds ratio (OR) of a confirmed recurrence during follow-up occurring in high risk compared to low-risk specimens (OR = 15.45, 95% CI: [1.45, 844.77]; p ≤ 0.05). No significant differences were observed in relation to MNG location (skull base/non-skull base), extent of resection (gross total resection [GTR]/subtotal resection [STR]), WHO grades (2&3/1), progression in post-surgical MRI reports (Progressive/non-enhanced and stable disease) or vital status (deceased/alive), among others (Fig. 3a). Additionally, in high-risk specimens we observed significant enrichment in the estimated proportions of neutrophils and depletion of B-cells (p = 0.002), NK (p = 4.00E−04) and CD4-T cells (p = 0.07) and high NLR compared to their low-risk counterparts (Fig. 3b).

In tissue specimens, similar to serum findings, we observed significantly higher odds of a confirmed recurrence during follow-up in high- compared to low-risk specimens; no differences regarding tumor location or grade (OR = 32.2, 95% CI: [5.71, 351.06]; p ≤ 0.05) (Supplementary Fig. S5b); estimated NLR (p = 0.08), neutrophils (p = 0.07), and NK (p = 0.1) proportions (Supplementary Fig. S5c). In contrast to serum findings, compared to low risk, high risk samples presented decreased odds of having a gross total resection (OR = 0.15, 95% CI: [0.01, 0.76]; p ≤ 0.05) and higher odds of progressive disease post-surgical MRI reports (OR = 11.91, 95% CI: [2.63, 77.84]; p ≤ 0.05) (Supplementary Fig. S5b).

## p-MeLB classifications are validated across external meningioma tissue cohorts

In order to further assess the robustness of our p-MeLB predictor, we compared the performance of p-MeLB with other tissue methylome-based prognostic classifiers using a common external meningioma tissue cohort. The p-MeLB recurrence risk predictions across external meningioma tissue-methylome cohorts aligned with prognostic and survival differences reported by other authors[16,18]. For instance, within the Choudhury[16] hypermitotic group characterized by the poorest 5-year recurrence-free survival probability (~35%), a majority of samples was classified as high risk for recurrence by p-MeLB; while the Merlin-intact group with the most favorable 5-year recurrence-free survival (~85%) was largely classified as low risk (Fig. 3c, Supplementary Fig. S3c, d). The Bayley[18] malignant MenG-C group with the poorest recurrence-free survival probability compared to their more benign counterparts (MenG-A and -B groups), was unanimously classified as high recurrence risk by p-MeLB (Fig. 3c, Supplementary Fig. S3e, f). Overall, the assessment of p-MeLB's ability to detect true recurrence during follow-up across these external cohorts was limited due to the lack of sufficient longitudinal information[16,18].

## Tissue-derived differentially methylated probes are detectable and moderately differentiate recurrence risk groups in serum specimens

We identified a subset of tissue-derived prognostically-relevant DMPs from the comparison between confirmed recurrence (CR) and confirmed non-recurrence (CNR) specimens (pFDR < 0.001 & |diff.mean| ≥0.55; n = 260 DMPs) which presented congruent DNA methylation levels with serum and distinguished a majority of high and low risk in our original (n = 63) and additional (validation i & ii; n = 18) independent serum cohorts (pFDR < 0.001 & |diff.mean| ≥ 0.27; n = 39 DMPs), with minor intermingling of risk-groups (Fig. 4a, b).

## Prognostic probes located in gene regulatory elements potentially control the expression of target genes associated with tumor development and growth—in silico functional analysis

Through the intersection between Choudhury and p-MeLB classifications, we identified and compared two prognostic groups in tissue specimens, i.e., high-risk hypermitotic vs low-risk merlin MNG to identify prognostic-specific DMPs. By performing an integrative analysis of paired methylome and transcriptome meningioma tissue data[16], we identified prognostic DMPs in regulatory regions which were differentially methylated and targeted genes which were differentially expressed between these prognostic groups (probe-gene pairs [PGPs]). For downstream analysis, to identify putative epigenetically regulated genes, we selected PGPs which possessed a negative correlation between DNA methylation and expression levels (n = 65 PGPs; Fig. 4c, Supplementary Data 4). Among these PGPs identified in tissue, 12 CpGs presented concordant DNA methylation between serum and tissue specimens and also differentiated recurrence risk groups across serum specimens (e.g., hypermethylated in both tissue and serum in high-risk specimens) (Fig. 4d, Supplementary Data 4).

Additionally, across serum specimens, we identified 70 risk related DMPs through the supervised analysis between high and low recurrence risk meningioma (Supplementary Fig. S5g, Supplementary Data 3). Mapping these serum derived DMPs to tissue sample methylomes and to their putative target genes in the Choudhury dataset, we selected PGPs which exhibited negative correlation between gene expression and CpG probe DNA methylation levels across risk group comparisons (n = 25 PGPs). CpG probes with concordant DNA methylation levels between tissue-serum PGP are highlighted (Fig. 4e, Supplementary Data 4).

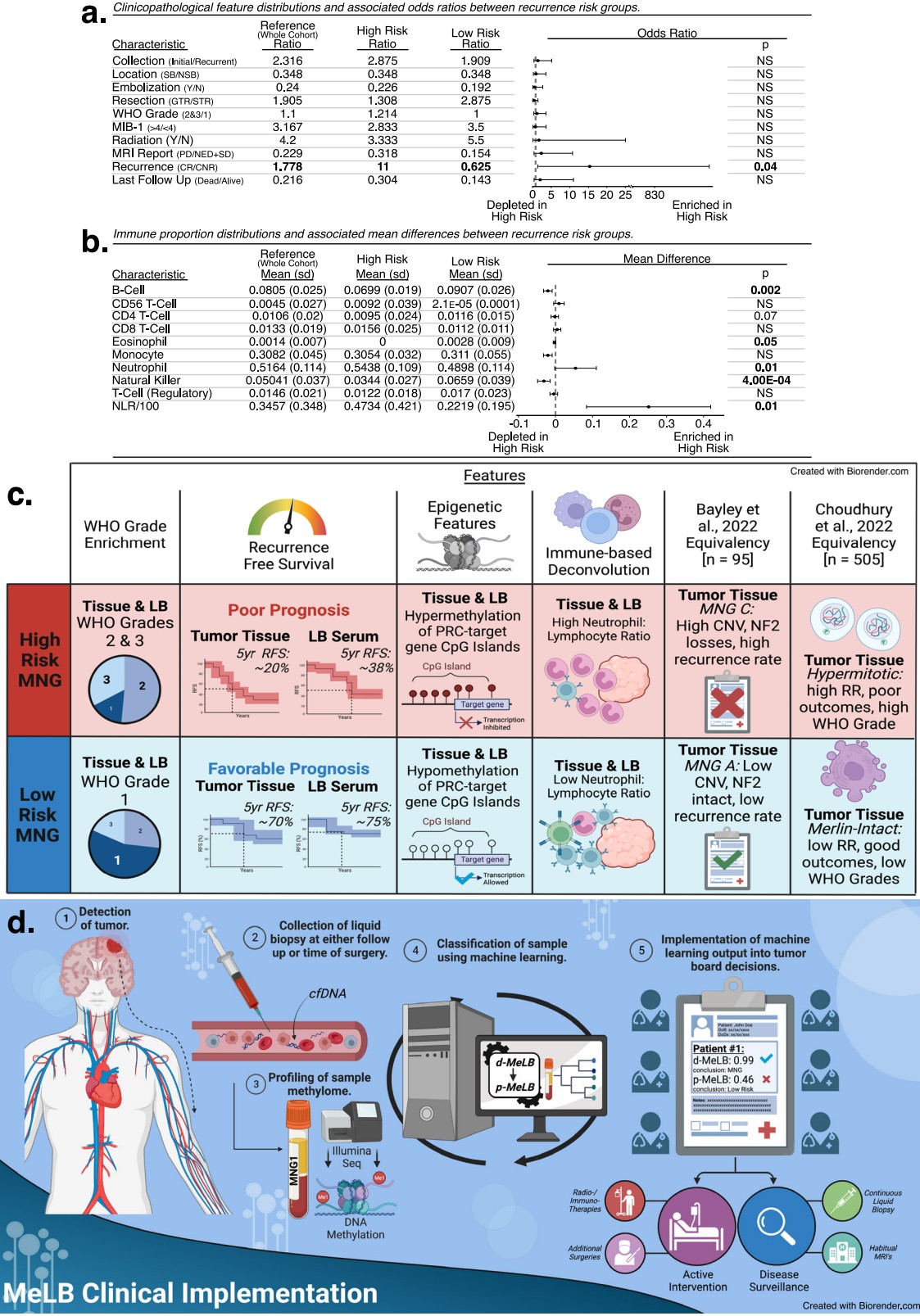

**a.** *Clinicopathological feature distributions and associated odds ratios between recurrence risk groups.*

| Characteristic | Reference (Whole Cohort) Ratio | High Risk Ratio | Low Risk Ratio | Odds Ratio | p |
|---|---|---|---|---|---|
| Collection (Initial/Recurrent) | 2.316 | 2.875 | 1.909 | | NS |
| Location (SB/NSB) | 0.348 | 0.348 | 0.348 | | NS |
| Embolization (Y/N) | 0.24 | 0.226 | 0.192 | | NS |
| Resection (GTR/STR) | 1.905 | 1.308 | 2.875 | | NS |
| WHO Grade (2&3/1) | 1.1 | 1.214 | 1 | | NS |
| MIB-1 (>4/<4) | 3.167 | 2.833 | 3.5 | | NS |
| Radiation (Y/N) | 4.2 | 3.333 | 5.5 | | NS |
| MRI Report (PD/NED+SD) | 0.229 | 0.318 | 0.154 | | NS |
| Recurrence (CR/CNR) | **1.778** | **11** | **0.625** | | **0.04** |
| Last Follow Up (Dead/Alive) | 0.216 | 0.304 | 0.143 | | NS |

0  5  10 15 20 25  830
Depleted in High Risk — Enriched in High Risk

**b.** *Immune proportion distributions and associated mean differences between recurrence risk groups.*

| Characteristic | Reference (Whole Cohort) Mean (sd) | High Risk Mean (sd) | Low Risk Mean (sd) | Mean Difference | p |
|---|---|---|---|---|---|
| B-Cell | 0.0805 (0.025) | 0.0699 (0.019) | 0.0907 (0.026) | | **0.002** |
| CD56 T-Cell | 0.0045 (0.027) | 0.0092 (0.039) | 2.1E-05 (0.0001) | | NS |
| CD4 T-Cell | 0.0106 (0.02) | 0.0095 (0.024) | 0.0116 (0.015) | | 0.07 |
| CD8 T-Cell | 0.0133 (0.019) | 0.0156 (0.025) | 0.0112 (0.011) | | NS |
| Eosinophil | 0.0014 (0.007) | 0 | 0.0028 (0.009) | | **0.05** |
| Monocyte | 0.3082 (0.045) | 0.3054 (0.032) | 0.311 (0.055) | | NS |
| Neutrophil | 0.5164 (0.114) | 0.5438 (0.109) | 0.4898 (0.114) | | **0.01** |
| Natural Killer | 0.05041 (0.037) | 0.0344 (0.027) | 0.0659 (0.039) | | **4.00E-04** |
| T-Cell (Regulatory) | 0.0146 (0.021) | 0.0122 (0.018) | 0.017 (0.023) | | NS |
| NLR/100 | 0.3457 (0.348) | 0.4734 (0.421) | 0.2219 (0.195) | | **0.01** |

-0.1  0  0.1  0.2  0.3  0.4
Depleted in High Risk — Enriched in High Risk

**c.**

| | WHO Grade Enrichment | Recurrence Free Survival | Epigenetic Features | Immune-based Deconvolution | Bayley et al., 2022 Equivalency [n = 95] | Choudhury et al., 2022 Equivalency [n = 505] |
|---|---|---|---|---|---|---|
| **High Risk MNG** | **Tissue & LB** WHO Grades 2 & 3 | **Poor Prognosis** Tumor Tissue 5yr RFS: ~20% / LB Serum 5yr RFS: ~38% | **Tissue & LB** Hypermethylation of PRC-target gene CpG Islands — Transcription Inhibited | **Tissue & LB** High Neutrophil: Lymphocyte Ratio | **Tumor Tissue** MNG C: High CNV, NF2 losses, high recurrence rate | **Tumor Tissue** Hypermitotic: high RR, poor outcomes, high WHO Grade |
| **Low Risk MNG** | **Tissue & LB** WHO Grade 1 | **Favorable Prognosis** Tumor Tissue 5yr RFS: ~70% / LB Serum 5yr RFS: ~75% | **Tissue & LB** Hypomethylation of PRC-target gene CpG Islands — Transcription Allowed | **Tissue & LB** Low Neutrophil: Lymphocyte Ratio | **Tumor Tissue** MNG A: Low CNV, NF2 intact, low recurrence rate | **Tumor Tissue** Merlin-Intact: low RR, good outcomes, low WHO Grades |

Created with Biorender.com

**d.** MeLB Clinical Implementation

1. Detection of tumor.
2. Collection of liquid biopsy at either follow up or time of surgery.
3. Profiling of sample methylome. — Illumina Seq — DNA Methylation
4. Classification of sample using machine learning. — d-MeLB → p-MeLB
5. Implementation of machine learning output into tumor board decisions. — Patient #1: d-MeLB: 0.99 conclusion: MNG / p-MeLB: 0.46 conclusion: Low Risk

Radio-/Immuno-Therapies, Additional Surgeries → Active Intervention ← Disease Surveillance → Continuous Liquid Biopsy, Habitual MRI's

Created with Biorender.com

Finally, we explored the potential biological functions and diseases associated with these PGPs through gene set enrichment analyses. We identified that tissue- or serum-derived prognostically relevant PGPs were related to tumorigenesis processes (n = 38 genes), specifically related to meningioma (n = 5 genes), cell growth/proliferation/movement (n = 26 genes), cell cycle (n = 7 genes), and immune response (n = 21 genes), amongst others (Table 3, Supplementary Data 4).

Identified prognostic DMPs exhibited overall DNA hypermethylation in CR or high-risk samples compared to NCR or low-risk samples. We also observed, particularly in the regulatory regions of gene promoters associated with Polycomb repressive complexes (PRC), strong DNA hypermethylation. This DNA hypermethylation was detected across liquid biopsy specimens (serum and plasma), as well as tumor tissue specimens (Supplementary Fig. S5d–f).

**Fig. 3 | Clinicopathological and molecular characterization of serum from patients with meningioma predicted to present distinct recurrence risk outcomes through p-MeLB. a** Clinicopathological feature proportions and associated odds ratios (*p*-values: two-sided Fisher's Exact test; error bars: 95% confidence interval estimates) derived from the comparison between meningioma serum samples predicted to present high or low recurrence risks. Reference column depicts the mean proportion of each feature across the whole cohort. SB Skull-base, NSB Non-Skull Base, Y Yes, N No, GTR Gross Total Resection, STR Subtotal Resection, PD Progressive Disease, SD Stable Disease, NED Non-Enhancing Disease, CR Confirmed Recurrence, CNR Confirmed No Recurrence; Bolded features are those with observed statistical significance. **b** Immune cell proportions and associated mean differences derived from the comparison between meningioma serum samples predicted to present high or low recurrence risks (error bars: mean difference 95% confidence interval; *p*-values: two-sided *t*-test). Reference column depicts the mean proportion across the whole cohort. NLR Neutrophil-Lymphocyte Ratio. Bolded features are those with observed statistical significance. **c** Schematic summarization of observed clinicopathological and molecular features across samples (LB-serum and/or tissue) from patients with meningiomas predicted to present high or low risk of recurrence through p-MeLB. LB liquid biopsy, RFS Recurrence Free Survival, RR Recurrence Risk, MNG C Bayley Meningioma C group, CNV copy number variation, RR recurrence risk, PRC Polycomb Repressive Complex. **d** Schematic representation−clinical application of liquid biopsy DNA methylation-based diagnostic and prognostic classifiers in patients suspected to present meningioma. MeLB Meningioma epigenetic Liquid Biopsy, cfDNA cell-free DNA, d-MeLB and p-MeLB diagnostic- and prognostic- Meningioma Epigenetic Liquid Biopsy, respectively, MRI magnetic resonance imaging.

## Discussion

Genome-wide DNA methylation assessment provides an objective, robust and unbiased approach to define discrete molecular groups of CNS tumors. This approach overcomes the limitations and subjective biases associated with histopathological and grading classification approaches[8,10,11,15–18]. Detection of distinct DNA methylome patterns is reproducible and stable within and across diseases and allows for the fine-tuning of molecular subtyping associated with distinct recurrence and growth-prone behaviors in many tumors[7–11,16,18,26,28,40]. Capitalizing on this knowledge, several reports have shown that specific DNA methylation signatures identified in tumor tissue specimens are amenable to the development of machine learning classifiers able to accurately diagnose and prognosticate several tumor types and subtypes, including meningiomas[8,11,25,28,41,42]. However, we observed that these previous classifiers were not able to classify our liquid biopsy samples into their diagnostic or prognostic memberships, possibly due to their formulation being solely based on tissue-derived methylomes[8–10,16,18]. To circumvent this limitation, herein, we developed machine learning classifiers using meningioma-specific DNA methylation markers suitable to diagnose and prognosticate these tumors using either liquid biopsy or tissue specimens (Figs. 1f−h and 2b, c).

Confirming their tumor-of-origin specificity, these detected markers clustered together liquid biopsy (serum and plasma) and tissue specimens from patients with meningioma, while simultaneously distinguishing these tumors from other CNS entities, when applied to external and independent tumor tissue cohorts[13,32–36] (Fig. 1e). The d-MeLB signatures generated during our diagnostic model development presented an overall accuracy of ~85% to classify serum samples according to meningioma or non-meningioma memberships (Fig. 1f). These signatures were also able to correctly classify tissue samples with ~94% accuracy using additional linear machine-learning methods (Supplementary Fig. S4a). Altogether, the current findings corroborate our previous reports showing the viability to use LB-oriented classifiers for diagnosing CNS tumors[26,29]. Notably, this classifier allowed for the accurate identification of recurrent meningioma using serum samples (Fig. 1f−h), which could prove useful as an standalone or complementary noninvasive tool along with imaging to monitor these tumors[43].

Through p-MeLB classifier development, we identified risk-specific DNA methylation markers in serum useful for the stratification of serum or tissue specimens according to their recurrence risk, with an observed total accuracy of 87.7% across independent cohorts (Fig. 2b, c). In addition to the noninvasive application of p-MeLB, its accuracy is comparable to or even surpasses other individual benchmark methods evaluated in surgical specimens, such as Ki-67/MIB1 immunoexpression (AUC: 87.7%)[44], transcriptome-base markers (AUC: 0.81)[45,46] or composite scores involving multiple risk factors (AUC: 0.849)[47] with or without consideration for imaging features (AUC: 0.75-0.78)[48].

The observed agreement between the prognostic classification using p-MeLB or existing classifiers, further reinforced the validity and robustness of p-MeLB in assessing the likelihood of recurrence[10,16,18]. However, compared to the Nassiri nomogram[10], the p-MeLB model excelled in predicting the risk to recur in an independent meningioma tissue cohort (i.e., p-MeLB vs nomogram accuracies - within 5 years: 76.5 vs. 48%; after 5 years: 88.9 vs. 56.6%) (Fig. 2c, Fig. 3c, Supplementary Fig. S3c−f). Additionally, external samples predicted as high risk through p-MeLB classification exhibited significantly poorer overall survival, with an approximate 20% probability without a tumor recurrence after 5 years' time. These observed survival trends are concordant with recurrence-free survival rates reported for subtypes with higher propensities for progression by other authors (e.g., Bayley's MNG-C: ~45%; Choudhury's Hypermitotic:~35% and Nassiri high risk/grade 3-25%)[10,16,18] (Fig. 2d).

Additionally, considering the differing sensitivities of Choudhury's Hypermitotic and Merlin-intact subtypes to cytotoxic agents in preclinical studies (with decreased and increased vulnerabilities, respectively)[16], and the consistent alignment of p-MeLB classifications with high and low risk for recurrence in these subtypes, we suggest that p-MeLB has the potential to guide experimental therapeutic decisions.

Notably, the p-MeLB classifier presents some unique advantages compared to these existing models: (1) In contrast to the cross-sectional information utilized as input in existing models[16,18], our p-MeLB signatures were derived from longitudinal data. This was accomplished by comparing cases with confirmed recurrence or no recurrence over a minimum 5-year period of clinical and radiographic surveillance follow-up; (2) p-MeLB requires solely DNA methylation data as input, in contrast to other tissue-based meningioma classifiers which rely on the integration of methylomic data with clinicopathological features prone to subjectivity (e.g., extent of resection) or multiomic profiling, which may be financially detrimental for a potential clinical application; and (3) it is able to accurately predict outcomes when applied across different specimen sources (tissue, serum, and potentially plasma).

Overall, these findings suggest that the application of d-MeLB and/or p-MeLB classifiers could be a valuable noninvasive approach for diagnosing and distinguishing meningiomas from other mimicking diseases in preoperative assessments and possible monitoring tumor progression and treatment response through a blood draw (Fig. 3d). Additionally, they may complement traditional and advanced imaging approaches, such as radiomics[49], to provide a more comprehensive and accurate evaluation of meningioma status.

Through the integration of paired methylome and transcriptome data derived from meningioma tissue generated by Choudhury et al.[16], we identified genes whose expressions are possibly regulated by epigenetic control (Fig. 4c). Interestingly, many of the probes associated with these genes in meningioma tissue were also detected in serum specimens (Fig. 4d). Among these gene sets, we found enrichments for

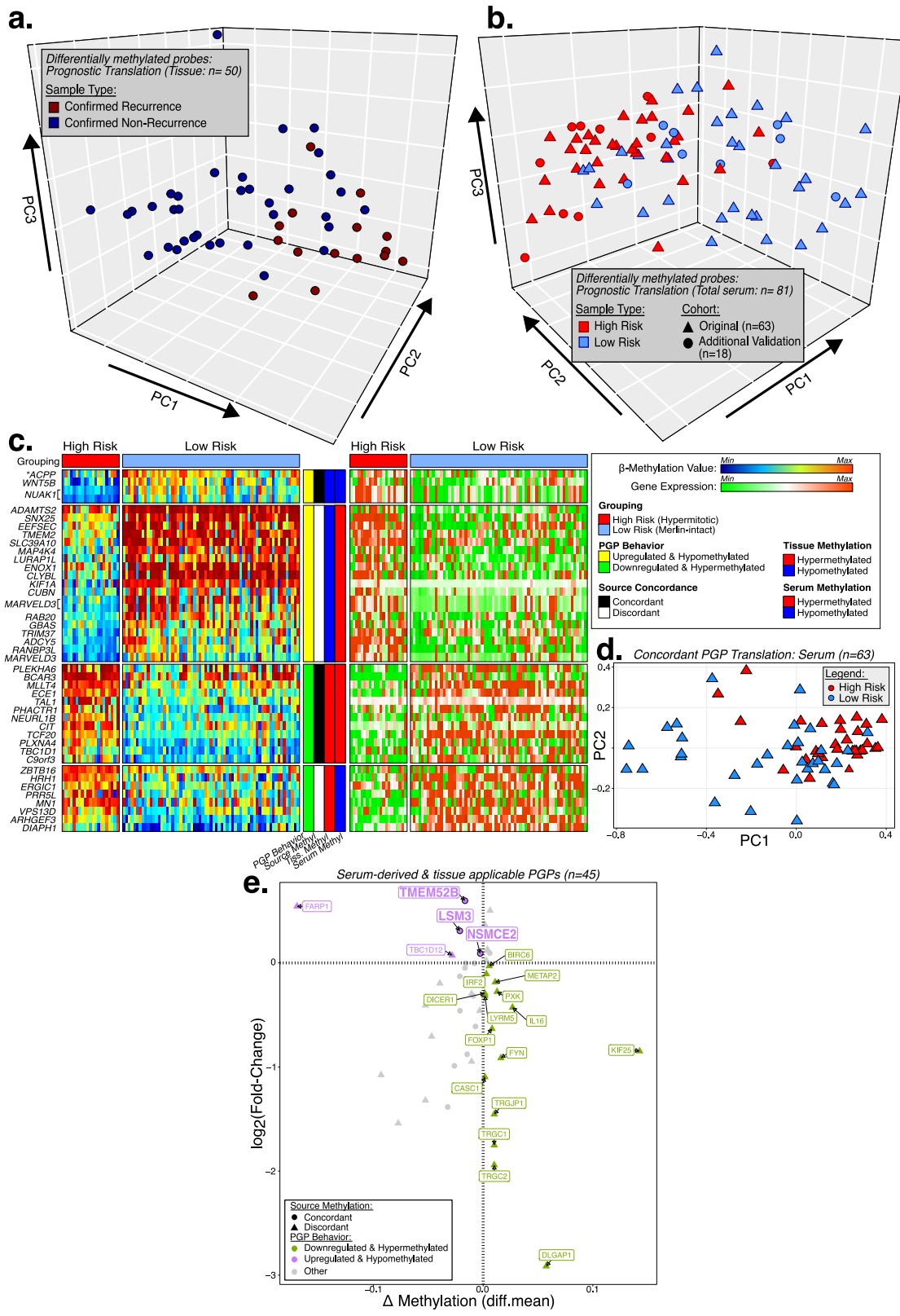

biologically relevant terms such as CNS tumor development, cellular growth/proliferation and movement, and prognosis (Table 3, Supplementary Data 4). We also detected distinct hypermethylation within regulatory regions of gene promoters associated with PRC across high-risk samples. DNA hypermethylation in promoter regions of this complex, as observed across multiple sample sources (serum, plasma and tissue), has been previously linked to malignancy in

meningiomas[33,50]. Altogether these results indicate that the identified signatures could be mechanistically involved in the recurrence risk of these tumors and could be used as prognostic markers detectable in serum specimens.

Most reported LB-oriented studies have used plasma instead of serum as a source of cfDNA to perform omics analysis. Herein, we mainly profiled serum, the sole blood component available in our

**Fig. 4 | Characterization and functional analysis (in silico) of recurrent-risk differentially methylated probes.** Principal component analysis depicting the tissue-derived recurrence risk group DMPs (High vs Low) and respective outcomes (Confirmed Recurrence vs. Confirmed No Recurrence) across (**a**) the tumor tissue cohort, (**b**) the original liquid biopsy serum cohort and an additional serum cohort of samples from patients with meningiomas. **c** Heatmap displaying methylation and expression levels of differentially methylated probes and differentially expressed target genes that are negatively correlated across high and low recurrence risk groups identified in an external molecular meningioma tissue dataset (Choudhury et al., 2022). *PGP Promoter-linked probe-gene pair. **d** Principal component analysis of liquid biopsy serum samples using the identified and concordantly methylated probe-gene pairs as input. PGP probe-target gene pair. **e** Scatter plot depicting serum-derived risk-specific probes that are also detected in meningioma tissue and respective target genes expression changes between high and low risk sample cohorts. PGP probe-target gene pair, Diff mean differential mean.

tumor bank at the period of our data freeze. Although certain molecular results (e.g., detection of somatic mutations)[51,52] could be impacted by the use of serum profiling due to potential dilution or contamination with genomic DNA derived from blood and other cells during the coagulation process, it does not seem to interfere with the detection of cell-specific cfDNA methylation markers as shown in this and other studies[26,29,41,53,54]. Additionally, even after considering potential dilution of tumor derived cfDNA in serum specimens, DNA methylation array platform (EPIC) or whole-genome sequencing are sensitive approaches to detect abnormalities in minute amounts of intact or fragmented cfDNA in liquid biopsy specimens (e.g., <1 ng)[55–57]. Notably, among our detected risk-specific probes, derived through serum- or tissue-based analyses, some of their targeted genes are implicated in immune response pathways. In serum this enrichment could arguably reflect contamination with cfDNA from lysed white

## Table 3 | Gene set enrichment analysis results using Ingenuity Pathway Analysis (IPA)

| Term | Geneset | # Of genes | p-value(s): $U_L$, $L_L$ |
|---|---|---|---|
| Diseases & Disorders | | | |
| CNS tumor related | Tissue | 38 | 4.6e−05, 0.006 |
| | Serum | 26 | 0.0028, 0.0155 |
| Cancer | Serum | 36 | 0.0012, 0.0175 |
| Tumor development | Tissue | 36 | 7.6e−05, 0.0012 |
| Meningioma-related | Serum | 5 | 1.47e−04, 0.011 |
| MN-1 related | Tissue | 1 | 0.0135 |
| Molecular and cellular functions | | | |
| Immune Response | Serum | 21 | 4.85e−04, 0.0178 |
| | Tissue | 1 | 0.00678 |
| Cell growth/proliferation | Tissue | 19 | 2.32e−04, 0.0284 |
| | Serum | 9 | 5.47e−04, 0.0145 |
| Cellular movement | Tissue | 17 | 5.31e−04, 0.0269 |
| | Serum | 14 | 0.0047, 0.0179 |
| Cell assembly | Tissue | 15 | 1.74e−04, 0.0282 |
| | Serum | 5 | 0.0011, 0.0163 |
| Cell death | Tissue | 15 | 0.0034, 0.0227 |
| | Serum | 12 | 0.0011, 0.0145 |
| Embryonic development | Tissue | 13 | 0.0022, 0.0132 |
| Cell cycle | Serum | 13 | 0.0018, 0.0163 |
| | Tissue | 7 | 1.26e−04, 0.0285 |
| CNS development | Serum | 11 | 0.0017, 0.0018 |
| Cell-to-cell signaling | Tissue | 10 | 0.0017, 0.0269 |
| | Serum | 10 | 4.56e−04, 0.0163 |
| Cell function | Serum | 6 | 0.0018, 0.0181 |
| Cell-mediated immune response | Tissue | 3 | 0.0034, 0.0285 |
| Cell development | Serum | 3 | 0.0012, 0.0018 |

Most relevant disease and biofunctions predicted by Ingenuity Pathway Analysis (IPA) to be regulated by mRNA expression profiles of high-risk hypermitotic meningioma samples (p-values: right-tailed Fisher's Exact Test).
*MN-1* menin-1, *mRNA* messenger RNA, *IPA* Ingenuity Pathway Analysis, $U_L$ Upper Limit, $L_L$ Lower Limit.

blood cells potentially introducing an immune bias into our signatures (Table 3). However, we gathered several lines of evidence to support the hypothesis that the immune-related findings we observed are genuine and associated with the presence of meningiomas. To formally address concerns about genomic DNA contamination in serum, we profiled and compared the methylomes of paired serum and plasma samples from an additional meningioma cohort. We found a high correlation in the genome-wide DNA methylation levels, estimated immune cell proportions, and diagnostic and prognostic classifications between the two blood elements in most samples (Supplementary Table S1). Additionally, meningioma-specific probes detected in plasma clustered together with their matching serum and tissue counterparts, confirming the specificity of these CpG probes regardless of the specimen source (Fig. 1e). Furthermore, we observed a significant difference in several estimated proportions of immune cells in comparison of whole blood and meningioma liquid biopsy (LB) serum samples (Supplementary Fig. S4d–l). Additionally, our d-MeLB model accurately classified whole blood and purified immune cells samples as non-meningiomas, indicating that our d-MeLB signatures are not biased towards spurious immune enrichment (Supplementary Fig. S2c, Supplementary Table S3). We observed that samples predicted to have a high risk of recurrence exhibited higher neutrophil-to-lymphocyte ratios (NLR), increased neutrophil levels, and reduced proportions of B-cells and natural killer cells (Fig. 3b). These alterations have been associated with poor prognosis in other tumors[45,58,59]. We also found that the immune compositions between serum and matching tissue were not significantly correlated, suggesting that the systemic immune or inflammatory response to the presence of meningiomas is distinct from the local immune response in the tumor microenvironment, consistent with findings from other studies[60–62].

Altogether, these immune-related findings in serum specimens appear to be authentic markers of a systemic inflammatory response to the presence of meningiomas with varying recurrence risk, rather than a spillover of the local immune response or contamination with DNA from white blood cells (Fig. 3b, Supplementary Fig. S5c, Supplementary Data 5). While confirmation with a gold standard approach such as flow cytometry is needed, our results suggest that this DNA methylation-based deconvolution approach could offer additional insight into our proposed prognostic classifications. It has the potential to stratify patients with meningiomas based on their immune landscape and guide future immunotherapy strategies[26,29,63–67].

Currently we are developing a user-friendly platform containing the diagnostic and prognostic classifiers, similar to the available tissue-based web tool detailed by Capper et al.[32]. We aim to have this webtool fully operational for research purposes in the near future. As we aggregate more serum and tissue methylome data along with clinicopathological information, we hope to refine our models before they are made available for potential clinical application.

In summary, we showed that blood-based specimens, specifically serum, are amenable for the detection of tumor-specific DNA methylation signatures. The identified signatures not only enabled differentiation between meningiomas and other intracranial entities but also showed accuracy in identifying meningiomas with distinct recurrence risks. Potentially, these signatures may serve as a valuable surveillance tool for detecting meningioma recurrence during follow-up. The successful clinical implementation of these DNA methylation-based

classifiers will refine meningioma recurrence risk stratification at the time of diagnosis and possibly during follow-up, ultimately impacting management and outcomes of these patients. Our machine learning classification approach, based on methylome analysis, has the potential to be extended for the diagnosis and prognostication of a broader spectrum of tumors using liquid biopsy-derived specimens.

## Methods
Our research complies with all ethical regulations within our Institution. This project was approved by the Institutional Review Boards (IRB) and patients consented to have their specimens used for research purposes (Henry Ford Health (HFH): IRB#12490; University of Sao Paulo (USP): IRB#1572/2016).

We collected archival serum from 204 patients who underwent resection of meningiomas (MNG group) and other CNS entities and controls (non-neoplastic diseases) at the Neurological Surgery Department at Henry Ford Health from 06/2011 through 08/2019, namely 'original' cohort. We also retrieved and analyzed meningioma tissue methylomes generated internally at Henry Ford Health ($n = 31$), and provided by the Department of Neurosurgery of the University of Sao Paulo ($n = 72$), or from publicly available repositories ($n = 900$)[16,18,32,33,50,68]. Longitudinal follow-up information was available for 50 tissue (Henry Ford Health and University of Sao Paulo) and 25 liquid biopsy serum collections (Henry Ford Health). Serum specimens collected at recurrence were available for 19 meningioma collections (two paired with serum collected at first/initial surgery).

Besides these cohorts, we collected an "additional" MNG cohort (namely, validation i & ii) consisting of 69 archival tissue and blood-derived liquid biopsy collections obtained between 12/1999 and 07/2021 at Henry Ford Health: 8 paired tumor tissue and liquid biopsy (serum and plasma) pairs, 9 pairs of tissue and serum, 2 paired serum and plasma, and 23 individual tissue collections. Longitudinal follow-up data attributed to this cohort was available for a subset of samples (tissue, $n = 27$; liquid biopsy serum, $n = 8$; liquid biopsy plasma, $n = 4$). Simultaneous collections of tissue and serum samples were available for 25 MNG patients.

Additional collections of non-neoplastic diseases ($n = 6$) were also profiled for expansion of the control arm of this study. Information characterizing internal and external cohorts are displayed in Tables 1 and 2.

Congruent to definitions reported by others[10], meningioma recurrence was defined as tumor growth/progression or additional surgery following gross or subtotal resection through review of the immediate postoperative imaging and/or information found in medical records during follow-up (person-time), across initial or recurrent collections (namely Confirmed Recurrence (CR) group, Henry Ford Health and University of Sao Paulo tissue and liquid biopsy serum cohorts; $n = 84$; Table 1). Non-recurrent MNG was defined as the absence of growth/progression in any post-surgical MRI or medical reports or absence of further tumor resection across a minimum attributed follow-up of 5-years (namely Confirmed No Recurrence [CNR]; Henry Ford Health and University of Sao Paulo tissue: $n = 17$; Henry Ford Health liquid biopsy: $n = 9$; Table 1). In order to ensure precise categorization, serum samples or publicly available tissue samples without available follow-up information were labeled as either high risk or low risk for recurrence through the prognostic classifier (p-MeLB). We also retrieved publicly available methylomes, some paired with transcriptome data (RNA-sequencing or microarray) from meningiomas (Table 2)[16,18]. To perform correlative analyses between our prognostic (p-MeLB) classifier with others, we excluded samples from the Bayley cohort[18] which were not correctly classified by their algorithm ($n = 15$), at the request from the authors. Updated clinical and follow-up information was kindly provided by the Bayley group[18]. Within the Choudhury cohort[16], we excluded samples which were not fully annotated across provided clinical data ($n = 60$). Prognostic

classification of our internally generated samples through Nassiri et al.[10] nomogram was kindly performed by the Nassiri group.

### DNA isolation, quantification, quality control, DNA methylome data generation and preparation
Extracted DNA from meningioma-derived specimens (serum, plasma and tissue) were bisulfite converted using the Zymo EZ DNA methylation kit as specified by the manufacturer (Zymo Research, Irvine, CA, USA) and profiled using an Illumina Human EPIC array (EPIC) at the USC Norris Molecular Genomics Core Facility[26]. Prior to profiling, the isolated DNA was restored using a restoration kit provided by Illumina. This allowed us to restore fragmented DNA and concentrate the low yield[26]. Bisulfite converted DNA samples were recovered in a 10ul volume, and 1ul was used to evaluate bisulfite conversion completeness and recovery[26]. DNA methylation was profiled using the Illumina Human 850k (EPIC) and a matching subset using whole genome bisulfite sequencing (WGBS). Publicly available tissue methylome was profiled using 450k (HM450k; tissue)[13,32–36,69] or 850k (EPIC)[16,18,31,70] arrays (Table 2). Data quality of the methylome data was assessed with the R-based graphical user interface shinyMethyl v3.16 (details in Supplementary Methods)[71].

### DNA methylation preprocessing
DNA methylation array (EPIC) data were preprocessed (e.g. removal of masked probes and SNP) using the minfi package, as detailed in Supplementary Methods[72,73]. Before downstream analysis, liquid biopsy methylome data was examined for potential batch effects regarding plate number and extraction dates using commonplace methodologies (ComBat v3.20.0- sva). Tissue-based MNG methylome data was corrected for batch effect by institution prior to t-distributed stochastic neighbor embedding (t-SNE) visualizations. For prognostic classifier derivation, only probes common between 450 K and 850 K arrays were selected to maintain retrospective cohort applicability ($n = \sim339$k CpGs). For technical validation, we profiled the methylation levels of a subset of meningioma LB samples using whole-genome bisulfite sequence (WGBS) as detailed in Supplementary Methods.

### DNA methylation exploratory analysis
**Unsupervised and supervised analyses.** We explored methylome patterns across all serum samples (CNS tumor types and non-neoplastic controls) using standard unsupervised approaches and visualizations, described in detail in Supplementary Methods. To identify MNG molecular groups, we applied hierarchical consensus k-means clustering to the most variant methylated probes ($n = 1000$) across MNG serum specimens and selected the optimal number of clusters based on statistical parameters such as the Cumulative Distribution Function (CDF) and the Calinski-Harabasz curve[74].

**Supervised analyses.** In order to identify Methylation Epigenetic Liquid Biopsy (MeLB) probes (serum-derived) or tissue-derived MNG-specific differentially methylated probes (DMPs), we performed supervised comparisons between MNG- and non-MNG group methylomes (serum: 44 primary and 19 recurrent MNG, 141 non-MNG; tissue: 326 MNG; 367 non-MNG). To identify prognostically relevant probes, we conducted comparison between predicted high and low risk for recurrence groups (31 high risk and 32 low risk MNG) in the serum or confirmed recurrence (CR: $n = 35$) and non-recurrence (CNR: $n = 15$) groups in the tissue.

To reduce the potential for capturing background noise, we used volcano plot visualization techniques to guide selection of the DMPs which presented significant FDR adjusted $p$-values and mean methylation differences across comparisons within a variable range reported in the literature (differential mean DNA methylation $\geq 10$–$15\%$; Wilcoxon rank sum tests; $p$-value$_{FDR} \leq 0.05$)[75–78].

Each DMP was mapped to their CpG genomic location previously defined as CpG islands (CGI), shores, shelves, and open sea regions[64] and to their putative target gene using EPIC manifest (hg38). Enhancer elements were defined using the GeneHancer database (hg38) provided by the UCSC Genome Browser[79]. Promoter elements were defined using GENCODE v.31 annotations, with consideration of CpGs located 200 bp up/downstream from the target gene[80].

**Methylome-based predictions—Random Forest machine learning approach.** To investigate the potential diagnostic and prognostic applications of MeLB, we used the random forest machine-learning (ML) approach to generate a binary classifier to differentiate MNG vs non-MNG specimens and low vs high recurrence risk groups using serum cfDNA- and/or tissue-derived DNA methylation-based signatures. Specific feature selection processes are detailed in Supplementary Methods.

Cohorts were randomized through machine-driven processes into sets encompassing representative and proportional samples of each comparison group: 1) the discovery set used to construct the classifier, further subdivided into 1a) a training set for the identification of relevant signature sets and algorithm training and 1b) a model selection set, used to evaluate the resulting classifier's performance, or 2) an Independent validation set, not involved in any of the previous development steps, used to validate the finalized classifier. Following the completion of classifier formulation and independent validation, we included an additional set of MNG & Non-MNG samples to further validate the generalization of our diagnostic classifier (Additional validation set).

Supervised feature extraction processes were automated within machine learning construction and specialized for each classification task, as detailed below, in efforts to isolate diagnostic or prognostically relevant signatures, and reduce the potential for the training classifiers using noisy signals.

The performances of both diagnostic and prognostic prediction models were assessed using Matthew's correlation coefficient (MCC), which measures the quality of a binary classification by the agreement between predicted and actual (observed) values (ranging from +1 to −1, i.e. perfect agreement [perfect prediction] to total disagreement [poor prediction][81] and the Clinical Utility Index (CUI), interpreted as excellent, good, satisfactory or poor when values are ≥0.81, ≥0.64, ≥0.49 or <0.49, respectively. This index is primarily used to express the relative benefit of using our classifiers, compared to use of an optimal test, when making clinical decisions (CUI positive [+], and CUI negative [−])[82,83].

**Meningioma diagnostic classification—Diagnostic MeLB (d-MeLB).** In diagnostic classifier construction, following exclusionary measures (recurrent gliomas, inflammatory non-neoplastic diseases), serum samples were randomly assigned to discovery ($n = 117$) or independent validation cohort sets ($n = 30$), both encompassing analogous proportions of meningioma (initial and recurrent) and non-meningioma serum specimens. The discovery cohort was further randomly partitioned into training (80%) and model selection sets (20%). To instill inherent MNG-specificity to the identified signature, we performed dimensionality reduction of the entire genome through two distinct methods: 1) conducting genome-wide supervised analysis between internally-profiled MNG tissue specimens ($n = 31$) and publicly available nontumor control specimens ($n = 21$)[31], selecting CpG probes which exhibit differential methylation between the two groups according to a randomly selected significance measure, namely tumor-specific DMPs ($n = $ -256 k; Wilcoxon rank sum test $p$-value range: $1e^{-4}$–0.05); and 2) among the tumor-specific DMPs, selecting those CpG probes which possessed the greatest DNA methylation level

similarities between matching training set serum and tissue samples, namely similarly methylated probes (SMPs, $n = 7659$ CpGs; range: mean difference ≤ 0.1–0.2%). To explore the efficacy of our reduction technique and further solidify the MNG-specificity built into our SMPs, we employed unsupervised t-distributed stochastic neighbor embedding (t-SNE) across a wide array of tumor types, some not included within our liquid biopsy cohort. The entire cohort included within the t-SNE was derived from both external ($n = 2038$, details in Table 2)[13,32–36] and internal ($n = 229$) collections.

Further specialization of the DNA methylation signature was completed through comparison of untreated MNG (i.e., no pre-surgical radiation; $n = 27$) and non-MNG specimens (excluding glioblastomas, which were appreciably distinct across genome-wide visualizations; $n = 32$) across the aforementioned SMPs, to identify meningioma-specific DMPs with no potential for treatment-related epigenetic modifications. This final set of signatures was named diagnostic-Meningioma Epigenetic Liquid Biopsy (d-MeLB, range number: 20–30 CpGs; Wilcoxon rank sum test, $p$-value$_{FDR}$). Notably, we introduced variability across the machine-generated classifiers through randomization of parameters across iterations, including tissue-based nontumor differential significance, matching serum/tissue similarity score, sampling of non-MNG groups and final signature set size.

Using d-MeLB, we generated a classifier using the function 'train' (caret, v6.0.94) with 1000 decision trees and 10-fold cross validation conducted across the training set. To guide selection of the classifier, we applied the classifier to the model selection set and selected the score cutoff which optimized the relationship between true positive and false negative rates, defined through inspection of associated receiver-operating characteristic (ROC) curves, namely diagnostic or d-MeLB score. The selected classifier was validated using our original independent validation set ($n = 93$), including the excluded recurrent gliomas and inflammatory non-neoplastic diseases ($n = 63$). Further validations of the classifier were conducted across additional meningioma liquid biopsy-based validations, including both serum ($n = 19$) and plasma ($n = 10$) collections. Finally, we compared our diagnostic random forest-based classifier with other machine learning methods, as detailed in Supplementary Methods.

To investigate whether our d-MeLB signature was suitable for the diagnosis of tumor tissue samples (profiled through EPIC array), we constructed a simplified linear discriminant analysis (LDA) based machine learning classifier using MNG ($n = 138$; Henry Ford Health & University of Sao Paulo), intracranial mesenchymal tumors ($n = 20$; GSE164994)[70], and non-neoplastic collections ($n = 21$; GSE111165)[31]. These samples were randomized into discovery (80%) and independent validation (20%) sets, with discovery samples used to train and construct the LDA classifier, and the validation set further expanded with external collections of gliomas ($n = 16$; GSE147391), MNG ($n = 110$; GSE189521)[18], and internally profiled pituitary tumors ($n = 14$). Clustering of the total tumor tissue independent validation set ($n = 176$) across the d-MeLB signature was visualized using principal component analysis, and performance measures were calculated, as described.

**Meningioma prognostic prediction—Prognostic MeLB (p-MeLB).** To investigate the application of MeLB as a prognostic tool (p-MeLB), we compiled MNG tissue methylome data from confirmed recurrent (CR, $n = 35$) and confirmed non-recurrent (CNR, $n = 15$) collections, as defined in the Patients section. Machine-driven randomization assigned the total tissue cohort into training (CR, $n = 24$; CNR, $n = 10$) and independent validation sets (CR, $n = 11$; CNR, $n = 5$). Then, a series of machine-driven and sequential supervised analyses comparing methylomes across specific meningioma tissue prognostic groups (CR

and CNR) and/or serum k-clusters (k1, k2, k3 and k4) were implemented to define serum-applicable prognostic methylation signatures. The defined signatures were utilized as input to a machine learning random forest algorithm, generated through use of the function 'train' (caret, v6.0.94) with 1000 trees and 10-fold cross validation conducted across the training set. Signature set derivations and subsequent generation of the random forest algorithms were repeated across 1000 iterations, with each concluding in storage of the signature, training set classifications, and associated out-of-bag (OOB) error. Following 1000 iterations, we selected a p-MeLB classifier with an optimal score cutoff based on the smallest OOB error. Finally, the chosen classifier was validated, using the independent validation cohort ($n = 16$) and an additional set of internal and external samples meeting our criteria for definition of CR (Henry Ford Health: 29 tissue, 8 serum; Bayley et al.: 20 tissue)[18]. The classifier was then applied across the liquid biopsy sample cohort for stratification by recurrence risk.

**Comparison between published prognostic and p-MeLB classifiers.** We evaluated the agreement of p-MeLB predictions with prognostic groups involving the analysis of methylomes combined with additional molecular and/or clinicopathological features defined by previously reported tissue-based classifiers formulated in Nassiri et al.[10], Choudhury et al.[16] and Bayley et al.[18]. Through the provision of our DNA methylation data, the extent of resection and WHO Grade, our internal primary meningioma tissue cohort ($n = 69$) was classified according to a 5-year recurrence risk prediction nomogram developed by Nassiri et al.[10] and compared resulting classifications with p-MeLB's conclusions. We also applied the p-MeLB classifier across publicly available methylome data provided in Choudhury et al.[16] and Bayley et al.[18], following appropriate exclusions (see Patients and Methods), and explored the relationship between p-MeLB predictions and purported DNA methylation prognostic groups (Choudhury et al.: hypermitotic, Immune-enriched, Merlin-intact[16] and Bayley et al.: MenG A, MenG B and MenG C[18]). To circumvent the general lack of attributed person time for each sample in the publicly available cohorts, we generated local freedom from recurrence (LFFR) Kaplan-Meier curves across Choudhury methylation-based groups[16] and correlated the resulting p-MeLB risk scores with their reported outcome data (Cohen's unweighted kappa coefficient [κ] and Spearman's correlation coefficient [ρ]).

**Identification of biologically relevant and equivalent DNA methylation markers between tissue and serum.** To identify biologically relevant and equivalent DNA methylation markers between tissue and serum, we used an in silico functional analysis approach, capitalizing on the availability of tissue-derived and paired methylome and transcriptome analysis of meningioma provided by Choudhury et al.[16] ($n = 185$), as detailed in Supplementary methods.

**Methylation-based Deconvolution.** We applied previously described and validated DNA methylation-based methodologies to deconvolute the relative contribution of immune cell types to a given liquid biopsy and tissue specimen (package: MethylCIBERSORT v0.2) (details in Supplementary Methods)[39].

**Clinicopathological and molecular features across MNG subgroups.** We analyzed the distribution of clinicopathological and molecular features (categorical and continuous) across serum- and tissue-derived MNG specimens according to k-clusters and high and low recurrence risk groups, as detailed in Supplementary Methods.

**Comparison between serum and plasma specimen-derived DNA methylation profiles.** To compare results obtained through serum- and plasma-derived specimens, we correlated genome-wide DNA methylation levels (EPIC Array) and the estimated immune and non-immune cell proportions across a paired serum and plasma samples set ($n = 10$ pairs).

## Statistical analysis

All data processing and statistical analyses were completed using R (3.6.1). Non-parametric two-sided Kruskal-Wallis and Wilcoxon rank-sum tests and multiple testing adjustments (e.g., FDR) were used to identify significant DMPs and discrete variable differences across binary group comparisons. Machine-learning classifiers were formulated using a random forest (RF) algorithm. Receiver operating characteristic (ROC) curves were utilized to estimate the predictive power for each iteration of the diagnostic RF classifier (false positive rate (1-specificity [SP]), true positive rate (sensitivity [Se]). Concordance across classifiers was estimated using Cohen's unweighted kappa coefficient (κ); correlative relationships were quantified using Pearson's correlation coefficient (ρ). Statistical significance related to differences in survival probability for estimated risk groups were established through $p$-values (log-rank test; $p \leq 0.05$). K-nearest neighbor imputation machine learning was used in the event of missing DNA methylation β-values found within tumor tissue methylomes across downstream analyses (t-SNE).

## Reporting summary

Further information on research design is available in the Nature Portfolio Reporting Summary linked to this article.

## Data availability

The raw cfDNA methylation intensity data files (EPIC Array;.idat), as well as generated classifiers, have been deposited to Mendeley Data under accession https://doi.org/10.17632/zrc982rvjm.2 [https://data.mendeley.com/datasets/zrc982rvjm/2][84]. The Whole genome bisulfite sequencing (WGBS) files generated in this study have been deposited to the Sequence Read Archive at the NCBI under accession code PRJNA932734. Additional tumor tissue molecular data analyzed in this study was obtained from Gene Expression Omnibus (GEO) under accession codes GSE42882, GSE109381, GSE85135, GSE189521 GSE183656, GSE115783, GSE54415, GSE164994, GSE147391 and GSE111165. Other sources of tissue molecular data employed in this study also include Mendeley Data under accession https://doi.org/10.17632/5pzd2rg5ys.2 [https://data.mendeley.com/datasets/zrc982rvjm/2][85] and The Cancer Genome Atlas's GDC data portal [https://portal.gdc.cancer.gov/], as detailed in Table 2. Source of our data is provided with this paper. Public data repositories employed throughout this paper include GENCODE (GRCh38.p12) https://www.gencodegenes.org/human/release_31.html, Infinium Annotation Manifests (hg38) https://zwdzwd.github.io/InfiniumAnnotation, Ensembl (hg38) http://useast.ensembl.org/index.html, and GeneHancer (hg38) https://www.genecards.org/. Gene-ontology and gene-set enrichment was performed using Ingenuity Pathway Analysis (IPA) https://digitalinsights.qiagen.com/products-overview/discovery-insights-portfolio/analysis-and-visualization/qiagen-ipa/. Source data are provided with this paper.

## Code availability

Source codes necessary for production of the diagnostic- and prognostic-Meningioma epigenetic Liquid Biopsy (dMeLB, pMeLB) classifiers are available at GitHub [https://github.com/gherrgo/eLB-Random-Forests.git]. Complementary *.idat files from tumor tissue and liquid biopsy specimens required to construct, validate, and apply our random forest classifiers are available under the Mendeley Data Accession https://doi.org/10.17632/zrc982rvjm.2 [https://data.mendeley.com/datasets/zrc982rvjm/2], as well as within the supplementary data (Supplementary Data S1 - Clinical Information; Source Data).

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

## Acknowledgements

The authors are grateful to the HFH patients and USP patients who consented to the usage of meningioma samples for research purposes. We thank Heather Mengel, Nicolette Bicknell and Lisa Scarpace for their research and clinical administrative support; Simona L. Cazacu for helping with tumor collection. and Susan MacPhee for proofreading the manuscript. AVC and HN are supported by NIH R01CA270365 and Hermelin Brain Tumor Center and Henry Ford Hospital internal funds (A30935, A30957, H10345, H10346).

## Author contributions

A.V.C. and H.N. conceived the study; supervised all analyses and the manuscript writing; G.H. performed all the analyses, developed the classifiers, designed all figures, tables and schematics under A.V.C. and H.N. supervisions; G.H., R.S., T.M.M., T.S.S., M.S.M., N.M., V.F. collected molecular/clinical data generated in house and publicly available and collaborated on the development of the classifiers; J.M.S., J.Z., C.E.C., J.P., G.G.P.G. procured, extracted, analyzed and interpreted clinical and imaging data; K.N., B.T., K.M.T., R.N. obtained patients' consents, collected and handled the tissue and liquid biopsy specimens, and maintained the tumor bank at the Hermelin Brain Tumor Center (HBTC); A.C.D. and T.W. supervised all the functional operations of the tumor bank at the Hermelin Brain Tumor Center (HBTC); L.A.H. and A.T. procured the specimens and extracted molecular data; L.M.P. collaborated with the statistical analyses; D.C. and A.M. reviewed pathological findings and grading; K.P.A., I.Y.L., A.M.R., M.R., T.M., S.K., D.P.C.T., G.G.P.G., C.G.C.Jr. and J.R. obtained the specimens during surgery; provided surgical and clinical information; D.W. profiled the DNA methylation of all samples; A.V.C., H.N., G.H. and J.M.S. prepared and revised the manuscript; all authors contributed to the revision of the manuscript; G.H. and J.M.S. contributed equally as first authors and A.V.C. and H.N. contributed equally as senior authors.

## Competing interests

The authors declare no competing interests.
