## [Peer Review File · Nature Communications]

REVIEWER COMMENTS

Reviewer #1 (Remarks to the Author): expertise in brain cancer sequencing

The authors have conducted a study for which primarily they have produced cfDNA Methylation profiles from serum samples taken from patients with meningioma alongside various non-meningioma controls and relevant tissue samples. They create and validate a number of signatures/classifiers which variously distinguish between meningioma and non-meningioma, high-risk/low-risk, different sub-types. They postulate and provide some evidence for an underlying immune-biological difference between subtypes/risk groups. Overall I feel positive about this study and I think it makes a worthy addition to the literature.

Whilst the notion of detecting cancer specific methylation patterns in cfDNA itself is not especially unique Liu, J Mol Cancer 20, 36 (2021), Luo, B. BMC Med 20, 8 (2022), its fair to say that high-quality case studies are few and the field is relatively speaking in its infancy. This may doubly be said of CNS tumors for which the working assumption often seems to be that the CSF is most relevant as opposed to the plasma. Where this study goes a little further than most is in the detail with which the authors attempt to extract prognostic patterns. In summary, the concept and experimental design is not 100% unique but nevertheless a reasonable argument could be made for relevance of the approach as a proof of concept across a wider range of tumor contexts and most certainly with the meningioma field.

In general, the work supports the conclusions and claims with some caveats outlined below. Likewise, I don't believe there are any major flaws in data analysis or interpretation that should prohibit publication and I do think that in the methodology is appropriate but I do have some caveats and comments which in my view should ideally should be addressed prior to publication.

1: It seems that a key criteria for how this approach is beneficial is the extent to which prognostic performance exceeds that available through existing or even non-biological means. I can see that this is somewhat addressed but I would prefer if this were more clearly evidenced, quantified and stated as an output of the paper. There is perhaps more of a focus on the extent the new risk signature tracks with the old.

2: If a measure of performance is the ability to distinguish between other non-meningioma types is the range of tumour types LGG, etc a little to limited to support the conclusions. Please consider what could be done or what caveats could be given that would address this?

3: The recurrent samples classify with an accuracy of ~53% is this expected? problematic? It seems not to be commented on again?

4: saying that DMPs annotate in intergenic regions overlapping with enhancer regions and giving a % may not be saying much without referring to the location of non-significant CpGs as 450/850K arrays are non-randomly designed and cfDNA non-randomly fragments.

5: putative DMP-target gene pairs as an input into IPA or other gene ontologies is always a tricky prospect (no guarantee of a correlative/functiona link) and I could just make the slightly non-scientific point that the heatmaps in 1D, 2C, 4C don't appear particularly convincing. Suggest either find a different way or strongly consider removing I don't believe they add much to the story the conclusions don't rely on them especially and they may be spurious.

6: I would query if its totally clear from the text if you consider deconvolution of the liquid biopsy profiles as indicative of the TME it seems that sometimes you are indicating that. If so that on the face of it seems quite hard to justify?

7: You say concordant but non-significant. Seems a long winded and possibly a bit misleading way to say there isn't a relationship as far as we can show.

8: The purpose and scope of the paper would be far better justified if you could outline how you might expect this to be used in clinical practice? When would LB take place how would treatment be altered, in what sense is this better or more practicable than simply dealing with bulk tissues samples.

9: To me the figures are very cramped often the text is too small to see without zooming in and out and it is just generally very busy. I think that the balance between what is in the main and what is in the supps isn't right. Additionally the colour scheme is quite loud and intense and divergent and it gives a slightly stroboscopic and busy effect. Normally I wouldn't complain but I think in this case I do think it inhibits the clarity of your message and makes it a harder read than it needs to be. The figures could use a bit of a rejig and some serious streamlining in my opinion and then by definition the results section text will be clearer too. Please consider carefully whether each track on the heat map or deconvoluted cell type etc is really necessary for the main message.

10: The signature/classifier generation is complex to follow. I think I just about could recreate but I do wonder if some posted code, explanation a GitHub or some fuller description might help. One issue in particular is important for me to understand. It may be there and I can't see it but the feature selection when dealing with the training set, are features reselected fresh each time during each iteration of any

cross-validation. This seems an important issue for the generalisation of the signature (although I note the independent validation, but also note sample size is relatively small and may not be reflective of general population). Could this be clarified/justified?

Other comments:

There seems to be results in the supplementary text which is a bit unorthodox and really should be moved to the main if it is still deemed important.

S1 and S4 seem to be introduced out of order?

Reviewer #2 (Remarks to the Author): expertise in cfDNA methylation profiling

The authors describe an analysis of biomarkers related to detection of disease and prognostic outcomes using DNA methylation.

The manuscript is very nicely presented, but I do have two significant concerns relating to the use of serum for their liquid biopsy profiling; and the lack of secondary wetlab validation by bisulfite resequencing of their CpG biomarkers, as detailed below

1) The use of serum

First, the authors use of serum for their liquid biopsy profiling methods is somewhat problematic. For example, in somatic mutation profiling of cfDNA isolated from cancer patients the use of Streck, Paxgene ccfDNA, and other tubes is exclusively used because they both prevent clotting and stabilize whole blood samples from lysis; conversely, the use of serum is avoided in cfDNA testing because serum samples result in high levels of buffycoat DNA due to lysis of the PMBCs which contributes significant levels of nontumour DNA to the extracted cfDNA, thereby reducing the amount of actual tumour material in the isolated DNA and the ablating the sensitivity of any genomic profiling methods.

This probably explains the reason for the enrichment in genes associated with Immune signalling in the authors analysis (line 299); as well as the estimation of the percentage of immune cells in the methylation data (line 318, supp figure 3b) which shows a ~50% neutrophil percentage - which is the exact proportion typically present in a normal sample of white blood cells; and again significant enrichment of sites related to immune function for their risk prediction (line 408). Indeed, each time mention of genes and pathways identified as differentially methylated for their biomarkers are mentioned, there is recurrent representation of immune system related genes and pathways.

Of course there is certainly the possibility that the signal represents a bone fide biomarker, however the difficulty is the fact that they are getting enrichment for CpGs related to immune genes in a sample from the blood, for which there would certainly be significant buffycoat gDNA contamination due to the use of serum.

2) The lack of secondary wet lab validation of their markers

Next, while the authors have applied adjusted p-values to their filtering, casting an eye over the list of differentially methylated cytosines I note that the mean difference between their cohorts hovers ~between 12-20%, which is near the threshold of noise on the beadchip arrays in my experience. In standard beadchip analysis a minimum mean difference of 20% is usually applied because of this noise. Moreover, I know from first hand experience that biological outliers in data easily skew means and adjusted p-values these and make them look significant. Moreover, within this analysis I also have concerns about multiple t-testing, despite the use of FDR correction. In this case the use of EPIC array constitutes ~850k cytosines but the authors used supervised analysis to identify only 130 differentially methylated probes – a small number which could be attributed to random chance due to high sampling, despite the prima face low p-values.

Indeed, if I understand their methods correctly, the initial identification of CpGs employed the entire cohort; only after specific CpGs were identified was the dataset split into separate training and validation cohorts to build their classifier. While I understand the motivation of performing discovery first then splitting the cohort for ML training, analysis done in this way significantly biases the validation dataset towards the appearance of significance.

As such, the only way to effectively address this is through secondary wet lab validation, which is the next major weakness of the current manuscript. While the authors have separated their data when generating their classifier, given both the use of serum and the shallow difference in mean methylation, secondary validation by bisulfite resequencing of their diagnostic and prognostic CpGs needs to be performed (more on this below in my specific points to address).

Specific reviewers points to be addressed

1. While I do have significant technical concerns about the use of serum, there is certainly the potential that despite the fact their serum samples were almost certainly contaminated with buffycoat DNA, there could be an underlying diagnostic/predictive signature. However, to be accepted for publication significant validation work needs to be undertaken.

First, to accept the manuscript, I believe technical validation using bisulfite resequencing must be performed on all serum sample originally used (for which there is sufficient material left), to verify the signal they are getting is not attributable to stochastic variation on the array platform. A good resequencing protocol only takes minimal sample input, and I have successful run effective resequencing applications with 10ng of input DNA, although more is certainly better.

Second, additional bisulfite resequencing needs to also be performed on an **independent** set of serum samples not originally used for discovery and training. In the absence of both sets of validation data I have significant concerns about their DMPs.

2. Building on this, the authors must incorporate a significant section in their results and discussion on why they elected to use serum; and why despite the (most certain) buffycoat contamination in the isolated DNA they believe they are able to derive a tumour signal. Specific points which need to be touched on are their expected levels of tumour DNA in their isolated liquid biopsy sample; and the overall sensitivity, accuracy, and precision of the beadchip platform to resolve those differences. If they are able to empirically determine the amount of tumour DNA in their sample using, for example ddPCR against known somatic mutations, that would also be of significant benefit.

3. Regarding the ML random forest classifier, the authors must export the classifier and supply it as a supplementary file with the manuscript; together with detailed instructions on how researchers can import and use the classifier with their own data (in either R or python). I know this can easily be done in python in sklearn; if the authors have used R or another package they will have to investigate different strategies for exporting and importing their RF model.

I am requesting this as I do not believe it would be easy to replicate the authors RF classifier based on their methods description starting from the raw data, and for the purposes of reproducibility and enabling other researchers to validate their results, this is an absolute necessity.

Reviewer #3 (Remarks to the Author): expert in meningioma

The authors carried out methylation profiling on liquid biopsies from 208 patients with meningiomas (44 primary, 19 recurrent MNG) or other CNS and non-neoplastic diseases (145 non-MNG samples) to determine whether they are useful as a non-invasive method of diagnosing patients and determining their likelihood of meningioma recurrence. Using a combination of Illumina 450k and 850k arrays, they identified 130 significant meningioma-specific differentially methylated probes in genes in pathways involved in retinoid X receptor activation and the immune response that could be used to distinguish meningioma patients from glioma, pituitary tumors, other CNS tumors and non-neoplastic diseases, using a non-invasive assay.

Comments to the authors

I am not an expert on the statistical methodology, but the data look impressive to me. I just have a couple of questions about the data analysis:

p6, line 119, The methods section states that only probes in common between Illumina 450k and 850k were used for downstream analysis, but I couldn't find the number in the text. How many probes were assessed altogether?

p14 line 303-4, for the d-MeLB prediction model in Serum cfDNA methylation samples, I believe it is very unusual to obtain an AUC of exactly 1.0. Was this really the case?

Minor comments

Gene names should be italicised.

I'm not sure if it was part of the anonymization process, but the Mendeley Data DOI needs to be added.

Check for typos e.g. p9 line 208 replace "biological" with "biologically"; p10 line 210 insert "relevance" after "biological".

Reviewer #4 (Remarks to the Author): expert in machine learning

In this work, the authors explored the epigenetic signatures (DNA methylation) of patients with meningioma and other CNS diseases. Two random forest based supervised tool, d-MeLB and p-MeLB, are proposed and evaluated for diagnosis and prognosis predictions. Further analysis on the clinically relevant epigenetic signatures was performed to explore the potential of methylome profiling from liquid biopsy or tissue in the management and treatment of meningioma patients.

While the study is clinically interesting and studies an important topic, I have several major concerns regarding the model evaluation and key feature analysis. Overall, the dataset is quite small and there was no external validation using a different study cohort. Especially given the small size of the test data (27 samples) and the confusion around some of the methodology, it's not clear that the predictive models could generalize reliably to new patients.

1. (Line 301) Authors proposed a diagnostic-MeLB model that distinguishes meningioma from other CNS diseases/tumors. The presentation of the model performance is confusing:

a. There are in total 208 liquid biopsy samples, of which 107 are assigned to discovery cohort (composed of 86 training samples and 21 model selection samples), 27 are assigned to an independent cohort (referred to as the test cohort). Why are the remaining samples (74) not included in the analysis?

b. (SI, Line 67) The authors mentioned that models are trained with 10-fold cross validation in the supplementary methods. Is the cross validation done on the training cohort? How is the model discovery cohort used in this procedure?

c. In the text authors provide accuracies for model selection cohort and test cohort separately, while Fig. 1E shows the combination of both (using different markers). It would be better to see these two cohorts in separate box plots so readers can better evaluate the generalizability of the model.

d. (Line 305) The authors mentioned that model has a 53% accuracy on recurrent meningioma samples. How is this number calculated? What (if any) are the negative samples being used? Why is this accuracy significantly lower than primary meningioma, is this expected? It'd be useful to see a more detailed discussion/explanation of this observation.

e. As shown in Fig.1B, genome-wide mean methylation level can already segregate meningioma patients from other non-meningioma conditions. what's the advantage of using the 130 DMPs and random forest model? Can you compare d-MeLB against a naïve baseline that utilize mean methylation level?

f. Following the previous point, is there any previous work doing the same/similar prediction using measurements of the same(or different) modalities? It would be helpful for evaluating the model performance in context.

g. The dataset size is quite small in terms of evaluation (21 + 27), have the authors tried looking for external datasets for further evaluations?

2. (Line 339) Authors then proposed a different model: prognostic-MeLB, which is designed to predict recurrence risk and is trained on tissue specimens. I have some concerns on the performance and applicability of the model:

a. (Line 172) Out of all the 66 tissue samples, 34 are assigned to the training cohort, 16 (11 positive and 5 negative) are assigned to the independent validation cohort, the remaining 16 (all positive) are assigned to another independent validation set. The naming of these cohorts are confusing and not consistent.

b. (Line 347) Authors stated that p-MeLB achieved 87.5% accuracy on the two independent validation sets above. Given the extreme class balance (27 positive, 5 negative). A simple classifier predicting all positives could have an accuracy of 84.3%. Similar to point 1-e and 1-f above, is there any other benchmarking methods that we can compare p-MeLB against?

c. (Line 372) Authors evaluated the p-MeLB on some external datasets in Fig. 3D and Fig. 3E. Results are only analyzed qualitatively, showing that p-MeLB scores correlate with our expectation of different patient groups. This is not sufficient for supporting the model's robustness and reliability. Could author generate a metric comparable to the accuracy used for independent validation sets?

3. (Line 386) Authors then applied the p-MeLB model (trained with tissue specimens) to serum specimens. If I understand correctly, the 66 serum samples of meningioma patients contain both primary and recurrent samples. The second yellow row of Fig. 4A (second row from top to bottom) indicates that all recurrent samples are in Q1-3 (lower recurrence scores), which contradicts my expectation as these are confirmed recurrent cases. Can authors clarify/explain this?

4. d-MeLB is trained with probes selected from the supervised comparison of meningiomas and non-meningiomas (SI, line 68). p-MeLB model is trained with probes identified through the k-clustering procedure of meningioma samples (SI, line 78). p-MeLB model's predictions on serum samples (high-risk and low-risk groups) picked another 70 DMPs through a supervised analysis (main text, line 403). The feature selection procedure in this work is very complicated, I am especially confused with the last step of using model prediction scores to separate patient groups, then performing supervised analysis to pick out key features. Shouldn't the random forest model be able to pick out key features without any human engineered selection/preprocessing? What's the reason behind each of the feature selection step?

5. (Line 410) Authors then used the k-cluster, recurrence risk groups (generated by p-MeLB) and tissue outcome groups (CR/CNR) to analyze immune composition differences. Given the intercorrelated relations between these groups: p-MeLB is trained on CR/CNR tasks, p-MeLB uses key features identified by k-cluster. I assume these groups to be highly correlated by design. The analysis should be simplified for better clarity.

6. Overall, the two models proposed in this work (d-MeLB and p-MeLB) serve very different purposes. A lot of the downstream analysis on key DMPs is done by supervised comparison of patient groups (meningioma/non-meningioma, CR/CNR) and k-clusters. I don't see why the two models are necessary for this work, and how (if any) they are connected. Can authors present a clearer motivation for developing these models?

7. Figures in the main text contain a lot of redundant information not discussed, or only briefly touched upon, in the text. Could authors simplify them so each panel contains only the key information?

NCOMMS-22-24666

Reviewer #1 (Comments to the authors)

The authors have conducted a study for which primarily they have produced cfDNA Methylation profiles from serum samples taken from patients with meningioma alongside various non-meningioma controls and relevant tissue samples. They create and validate a number of signatures/classifiers which variously distinguish between meningioma and non-meningioma, high-risk/low-risk, different subtypes. They postulate and provide some evidence for an underlying immune-biological difference between subtypes/risk groups. Overall I feel positive about this study and I think it makes a worthy addition to the literature.

Whilst the notion of detecting cancer specific methylation patterns in cfDNA itself is not especially unique Liu, J Mol Cancer 20, 36 (2021), Luo, B. BMC Med 20, 8 (2022), its fair to say that high-quality case studies are few and the field is relatively speaking in its infancy. This may doubly be said of CNS tumors for which the working assumption often seems to be that the CSF is most relevant as opposed to the plasma. Where this study goes a little further than most is in the detail with which the authors attempt to extract prognostic patterns. In summary, the concept and experimental design is not 100% unique but nevertheless a reasonable argument could be made for relevance of the approach as a proof of concept across a wider range of tumor contexts and most certainly with the meningioma field.

In general, the work supports the conclusions and claims with some caveats outlined below. Likewise, I don't believe there are any major flaws in data analysis or interpretation that should prohibit publication and I do think that the methodology is appropriate but I do have some caveats and comments which in my view should ideally be addressed prior to publication.

R1.1: It seems that a key criteria for how this approach is beneficial is the extent to which prognostic performance exceeds that available through existing or even non-biological means. I can see that this is somewhat addressed but I would prefer if this were more clearly evidenced, quantified and stated as an output of the paper. There is perhaps more of a focus on the extent the new risk signature tracks with the old.

Response: We appreciate and agree with the reviewer's remark. In the revised manuscript, we affirmed the advantage of our prognostic model over the available approaches more clearly and quantitatively. We updated classification agreements and correlations across ours and Nassiri classifiers (**Figure 2c**) and reported quantitative measurements derived from the comparison across ours and other prognostic classifiers or markers. For instance, compared to individual or composite benchmark markers, p-MeLB results (ACC: 0.877) rivaled or proceeded accuracies attributed to postoperatively tissue-based KI-67/MIB-1 labeling index (AUC: 0.820), transcriptomic-based markers (AUC: 0.81-0.866) or composite scores involving multiple risk factors (AUC: 0.849) integrated or not with imaging (AUC: 0.75-0.78) explored in prior publications (Liu et al., 2020, Liu et al., 2021; Morin et al. 2019; Zador et al., 2020; Zhu et al., 2020). In comparison of the performance between p-MeLB and Nassiri classifiers across an independent tissue cohort of primary meningiomas which possessed follow-up information (n= 47 samples), both classifiers presented identical specificity in detecting non-recurrent samples (n=6 samples); interestingly, we observed significant improvement in the ability to detect true recurrence within 5 years of follow-up (n=17 samples; $p < 0.01$) (**Figure 2c**). In addition, our classifier could reliably prognose across a larger timespan, for example, those tumors which recurred after 5-years (n=9 samples). We also highlighted the potential clinical relevance of our approach to promote an informed decision during the presurgical stage, based on recurrence risk classifications determined through a noninvasive blood-based liquid biopsy.

R1.2: If a measure of performance is the ability to distinguish between other non-meningioma types is the range of tumour types LGG, etc a little too limited to support the conclusions. Please consider what could be done or what caveats could be given that would address this?

Response: Thank you for the suggestion. To address this potential shortcoming, we profiled the methylome of an expanded internal independent cohort and retrieved the tissue methylome data from a comprehensive CNS tumor and nontumor cohorts provided by Capper et al., 2018 (e.g., embryonal, glioneuronal and mesenchymal tumors). We observed that d-MeLB signatures clustered meningioma liquid biopsy (serum and plasma) with tissue specimens and distinguished these tumors from other entities, supporting the generalization of our previous results across expanded CNS types besides LGG.

R1.3: The recurrent samples classify with an accuracy of ~53% is this expected? problematic? It seems not to be commented on again?

Response: Thank you for raising this concern. To address the limited applicability of our original classifier to recurrent meningiomas, we reconstructed the classifier by including recurrent collections in its construction, selection and validation. Through this approach, we were able to expand our original cohort (from n=44 to 63 collections), and the new d-meLB classifier presented 100% accuracy to classify an independent cohort of recurrent tumors (n=9) as meningiomas. This new analysis and results are reported in the revised manuscript.

R1.4: Saying that DMPs annotate in intergenic regions overlapping with enhancer regions and giving a % may not be saying much without referring to the location of non-significant CpGs as 450/850K arrays are non-randomly designed and cfDNA non-randomly fragments.

Response: Thank you for the opportunity to reevaluate and rectify our statements. In the revised version, we provide the genomic distribution of our DMPs relative to the observed genome-wide (Zhou et al., 2017). We also reframed our probe set descriptions to reflect the observed distributions of the non-significant CpGs across EPIC arrays. For example, we reframed the aforementioned description as: "These DMPs were primarily annotated in intergenic regions (overlapping enhancers: 18.4%; non-enhancers: 41.8%), compared to

frequencies observed across non-differential genome wide intergenic CpGs (overlapping enhancers: 15.1%; non-enhancers: 39.5%). This was also repeated for our prognostic DMP set description.

R1.5: Putative DMP-target gene pairs as an input into IPA or other gene ontologies is always a tricky prospect (no guarantee of a correlative/functional link) and I could just make the slightly non-scientific point that the heatmaps in 1D, 2C, 4C don't appear particularly convincing. Suggest either find a different way or strongly consider removing. I don't believe they add much to the story, the conclusions don't rely on them especially and they may be spurious.

Response: In agreement with the reviewer's remarks, we removed the alluded figures and reframed the analysis. Additionally, we reapproached our pathway analysis to solely include DMP-target gene pairs which possessed a potential functional interrelationship (negative correlation between methylation and expression) through the integration between tissue-derived DMPs and paired methylome and transcriptome datasets provided in Choudhury et al., 2022 (n=185 meningiomas). Briefly, we applied the p-MeLB model to classify the Choudhury cohort according to recurrence risk (high and low risk groups) and intersected these results with equivalent prognostic groups generated through their own classification, i.e. hypermitotic and merlin-intact groups¹⁶. We performed supervised analyses across the transcriptome of the resulting groups (high risk/hypermitotic and low risk/merlin intact) to identify differentially expressed genes (DEG) using the *DESeq2*, v. 1.36.0 package⁶⁹. Then, to infer a potential epigenetic regulation, we mapped the DEGs to CpGs located in their regulatory elements (promoters and enhancers) and selected those CpGs which simultaneously presented differential DNA methylation levels between the prognostic groups and negative correlation with the expression of their putative target genes (n=65 probe-gene pairs, **Figure 4d**). Next, we explored the alignment of these DMPs across the serum methylome to identify CpG which possessed identical DNA methylation levels compared to tissue while simultaneously distinguishing serum specimens by recurrence risk defined by p-MeLB (**Figure 4e**). In a complementary manner, serum DMPs derived from comparisons of high and low risk groups were mapped to tissue derived DEGs (n=25 PGPs, **Figure 4f**). These DMPs were analyzed for concordant DNA methylation across tissue groups (e.g., hypermethylation in high risk vs low risk serum groups and hypermethylation in high risk/hypermitotic vs low risk/merlin tissue groups), and inverse correlation with gene behavior (e.g. hypermethylation and downregulation of gene). Then, we conducted gene set enrichment analyses (GSEA) using the Ingenuity Pathway Analysis (IPA) platform, identifying relevant diseases and biological processes related to these recurrence risk-related DMP-gene pairs. We showed that many of these genes are potentially involved in tumor progression and growth, suggesting that these DMP-gene pairs could be mechanistically involved in the recurrence risk of meningiomas (e.g. cancer, cell proliferation and movement) (**Figure 4g**). Some may function as prognostic or actionable markers detectable in serum specimens. These new analyses and results have been incorporated into the revised manuscript.

R1.6: I would query if it's totally clear from the text if you consider deconvolution of the liquid biopsy profiles as indicative of the TME, it seems that sometimes you are indicating that. If so, that on the face of it seems quite hard to justify?

Response: Thank you for the opportunity to clarify and elaborate on this information. Considering the differential enrichments/depletions of specific immune cell proportions across the serum k-clusters and recurrence risk groups classified through p-MeLB, we hypothesized that the immune findings reflected either a systemic or possibly the spillover of the local (TME) immune response to the presence of meningiomas with different growth behaviors. Favoring the first hypothesis, we observed that the serum-related findings were not significantly correlated with the tissue cell proportion estimates in the majority of the cases (range: $\rho = 0.132-0.695$), insinuating they may not reflect TME (**Supplementary File 5**). When we analyze inter-group differences, e.g., between prognostic risk groups, across different sources, we observe that serum and tissue

profiles present distinct differential immune composition landscapes. For example, high risk serum groups display enrichment with B-Cell, Natural Killer cell type proportions while equitable groups in tissue display enrichment with monocytes proportions (**Figure 3b, Supplementary Figure S5c**).

Additionally, we also observed positive correlation between DNA methylation-based immune cell estimation, genome-wide (GW) DNA methylation patterns and concordance of diagnostic and prognostic classifications using paired serum and plasma samples (the latter less prone to genomic DNA contamination), providing some evidence that the observed immune composition landscape are *bonafide* findings in response to the presence of meningiomas and not due to a potential contamination with genomic DNA derived from lysed white blood cells (see below; **Supplementary Figure S2a**).

Correlation coefficients associated with continuous characteristics for matching LB serum & plasma collections.

Pair #	Sample IDs	Cohort	Correlation (ρ)		Concordance	
			GW Methylation	Immune Proportions	d-MeLB	p-MeLB
1	LB-ADD-03/04	Additional MNG	0.95**	0.7**	Yes	Yes
2	LB-ADD-05/06		0.93**	0.81**	Yes	Yes
3	LB-ADD-07/08		0.93**	0.98**	Yes	Yes
4	LB-ADD-09/10		0.89**	0.93**	Yes	No
5	LB-ADD-11/12		0.92**	1.00**	Yes	Yes
6	LB-ADD-19/20		0.96**	1.00**	No	No
7	LB-ADD-21/22		0.91**	1.00**	Yes	Yes
8	LB-ADD-23/24		0.95**	0.95**	Yes	Yes
9	LB-ADD-25/26		0.94**	0.87**	Yes	Yes
10	LB-ADD-27/28		0.91**	0.97**	No	No

R1.7: You say concordant but non-significant. Seems a long winded and possibly a bit misleading way to say there isn't a relationship as far as we can show.

Response: We thank the reviewer for the suggestion. We have reoriented our analysis in a way that focuses solely on the positive and significant correlative relationships observed between simultaneous serum and plasma, or serum and tissue collections (**Supplementary Figure S2a**).

R1.8: The purpose and scope of the paper would be far better justified if you could outline how you might expect this to be used in clinical practice? When would LB take place how would treatment be altered, in what sense is this better or more practicable than simply dealing with bulk tissues samples.

Response: We thank the reviewer for the opportunity to highlight the clinical relevance of our findings. In the revised manuscript, we outline this information and provide an infographic description of how and when our approach could be used to manage patients with meningiomas (**Figure 3d**).

R1.9: To me the figures are very cramped, often the text is too small to see without zooming in and out and it is just generally very busy. I think that the balance between what is in the main and what is in the supps isn't right. Additionally, the colour scheme is quite loud and intense and divergent and it gives a slightly stroboscopic and busy effect. Normally I wouldn't complain but I think in this case I do think it inhibits the clarity of your message and makes it a harder read than it needs to be. The figures could use a bit of a rejig and some serious streamlining in my opinion and then by definition the results section text will be clearer too. Please consider carefully whether each track on the heat map or deconvoluted cell type etc is really necessary for the main message.

Response: We thank the reviewer for the suggestions. We reworked the layout of the figures and panels to address this suggestion.

R1.10: R1.10a -The signature/classifier generation is complex to follow. I think I just about could recreate it but I do wonder if some posted code, explanation on GitHub or some fuller description might help. R1.10b - One issue in particular is important for me to understand. It may be there and I can't see it but the feature selection when dealing with the training set, are features reselected fresh each time during each iteration of any cross-validation. R1.10c - This seems an important issue for the generalization of the signature (although I note the independent validation, but also note sample size is relatively small and may not be reflective of the general population). Could this be clarified/justified?

Response: Thank you for the opportunity to clarify these concerns. R1.10a - We attempted to simplify the description of the classifier generation in the text and clarified details in the workflow schematic (**Supplementary Figure S3b**). Additionally, we have made our developed classifiers available within our data deposition provided to reviewers (Google Drive), along with necessary syntax, so that they may be applied and tested. **R1.10b** - To avoid biasing the signatures, our sets were only introduced to training sets during each iteration, and not to model selection or independent validation samples. **R1.10c** - Corroborating the generalization of our signature, we show the ability of the meningioma parent signature (similar methylated probes between meningioma serum and tissue [SMPs]: n=18,167 CpGs) to separate meningioma (liquid biopsy and tissue sources) from 2,000+ novel tissue samples, as depicted in the dimension-reduction t-SNE visualization (**Figure 1e**) and from an expanded independent validation encapsulating recurrent glioma (n=55) and additional meningioma collections (serum: n=18; plasma: n=10; tumor tissue: n=32)(**Figure 1f-g**).

Other comments:

R1.11 - There seems to be results in the supplementary text which is a bit unorthodox and really should be moved to the main if it is still deemed important.

Response: We thank the reviewer for the comment. Although unsure as to which specific results the reviewer alluded to, we have moved information we deemed important to understand key messages from the supplementary to the main text.

R1.12 -S1 and S4 seem to be introduced out of order?

Response: Thank you for pointing this out. We made sure to cite the figures in the order of their appearance in the text.

Reviewer #2 (Remarks to the Author): expertise in cfDNA methylation profiling

The authors describe an analysis of biomarkers related to detection of disease and prognostic outcomes using DNA methylation.

The manuscript is very nicely presented, but I do have two significant concerns relating to the use of serum for their liquid biopsy profiling; and the lack of secondary wetlab validation by bisulfite resequencing of their CpG biomarkers, as detailed below

R2.1) The use of serum

First, the authors use of serum for their liquid biopsy profiling methods is somewhat problematic. For example, in somatic mutation profiling of cfDNA isolated from cancer patients the use of Streck, Paxgene ccfDNA, and other tubes is exclusively used because they both prevent clotting and stabilize whole blood samples from lysis; conversely, the use of serum is avoided in cfDNA testing because serum samples result in high levels of buffycoat DNA due to lysis of the PMBCs which contributes significant levels of nontumour DNA to the extracted cfDNA, thereby reducing the amount of actual tumour material in the isolated DNA and the ablating the sensitivity of any genomic profiling methods.

This probably explains the reason for the enrichment in genes associated with Immune signalling in the authors analysis (line 299); as well as the estimation of the percentage of immune cells in the methylation data (line 318, supp figure 3b) which shows a ~50% neutrophil percentage - which is the exact proportion typically present in a normal sample of white blood cells; and again significant enrichment of sites related to immune function for their risk prediction (line 408). Indeed, each time mention of genes and pathways identified as differentially methylated for their biomarkers are mentioned, there is recurrent representation of immune system related genes and pathways.

Of course there is certainly the possibility that the signal represents a bone fide biomarker, however the difficulty is the fact that they are getting enrichment for CpGs related to immune genes in a sample from the blood, for which there would certainly be significant buffycoat gDNA contamination due to the use of serum.

Response: We appreciate the opportunity to address these concerns. We agree that, to some extent, lysed peripheral blood mononuclear cells (PMBC) DNA may be contributing to our immune findings. However, we provide evidence that the cell proportion alterations detected in the serum specimens may reflect a bonafide inflammatory or immune response to the presence of meningiomas such as 1) differential enrichment/depletion serum immune cell proportion across k-clusters and recurrence risk groups classified through p-MeLB (**Supplementary Figure S5a, Figure 3b**, respectively) and 2) positive correlation and concordance of our paired serum- and plasma-derived results (**Supplementary Figure S2a**). As a robust and validated biomarker, we are confident that the identified signature (whether immune or tumor) correctly diagnoses, prognosticates, and provides evidence for recurrence using serum, all of which was done non-invasively.

Additionally, we also observed positive correlation between DNA methylation-based immune cell estimation, genome-wide (GW) DNA methylation patterns and concordance of diagnostic and prognostic classifications using paired serum and plasma samples (the latter less prone to genomic DNA contamination), providing some evidence that the observed immune composition landscape are *bonafide* findings in response to the presence of meningiomas and not due to a potential contamination with genomic DNA derived from lysed white blood

cells (see below; **Supplementary Figure S2a**).

Correlation coefficients associated with continuous characteristics for matching LB serum & plasma collections.

Pair #	Sample IDs	Cohort	Correlation (ρ)		Concordance	
			GW Methylation	Immune Proportions	d-MeLB	p-MeLB
1	LB-ADD-03/04	Additional MNG	0.95**	0.7**	Yes	Yes
2	LB-ADD-05/06		0.93**	0.81**	Yes	Yes
3	LB-ADD-07/08		0.93**	0.98**	Yes	Yes
4	LB-ADD-09/10		0.89**	0.93**	Yes	No
5	LB-ADD-11/12		0.92**	1.00**	Yes	Yes
6	LB-ADD-19/20		0.96**	1.00**	No	No
7	LB-ADD-21/22		0.91**	1.00**	Yes	Yes
8	LB-ADD-23/24		0.95**	0.95**	Yes	Yes
9	LB-ADD-25/26		0.94**	0.87**	Yes	Yes
10	LB-ADD-27/28		0.91**	0.97**	No	No

R2.2) The lack of secondary wet lab validation of their markers

R2.2a - Next, while the authors have applied adjusted p-values to their filtering, casting an eye over the list of differentially methylated cytosines I note that the mean difference between their cohorts hovers ~between 12-20%, which is near the threshold of noise on the beadchip arrays in my experience. In standard beadchip analysis a minimum mean difference of 20% is usually applied because of this noise. Moreover, I know from first hand experience that biological outliers in data easily skew means and adjusted p-values and make them look significant.

Response: We thank the reviewer for these relevant remarks and for the opportunity to elaborate on these concerns. We expanded the 'diff.mean' stringency (Δ DNA methylation) between sample groups to over 15% which is within the variable range reported in the literature (i.e. 10-20%) (Schenkel et al., 2022; Liu et al., 2019; Meyer et al., 2021; Mansell et al. 2019) and selected DMPS with the highest significance through FDR adjustment. Additionally, to avoid selection of spurious signatures, we filtered serum signatures which presented DNA methylation level similarities with meningioma tissue and presented the ability to distinguish meningiomas from other tumors across both independent and external cohorts (Capper et al., 2018) (**Figure 1e**), prior to formulation of our diagnostic machine learning classifier (d-MeLB).

R2.2b-Moreover, within this analysis I also have concerns about multiple t-testing, despite the use of FDR correction. In this case the use of EPIC array constitutes ~850k cytosines but the authors used supervised analysis to identify only 130 differentially methylated probes – a small number which could be attributed to random chance due to high sampling, despite the prima facie low p-values.

Response: We implemented multiple stepwise supervised analyses to eliminate the potentiality for random or background methylation noise, thereby ensuring the meningioma-specificity of selected signatures. Despite the final number of our discriminatory signatures being lesser than those reported in related studies ranging between 283-9529 CpGs (Olar et al, 2015, Bayley et al., 2022, Nassiri et al., 2019, Capper et al., 2018), the validation and generalization of our findings in independent and external cohorts suggest these signatures are non-stochastic (**Figures 1f-g, 2b**).

R2.2c-Indeed, if I understand their methods correctly, the initial identification of CpGs employed the entire cohort; only after specific CpGs were identified was the dataset split into separate training and validation cohorts to build their classifier. While I understand the motivation of performing discovery

first then splitting the cohort for ML training, analysis done in this way significantly biases the validation dataset towards the appearance of significance.

Response: Thank you for the opportunity to clarify our strategy and include this information in our revised manuscript. Definition of signature sets was conducted using only training set samples across each iteration within classifier formulation (both d-MeLB and p-MeLB), in order to eliminate any potential biasing towards significance in the model selection (where implemented) or independent validation cohorts (not used in signature generation) (**Supplementary Figures S3b, S4a**).

Specific reviewers points to be addressed

R2.2d-As such, the only way to effectively address this is through secondary wet lab validation, which is the next major weakness of the current manuscript. While the authors have separated their data when generating their classifier, given both the use of serum and the shallow difference in mean methylation, secondary validation by bisulfite resequencing of their diagnostic and prognostic CpGs needs to be performed (more on this below in my specific points to address).

R2.2e-While I do have significant technical concerns about the use of serum, there is certainly the potential that despite the fact their serum samples were almost certainly contaminated with buffycoat DNA, there could be an underlying diagnostic/predictive signature. However, to be accepted for publication significant validation work needs to be undertaken.

First, to accept the manuscript, I believe technical validation using bisulfite resequencing must be performed on all serum samples originally used (for which there is sufficient material left), to verify the signal they are getting is not attributable to stochastic variation on the array platform. A good resequencing protocol only takes minimal sample input, and I have successfully run effective resequencing applications with 10ng of input DNA, although more is certainly better.

R2.2f-Second, additional bisulfite resequencing needs to also be performed on an *independent* set of serum samples not originally used for discovery and training. In the absence of both sets of validation data I have significant concerns about their DMPs.

Response: Thank you for the suggestions. In recent years, we and others have shown the feasibility, accuracy and generalization of DNA methylation array to detect specimen-specific circulating cell-free DNA released in several types of biofluids (CSF, blood, urine, aqueous humor) (Zhai et al. 2012; Widschwendter et al., 2017; Sabedot et al. 2021; Herrgott et al. 2022) and distinguish across and within CNS and non-CNS tumor types

As a reasonable request, we conducted whole genome sequencing in a subset of leftover and new serum samples to perform the requested technical validation. However, we argue that a technical validation would not be necessary to assure the applicability or generalization of our diagnostic and prognostic signatures in the context of the approach we undertook to develop our machine learning-based models, as follows: 1) to assure our selected signatures were meningioma-specific and reduce the detection of background noise, we applied several layers of contingencies including the refinement of our signatures addressing one of this Reviewer's comments. 2) our machine model was trained using EPIC array DNA methylation values; therefore, technical validation could show qualitative trends; however, the use of DNA methylation levels determined by any other technique other than EPIC/450K arrays would not be suitable as input to our model.

Furthermore, in this revised manuscript, the accuracy of our DNA methylation-based models to classify the samples according to memberships was validated in several independent cohorts, including the new samples,

Additionally, addressing the reviewer's concerns, we performed methylPCR, as previously described (Campan et al. 2018), and WGBS, as secondary benchmark methods to validate the DNA methylation patterns observed across diagnostic and prognostic sets detected through EPIC array. To perform this validation, we used cfDNA leftovers from the cohort originally reported in the manuscript as well as a new meningioma cohort set of paired serum/plasma and tissue samples (serum: n=8; plasma: n=5; and tissue: n=12).

For the MethylPCR assay, we selected 9 differential CpGs representing an overlap between diagnostic- and prognostic-oriented supervised analyses, previously profiled through EPIC array (**Supplementary Figure S2b; Supplementary Figures S3a, S5g**). Designed primers are described in (**Supplementary File 5**).

Across samples with low cfDNA concentration, the controls (HB-313) presented high PMR values. To avoid potentially skewed data at the assayed sites, we masked data for controls with PMR values >21 with "N/A"

Correlation statistics for CpG methylation across EPIC & MethylPCR arrays.

CpG	LB: Serum (n=8)	LB: Plasma (n=5)	Tumor Tissue (n=12)
	ρ	ρ	ρ
cg01181584	-0.48	0.97	0.37
cg01465838	0.6	0.74	0.31
cg03845383	0.38	0.89	0.02
cg04208666	0.65	-0.33	0.01
cg08402107	0.4	0.97	-0.03
cg12942155	0.03	-0.03	0.29
cg21245981	-0.03	-0.88	0.03
cg21699881	0.7	0.22	0.79
cg25112312	0.21	0.72	0.81

(**Supplementary File 5**). Following masking procedures, we observed positive correlations between overall CpG methylation levels observed across EPIC (13-value) and methylPCR (PMR value) arrays for a few markers across all matching sources (e.g. cg01465838 and cg21699881) (see below). Similarly, we observed concordant DNA methylation differences between recurrence risk groups (i.e. hypermethylated in high risk compared to low risk) across both arrays in the serum for a majority of the isolated markers (n=5/9 CpGs) (**Figure 2c**).

Circulating free DNA profiling through WGBS, across 110 samples composed of MNG predicted to have high and low risk to recur, displayed on average 22,647,947 CpGs and 2.56X depth coverage, prior to any filtering based on coverage. Correlations across unfiltered CpG sets between WGBS and unmasked paired EPIC arrays ranged between $p=0.816$ and 0.863 ($p<2.2e-16$) (**Supplementary Figure S2b**). Following all quality control, WGBS captured an average of 9,256,306 CpGs with total coverage >3X depth across the 10 cfDNA samples. An average of 239,333 CpG sites were shared between EPIC array and WGBS, with genome-wide DNA methylation correlations ranging between $p=0.841$ and 0.896 ($p<2.2e-16$) (**Supplementary Figure S2b**). Additionally, the DNA methylation levels of CpGs present among p-MeLB signatures (n=13) and supervised analysis between CR and NCR comparisons, determined by EPIC arrays (13-values) were highly correlated with DNA methylation percentage values obtained through WGBS ($p=0.73$, $p=3.1e-16$ and $p=0.68$, $p=2.2e-16$, respectively) across 10 matching cfDNA samples (**Supplementary Figure S2c-d**). Altogether, our EPIC-based results in this set were corroborated by a secondary benchmark method such as WGBS.

R2.2g - Building on this, the authors must incorporate a significant section in their results and discussion on why they elected to use serum;

Response: Thank you for the opportunity to address these questions in the manuscript. The DNA methylation profile of serum has been previously reported as a reliable approach for the diagnosis of other CNS and non CNS tumors (Zhai et al. 2012; Widschwendter et al., 2017; Sabedot et al. 2021; Herrgott et al. 2022). In contrast to the detection of mutational burden which can be affected by dilution and contamination with cfDNA from lysed white blood cells (buffy coat), DNA methylation profiling is more resistant to this kind of interference due to the preservation of cell-type specific DNA methylation patterns across different sources and the sensitivity of DNA methylation array platform (EPIC) or whole-genome sequencing to detect minute amounts of intact or fragmented cfDNA as encountered in liquid biopsy specimens (e.g. <1ng)(Pittella-Silva et al. 2020; Gerber et al. 2020; Herrgott et al. 2022; Sabedot et al. 2021; Constâncio et al. 2019; Moss et al. 2018; Kang et al. 2017, Li et al. 2022; Gao et al. 2022). Herein, we used serum as it was the sole blood component available in our tumor bank at the period of our data freeze. To our knowledge, there are no formal comparisons between DNA methylation results using serum or plasma reported in the literature. Therefore, to properly address this reviewer's comment, we profiled the methylome of a new meningioma cohort set encompassing paired serum, plasma and tissue samples. We showed concordant results regarding diagnostic and prognostic classifications through our classifiers across the three specimen sources as well as a positive relationship between genome-wide DNA methylation patterns and the estimation of immune cell proportions through a DNA methylation-based deconvolution method (**Supplementary Figure S2a**). Altogether, corroborating previous reports, our findings suggest that DNA methylation patterns are sensitive to detect tumor-specific independently of the source it is extracted from, i.e., serum, plasma or tissue. The stability, reproducibility, and robustness of detecting cell-specific DNA methylation patterns through EPIC arrays across FFPE and fresh frozen tumor specimens has also been previously reported (Mosella et al., 2021; Herrgott et al. 2022; Sabedot et al. 2021; Capper et al., 2018; Constâncio et al. 2019; Moss et al. 2018; Kang et al. 2017, Li et al. 2022; Gao et al. 2022). We added these results and information in the revised manuscript.

R2.2h - and why despite the (most certain) buffycoat contamination in the isolated DNA they believe they are able to derive a tumour signal.

Response: We thank the reviewer for the opportunity to explain this more clearly. Although we do not provide a direct measurement of the circulating tumor DNA, the ability of our cfDNA-derived signatures to discriminate meningiomas from other tumor types and against nontumor controls suggests we are detecting *bonafide* meningioma-specific signals. Additionally, 1) DNA methylation signatures detected in serum clustered liquid biopsy samples with meningioma tissue and not with other CNS tumor types as depicted in **Figure 1e** and DNA methylation signatures which are common across serum and tissue accurately classify independent and external serum or tissue cohorts as meningiomas (**Figure 1f-g**).

R2.2i Specific points which need to be touched on are their expected levels of tumour DNA in their isolated liquid biopsy sample; and the overall sensitivity, accuracy, and precision of the beadchip platform to resolve those differences. If they are able to empirically determine the amount of tumour DNA in their sample using, for example ddPCR against known somatic mutations, that would also be of significant benefit.

Response: Thank you for the comment. As mentioned by this reviewer and stated in the literature, the potential contamination with buffy coat DNA coupled with the low mutational burden associated with meningiomas, decreases the sensitivity to detect mutational burden using serum specimens (Gerber et al. 2020; Pittella-Silva et al., 2020). However, although we have not directly measured the amount of tumor DNA in our study, due to resource limitation, the accuracy of our identified meningioma-specificity signatures to distinguish between meningioma and non-meningioma groups suggest that we had sufficient amounts of meningioma-related DNA and as suggested above, a systemic immune signature or non-tumor signature associated with meningioma.

R2.3 - Regarding the ML random forest classifier, the authors must export the classifier and supply it as a supplementary file with the manuscript; together with detailed instructions on how researchers can import and use the classifier with their own data (in either R or python). I know this can easily be done in python in sklearn; if the authors have used R or another package they will have to investigate different strategies for exporting and importing their RF model.

I am requesting this as I do not believe it would be easy to replicate the authors RF classifier based on their methods description starting from the raw data, and for the purposes of reproducibility and enabling other researchers to validate their results, this is an absolute necessity.

Response: Thank you for the opportunity to clarify our model development and analytical processes. Codes and complementary instructions on how to process our generated methylome data and apply the detailed classifiers are located within the data deposition provided to the reviewers (Google Drive). The construction of a user-friendly platform containing the classifiers available to researchers is underway and is expected to be fully operational in 2023 for research purposes. Clinicopathological and serum- or tissue-derived methylome data can be uploaded and classified through our diagnostic and prognostic classifiers through this webtool, similar to the available web tool designed by Capper et al. 2018 using tissue-derived data. The generated data will allow refinement of our models before available for clinical use.

Reviewer #3 (Remarks to the Author): expert in meningioma

The authors carried out methylation profiling on liquid biopsies from 208 patients with meningiomas (44 primary, 19 recurrent MNG) or other CNS and non-neoplastic diseases (145 non-MNG samples) to determine whether they are useful as a non-invasive method of diagnosing patients and determining their likelihood of meningioma recurrence. Using a combination of Illumina 450k and 850k arrays, they identified 130 significant meningioma-specific differentially methylated probes in genes in pathways involved in retinoid X receptor activation and the immune response that could be used to distinguish meningioma patients from glioma, pituitary tumors, other CNS tumors and non-neoplastic diseases, using a non-invasive assay.

Comments to the authors

R3.1 - I am not an expert on statistical methodology, but the data look impressive to me. I just have a couple of questions about the data analysis:

p6, line 119, The methods section states that only probes in common between Illumina 450k and 850k were used for downstream analysis, but I couldn't find the number in the text. How many probes were assessed altogether?

Response: Thank you for the comment. The number of probes common between the two arrays (EPIC & 450k) was approximately 339k CpGs and we state in the proper section in the revised manuscript.

R3.2 - p14 line 303-4, for the d-MeLB prediction model in Serum cfDNA methylation samples, I believe it is very unusual to obtain an AUC of exactly 1.0. Was this really the case?

Response: We appreciate the comment. We confirmed the value of the original reported AUC by conducting reformulation of the model, the resulting AUC across the model selection remained 1.0 as described in **Supplementary Figure S2b**.

Minor comments

R3.3 - Gene names should be italicized.

Response: We appreciate the reviewer's thoroughness and have italicized the gene names as requested.

R3.4 -I'm not sure if it was part of the anonymization process, but the Mendeley Data DOI needs to be added.

Response: Yes, the reviewer's assumption about the anonymization process is correct. The data has been deposited and will be released upon acceptance of the manuscript, alongside the formulated classifiers, and we stated this information in the revised text. In the meantime, we have provided the reviewers with a google drive link with all the necessary materials for reproduction of results, including source data, raw DNA methylation IDATs, and generated classifiers.

R3.5 - Check for typos e.g. p9 line 208 replace "biological" with "biologically"; p10 line 210 insert "relevance" after "biological".

Response: We appreciate the reviewer's thoroughness and have corrected the typos as requested. Additionally, we have submitted the revised version of the manuscript to our scientific editor.

Reviewer #4 (Remarks to the Author): expert in machine learning

In this work, the authors explored the epigenetic signatures (DNA methylation) of patients with meningioma and other CNS diseases. Two random forest based supervised tool, d-MeLB and p-MeLB, are proposed and evaluated for diagnosis and prognosis predictions. Further analysis on the clinically relevant epigenetic signatures was performed to explore the potential of methylome profiling from liquid biopsy or tissue in the management and treatment of meningioma patients.

While the study is clinically interesting and studies an important topic, I have several major concerns regarding the model evaluation and key feature analysis. Overall, the dataset is quite small and **there was no external validation using a different study cohort**. Especially given the small size of the test data (27 samples) and the confusion around some of the methodology, it's not clear that the predictive models could generalize reliably to new patients.

Response: Thank you for the suggestion. Addressing this point, please see below response **R4.1g**.

R4.1. (Line 301) Authors proposed a diagnostic-MeLB model that distinguishes meningioma from other CNS diseases/tumors. The presentation of the model performance is confusing:

R4.1a. There are in total 208 liquid biopsy samples, of which 107 are assigned to the discovery cohort (composed of 86 training samples and 21 model selection samples), 27 are assigned to an independent cohort (referred to as the test cohort). Why are the remaining samples (74) not included in the analysis?

Response: We thank the reviewer for the opportunity to clarify this concern. We originally excluded 74 recurrent samples (gliomas, n=55; meningiomas, n=19) to avoid potential genome-wide DNA methylation alterations due to treatment. However, upon reconsideration spurred from the reviewer's inquiry, we adjusted our classification goals and included recurrent meningiomas in classifier construction, selection and validation. Additionally, to satisfy any concerns, recurrent glioma samples were added to independent validation (**Figure 1f**).

R4.1b. (SI, Line 67) The authors mentioned that models are trained with 10-fold cross validation in the supplementary methods. Is the cross validation done on the training cohort? How is the model discovery cohort used in this procedure?

Response: We appreciate the opportunity to further clarify this methodology. Cross validation was conducted using traditional methods across the training sample set, independent of any model selection or validation set samples, to allow the machine to predict how the classifier would react to unknown data (i.e., independent data). The text has been appropriately altered to indicate this process.

R4.1c. In the text authors provide accuracies for model selection cohort and test cohort separately, while Fig. 1E shows the combination of both (using different markers). It would be better to see these two cohorts in separate box plots so readers can better evaluate the generalizability of the model.

Response: We agree with this remark. We have implemented this reviewer's recommendation and have removed model selection samples from the classifier evaluation (**Figure 1f**).

R4.1d. (Line 305) The authors mentioned that the model has a 53% accuracy on recurrent meningioma samples. How is this number calculated? What (if any) are the negative samples being used? Why is this accuracy significantly lower than primary meningioma, is this expected? It'd be useful to see a more detailed discussion/explanation of this observation.

Response: Accuracy was calculated through traditional methods, i.e. taking the sum of true positives and negatives over total sample size. To address the limited applicability of the original model to recurrent meningioma specimens using serum, we expanded the meningioma cohorts used to build the models (from 44 to 63 specimens) and readdressed our machine learning approach by including the recurrent collections in classifier construction, selection and validation as explained in response to **R4.1a**. Our readjusted d-meLB classified independent recurrent samples (n=9) as meningiomas with a 100% accuracy. We have updated this section of the text accordingly.

R4.1e. As shown in Fig.1B, genome-wide mean methylation level can already segregate meningioma patients from other non-meningioma conditions. What's the advantage of using the 130 DMPs and random forest model? Can you compare d-MeLB against a naïve baseline that utilizes mean methylation level?

Response: We thank the reviewer for the opportunity to justify our chosen methodologies. As depicted in **Figure 1b**, mean DNA methylation is significantly differential across meningiomas and a majority of their included counterparts; however, its variability hinders the accuracy and generalization to other serum or tissue cohorts. By constructing a random forest model using meningioma-specific CpGs as input we resolved these limitations as displayed in **Figure 1f-g**. However, to address the reviewer's suggestion, we formulated univariate and multivariate linear regression techniques using calculated mean DNA methylation data points across whole genome and meningioma specific CpG sets as input (n= -573k and n=98 CpGs, respectively). Overall, the resulting algorithms showed lower performance measures compared to our RF-based d-MeLB methodology when applied to an additional independent cohort (see below, **Supplementary Figure S3c**). For example, genome-wide univariate analyses were able to classify independent meningioma and non-meningioma with an accuracy of ~89%, proximate to overall accuracy observed in d-MeLB. However, sensitivity fell to 50% when the model was applied to an additional liquid biopsy meningioma cohort.

We also compared performance parameters of the d-MeLB approach to other more "traditional" machine learning methods using varying inputs (e.g., Dimension Reduction-DR and linear-discriminant analysis-LDA derived CpGs). For the DR approach, we used signatures derived from mean DNA methylation similarities between meningioma tissue and serum (n= -18k CpGs) to explore whether this method could accurately diagnose meningioma. For the LDA approach, we used signature sets (supervised-derived DMPs) identified in an identical manner to the implemented random forest algorithm, namely d-MeLB. Ultimately, as detailed in the table below, the performance of these two different approaches were appreciably worse than d-MeLB measures (see below), solidifying the application of our methodology over more naive approaches.

Comparisons of diagnostic Meningioma Liquid Biopsy (d-MeLB) performance to other classifier approaches

	Machine Learning Approaches			Univariate Analysis		Multivariate Analysis
	d-MeLB [RF]	DR [RF]	LDA	Genome-Wide	Supervised CpGs	Genome Wide & Supervised CpGs
Signature Set (n):	20 CpGs	~18,000 CpGs	21 CpGs	~573,000 CpGs	98 DMPs	~573,000 CpGs & 98 DMPs
Model Selection (n=27)						
AUC:	1.00	0.893	0.857	N/A	N/A	N/A
Independent Validation: Serum (n=107)						
ACC (%):	90.7	75.7*	73.8*	31.8*	86.9	83.2
SE (%):	87.5	40.6*	34.4*	100	75.0	65.6*
SP (%):	92.0	90.7	90.7	2.7*	92.0	90.7
CUI (+):	0.721	0.264	0.210	0.305	0.60	0.492
CUI (-):	0.870	0.709	0.693	0.027	0.824	0.78
Additional MNG Accuracies (%):						
Plasma (n=10):	90.0	10.0*	40.0	100	100	80.0
Tissue (n=32):	68.8	34.4*	31.3*	50.0	6.25*	43.8

*Notes: All statistical comparisons are between dMeLB and the annotated classifier. RF: Random Forest; DR: Dimension reduction; CUI (+): clinical utility index for positive samples; CUI (-): clinical utility index for negative samples; LDA: linear discriminant analyses; DMPs: differentially methylated probes; *: comparison to d-MeLB statistics is statistically significant at $p \leq 0.05$.*

R4.1f. Following the previous point, is there any previous work doing the same/similar prediction using measurements of the same(or different) modalities? It would be helpful for evaluating the model performance in context.

Response: We appreciate the suggestion. In response to the reviewer's questions, noninvasive machine learning models were developed using methylome profiling of liquid biopsy samples for the diagnosis of CNS (pituitary tumors, gliomas) and non-CNS entities (Shen et al., 2019; Herrgott et al, 2022, Sabedot et al., 2021) or imaging (radiomics) to differentiate meningiomas from mimicking diseases (Dong et al., 2020). Three other DNA methylation-based prognostic classifiers have been reported using meningioma tissue samples (Nassiri et al., 2019; Bayley et al., 2022; Choudhury et al., 2022). However, this is the first work using machine learning

models to diagnose and prognosticate meningiomas using liquid biopsy or tissue samples. In the revised manuscript we discuss the performance of our models (d- and p-MeLB) in comparison to those and to other benchmark markers (e.g., Ki-67, transcriptome etc.).

R4.1g. The dataset size is quite small in terms of evaluation (21 + 27), have the authors tried looking for external datasets for further evaluations?

Response: We performed a comprehensive search in the literature and have not found an external set of DNA methylation profiling in liquid biopsy specimens from meningiomas to include in our study. To address this limitation, we generated an additional cohort of liquid biopsy meningioma specimens (serum: n=18), as well as independent recurrent glioma (not included in classifier formulation; n=55). With inclusion of recurrent meningioma in the model formulation/validation, the size of our model selection increased to 28 samples. Future plans to continuously validate our classifiers with additional samples are in place.

R4.2a. (Line 339) Authors then proposed a different model: prognostic-MeLB, which is designed to predict recurrence risk and is trained on tissue specimens. I have some concerns on the performance and applicability of the model: (Line 172) Out of all the 66 tissue samples, 34 are assigned to the training cohort, 16 (11 positive and 5 negative) are assigned to the independent validation cohort, the remaining 16 (all positive) are assigned to another independent validation set. The naming of these cohorts are confusing and not consistent.

Response: Thank you for pointing this out. We solidified the nomenclature of the cohorts to clarify their distinctions and have addressed the workflow associated with p-MeLB formulation (**Supplementary Figure S4a**). Briefly, for the model development, machine-driven randomization assigned the total tissue cohort into training (CR, n=24; CNR, n=10) and independent validation sets (CR, n=11; CNR, n=5). To further validate our model we also added an additional cohort with follow-up data encompassing 29 tissue samples from Institution #1 and 20 tissue methylome data from Bayley et al., 2022. Additionally, we were able to incorporate 8 collections of independent serum with confirmed follow-up to our independent validation (**Figure 2b**).

R4.2.b. (Line 347) Authors stated that p-MeLB achieved 87.5% accuracy on the two independent validation sets above. Given the extreme class balance (27 positive, 5 negative). A simple classifier predicting all positives could have an accuracy of 84.3%. Similar to point 1-e and 1-f above, are there any other benchmarking methods that we can compare p-MeLB against?

Response: Thank you for the suggestion. We located relevant publications which employed individual or composite benchmark features and relevant meningioma-related gene markers and compared their overall performance with p-MeLB to prognosticate meningioma. In summary, p-MeLB results (ACC: 0.877) rivaled or outperformed the accuracies attributed to Ki-67/MIB-1 proliferation index (AUC: 0.82), prognostic gene markers (AUC: 0.81), composite scores of multiple risk factors (AUC: 0.849) and radiomic (AUC: 0.75-0.78) as explored in prior publications (Liu et al. 2020; Liu et al. 2021; Zador et al. 2020; Zhu et al. 2020; Morin et al. 2019) and other tissue DNA methylation-based models (Nassiri et al., 2019; Choudhury et al., 2022; Bayley et al., 2022). The results of these comparisons are stated in the revised manuscript.

R4.2.c. (Line 372) Authors evaluated the p-MeLB on some external datasets in Fig. 3D and Fig. 3E. Results are only analyzed qualitatively, showing that p-MeLB scores correlate with our expectation of different patient groups. This is not sufficient for supporting the model's robustness and reliability. Could the author generate a metric comparable to the accuracy used for independent validation sets?

Response: We appreciate the reviewer's suggestion. The main limiting factor for the quantitative comparison of p-MeLB against the included external datasets was the unavailability of follow-up data regarding recurrence to evaluate our recurrence risk predictions. To address this limitation and incorporate the reviewer's suggestion, we contacted the corresponding authors of one of the external sample cohorts who kindly provided us with updated follow up information from their patient cohort (Bayley et al., n=20). Additionally, we also generated an independent internal cohort (namely "additional" cohort) with confirmed follow up (tissue: n=29; serum: n=8). We verified the accuracy of our p-MeLB model to predict recurrence across these cohorts (**Figure 2b**). Furthermore, we provided quantitative measurements demonstrating improved performance of our approach to assess recurrence risk compared to current tissue-based classifier (Nassiri et al., 2019) as well as to standard benchmark methods described in response **R4.2b** (e.g., Ki-67 proliferation index, known genetic biomarkers, progression-relevant covariates).

R4.3. (Line 386) Authors then applied the p-MeLB model (trained with tissue specimens) to serum specimens. If I understand correctly, the 66 serum samples of meningioma patients contain both primary and recurrent samples. The second yellow row of Fig. 4A (second row from top to bottom) indicates that all recurrent samples are in Q1-3 (lower recurrence scores), which contradicts my expectation as these are confirmed recurrent cases. Can authors clarify/explain this?

Response: Thank you for the opportunity to clarify. Considering the misleading representation of recurrence risk score through the quantile heatmap (originally Fig. 4A), we removed the original heatmap from the manuscript and redesigned the figure to portray our message more precisely. To better clarify the distribution of clinicopathological and molecular features across recurrence risk groups, we created a proportion-focused table which considered the distribution of certain features across high & low risk samples, with a baseline reference composed of whole cohort averages for each feature (**Figure 3a-b**). For example, when examining time of sample collection distributions, we see a higher proportion of primary/recurrent samples in the high-risk group; however, when compared to equivalent proportions in low-risk samples, distributions were not significantly differential. This redesigned representation approach highlights the accuracy of p-MeLB to differentiate samples by their risk to recur through the differential proportion of confirmed recurrence to confirmed non-recurrence samples observed during surveillance (not at the time of collection). Associated forest plots showed that the odds of having an associated recurrence for samples assigned as high recurrence risk was significantly greater than the low-risk counterparts (OR = 15.45, $p < 0.05$). We believe that this representation of our results will eliminate any confusion in terms of clinicopathological differences observed between high and low risk serum groups.

R4.4. d-MeLB is trained with probes selected from the supervised comparison of meningiomas and non-meningiomas (SI, line 68). p-MeLB model is trained with probes identified through the k-clustering procedure of meningioma samples (SI, line 78). p-MeLB model's predictions on serum samples (high-risk and low-risk groups) picked another 70 DMPs through a supervised analysis (main text, line 403). The feature selection procedure in this work is very complicated, I am especially confused with the last step of using model prediction scores to separate patient groups, then performing supervised analysis to pick out key features. Shouldn't the random forest model be able to pick out key features without any human engineered selection/preprocessing? What's the reason behind each of the feature selection steps?

Response: Thank you for the opportunity to clarify this aspect of our classifier. The implementation of supervised steps to develop our classifier involving liquid biopsy collections across other CNS tumors has been employed in other publications and resulted in accurate prediction models (Sabadot et al. 2021, Herrgott et al., 2022). We implemented these supervised comparisons in order to distill the optimal signature for prediction of

tumor recurrence in both serum and tissue samples. Additionally, the downstream supervised comparison of the high and low risk classes resulting from p-MeLB application was used to further characterize and explore the predicted risk groups (**Supplementary Figure S5g**).

R4.5. (Line 410) Authors then used the k-cluster, recurrence risk groups (generated by p-MeLB) and tissue outcome groups (CR/CNR) to analyze immune composition differences. Given the intercorrelated relations between these groups: p-MeLB is trained on CR/CNR tasks, p-MeLB uses key features identified by k-cluster. I assume these groups to be highly correlated by design. The analysis should be simplified for better clarity.

Response: We agree with the reviewer's point of the inter-relatedness of the analyses. We have simplified the presentation of results, the representation of k-clusters immune landscape was kept in a z-proportion heatmap, as originally submitted (**Supplementary Figure S5a**). K-cluster immune compositions were analyzed to provide more granularity to characterization of the unsupervised groups themselves (**Supplementary Figure S5a**).

Additionally, we removed any attempted collation (stacked barplots) of serum risk (high and low) and tissue outcome group (CR & CNR) immune compositions. Instead, we chose to emphasize the differences in trends observed between systemic response (serum) and the TME in predicted risk groups (**Figure 3b**, **Supplementary Figure 5c**). Briefly, we observed that the immune landscape in these two sources is distinct, suggesting that systemic immune response (serum) is independent of the observed in the TME (tissue). Immune compositions, here, were explored to give us a better understanding of the connection between the systemic and TME immune landscapes present in distinct recurrence risk groups.

R4.6. Overall, the two models proposed in this work (d-MeLB and p-MeLB) serve very different purposes. A lot of the downstream analysis on key DMPs is done by supervised comparison of patient groups (meningioma/non-meningioma, CR/CNR) and k-clusters. I don't see why the two models are necessary for this work, and how (if any) they are connected. Can authors present a clearer motivation for developing these models?

Response: The reviewer is correct about the distinct purposes of d- and p-MeLB. The diagnostic MeLB (d-MeLB) was designed to confirm the presence of meningiomas and to potentially differentiate these tumors from mimicking diseases (comparison between the methylome of meningioma and other CNS conditions and controls); prognostic classifier development aimed to further distinguish meningiomas with distinct progression trends (comparison between the methylome of meningiomas that recurred or not during follow-up).

Considering the reviewer's remarks, we tested whether p-MeLB could also be useful as a diagnostic tool to distinguish across different CNS entities in comparison with d-MeLB. We observed that the predictive power of p-MeLB was substantially lower than that of d-MeLB (LACC: -10.8%, LSE: -35.7%). These results suggest that the specialization of individual models guarantee improved accuracy to classify according to the memberships they were designed to differentiate. Planning for a clinical implementation, the application of these models could involve a two-step process: first, using d-MeLB to classify whether a sample is meningioma, followed by the use of p-MeLB to predict recurrence risk of the samples classified as meningiomas (see **Figure 3d**).

R4.7. Figures in the main text contain a lot of redundant information not discussed, or only briefly touched upon, in the text. Could authors simplify them so each panel contains only the key information?

Response: Thank you for the suggestion. In the revised manuscript we attempted to avoid redundancies and simplify each figure without compromising the main message.

REVIEWER COMMENTS

Reviewer #1 (Remarks to the Author):

I read over the response to reviewers and I'm largely content with the responses given and the efforts to address my previous concerns. I have only some small outstanding queries.

I'm struggling with Figure 1E to know where the 2,069 samples come from or how that tallies with the numbers given in Table 2 which I can't get to add up to 2,069. Additionally the use of the Capper set seems to have been truncated in some way insofar as I think the Capper set has close to 3000 profiles. Were specific profiles selected for these analyses and if so by what rationale and which ones.

I think for the response to R1.10 I may have misunderstood the methodology and thought that features selection was being applied through an iterative process somewhere. I'm still not super clear if that is the case. I didn't receive the google drive link to the code but would have found it useful. I do have a persistent niggle about how easy or difficult this may be to recreate with the methods given. Could this be more transparent?

At the risk of sounding ungenerous I still find the figures quite cramped and dense but I'm willing to put that down to personal preference.

Reviewer #4 (Remarks to the Author):

I appreciate the responses and the additional analyses from the authors, which have improved the paper. I have a few remaining questions.

First, it'd be helpful if clearly label validation data I and validation data II (the new data) in the tables and figures, and then apply the model to both validation datasets wherever possible. For example, is Fig 2b using only validation data I? Why is validation II not used? What data is being used in Fig 2d? Similar questions for other figures and tables.

Second, from the ML methodology side, the authors only compared with simple baselines (dimensional reduction + RF and a linear discriminant model). It'd be interesting to see how models like XGBoost would do here.

Reviewer #5 (Remarks to the Author):

The manuscript: Detection of Diagnostic and Prognostic Methylation-based Signatures in Liquid Biopsy Specimens from Patients with Meningiomas details analysis of Meningiomas and non-Meningiomas through tissue, serum and plasma markers. They demonstrate classification of MNGs using a diagnostic algorithm as well as a prognostic algorithm.

The authors were previously asked to respond to a reviewer's comments regarding use of serum vs plasma.

Summary Comments:

My general response to the draft manuscript is that the presentation is confusing, the analyses are not transparent, and the conclusions are questionable. I strongly recommend clarifying the text, removing extraneous information, and streamlining the entire manuscript. Of critical importance, get to clarity on what is being measured and what biology it explains. Takeaway messaging:

1. There is no mention of ctDNA in this text. Furthermore, there is no comparison to normal samples (neither tissue, plasma, or serum), which would be necessary to conclude they are looking at ctDNA. Therefore, all we can talk about is cfDNA.
2. cfDNA represents a mixture of background contributed by peripheral blood cells and ctDNA, if there is any. I do not believe that the EPIC array is sensitive enough to detect ctDNA in biofluids. If cfDNA has 1/1000 ctDNA molecules, or worse 1/10,000 the average beta value at any location of a sample should not differ significantly from a control on an EPIC array.
3. Nevertheless, these results show differences of -0.29 to 0.20 beta values between MNG and nonMNG samples from serum. To get numbers this high from cfDNA indicates that a mixed population of cells is being measured. The question remains whether the mixture stems from systemic changes in the

peripheral blood, or contamination of the serum with DNA from white blood cells during processing. The prior reviewer is absolutely correct that serum is not used for the purposes of LB due to this issue.

4. The comparison of plasma and serum appears to show a high correlation rho value. That would suggest that contamination of the serum samples is not occurring. However, similar contamination can also happen with plasma, although it is less prevalent. More likely, a mixture of cells detected in the serum is also detected in the plasma at similar proportions. In this case, cfDNA could be picking up systemic immune cell differences.

5. Line 90- The comparison of tissue and serum immune proportions is not significantly correlated. That finding would suggest that one or the other sample type is biased. Why wouldn't these be significantly correlated? The tissue should be the most reliable sample for diagnostics and prognostics as the methylation signals should be 10-1000 times stronger than blood-based signatures. If the tissue samples lack the immune signatures, then the use of these markers indicates that the classifier is not utilizing disease-specific markers.

6. Line 126- for the reasoning stated above, it is counter intuitive that tissue would perform worse in the classifier than plasma.

7. Deconvolution of the data is useful to see the contribution of blood cell types. It does not distinguish between systemic blood cell composition differences and white blood cell contamination. Therefore, there are two options: A) remove all probes that carry the immune cell bias, using a tool like ComBat, and test the classification results on the full datasets as well as each of the four k means clusters. If the LB probes are reliable, only cluster 4 should have trouble with classification because it was enriched for neutrophils. Perhaps cluster 1 will be also worse, but clusters 2 and 3 should be reliable if they are not dependent on immune cell signatures for classification. If all four clusters fail to classify well, then the immune signature could be driving the diagnostics in ways that are unreliable/ not reproducible/ not disease specific. B) Use five or 10 -fold cross validation on the training data to assess whether there are outliers that are influencing the results and repeat cross-validation on the test set to check if it is significantly different.

8. Line 545 states a p-MeLB classifier with an optimal score cutoff based on the smallest OOB error and highest classification accuracy across the training set. Why select the highest classification accuracy? Please report the mean/median classification accuracy with SD and also repeat for the test data.

9. Ultimately the question is whether immune-specific methylation is a marker of MNG disease and is biologically relevant or is a confounding finding.

10. Line 100 - There are no statistics here: "These DMPs were primarily annotated in intergenic regions (overlapping enhancers: 18.4%; non-enhancers: 41.8%), compared to

frequencies observed across non-differential genome wide intergenic CpGs (overlapping enhancers: 15.1%; non-enhancers: 39.5%). This should have p-value or state not significant.

11. There appear to be other problems with the statistics, as in item 8. A thorough review should be applied.

12. Comparison to normal samples should be made. This will define the range of variability for immune cell composition and determine whether these samples can be distinguished from normal plasma or serum.

13. The comparison to other (external) disease types needs to be normalized to ensure fair comparison. I did not find what was done, and which samples were adjusted. Does preprocessing include removal of X and Y chromosomal regions?

14. These statements appear counter-intuitive and could be clarified: Line 98- mean DNA methylation levels in MNG were significantly lower compared to controls and non-MNG samples; Line 227 The prognostic DMPs displayed overall DNA hypermethylation in CR/high-risk samples; Line 250- the aggressive MNGs are hypomethylated across enhancer regions in high-risk samples.

Reviewer #1 (comments to the author):

I read over the response to reviewers and I'm largely content with the responses given and the efforts to address my previous concerns. I have only some small outstanding queries.

R1.1. I'm struggling with Figure 1E to know where the 2,069 samples come from or how that tallies with the numbers given in Table 2 which I can't get to add up to 2,069. Additionally the use of the Capper set seems to have been truncated in some way insofar as I think the Capper set has close to 3000 profiles. Were specific profiles selected for these analyses and if so by what rationale and which ones.

Response: Thank you for the chance to clarify this information. We have made changes to **Figure 1E** to accurately reflect the institutional cohorts included. As the reviewer pointed out, the prior version of the manuscript only used part of Capper's secondary validation cohort (GSE #: 109379). In the revised version, we have included the entire cohort and all meningiomas (MNG) from Capper's detailed cohorts (n=149 MNG; GSE #: 109381). We have also added internally profiled samples, including MNG (n=229; liquid biopsy and tumor tissue) and PitNET tissue (n=10; Mosella et al. Neuro-Oncol, 2021). The new total sample size is now 2,268 with details regarding the GSE identification and cohort information can be found in Table 2 and the provided source data (tab name: Figure 1e).

R1.2. I think for the response to R1.10 I may have misunderstood the methodology and thought that features selection was being applied through an iterative process somewhere. I'm still not super clear if that is the case.

Response: We appreciate the opportunity to elaborate on our methodology. As mentioned in the previous version of the manuscript, our signature set derivation methodology is entirely randomized/independent in each iteration. This means that the machine learning process performs a new, supervised derivation of methylation-based signatures for each of the 1,000 iterations it runs. By doing so, we can identify optimal and distinct meningioma-specific signatures in liquid biopsy serum samples in each iteration. To clarify this point, we have added a detailed explanation of our approach and further attempted to simplify the provided schematic (**Supplemental Figure S3b**).

R1.3. I didn't receive the google drive link to the code but would have found it useful. I do have a persistent niggle about how easy or difficult this may be to recreate with the methods given. Could this be more transparent?

Response: Thank you for your comment. We appreciate your feedback and understand the importance of transparency and reproducibility in our methodology. As part of our blinding process, we have decided not to include the model derivation syntax at this stage of the reviewing process. We are resending the Google link below, which was created by the Nature Communication Editorial team and contains the detailed liquid biopsy cohort *.idats*, generated classifiers and necessary classification syntax we have uploaded. We believe that the details provided in our submission are sufficient for the reviewers to test our random forest models. The syntax will be available upon publication.

(<https://drive.google.com/drive/folders/1cDTJOI4UG2t7HMJHM1btGUiRzEQDdnYm?usp=sharing>)

R1.4. At the risk of sounding ungenerous I still find the figures quite cramped and dense but I'm willing to put that down to personal preference.

Response: Thank you for the comment. We appreciate your flexibility in accepting the current presentation style, even though it may not align with your preferred standards.

Reviewer #4 comments:

I appreciate the responses and the additional analyses from the authors, which have improved the paper. I have a few remaining questions.

R4.1. First, it'd be helpful if clearly label validation data I and validation data II (the new data) in the tables and figures, and then apply the model to both validation datasets wherever possible. For example, is Fig 2b using only validation data I? Why is validation II not used? What data is being used in Fig 2d? Similar questions for other figures and tables.

Response: Thank you for this suggestion. To eliminate any confusion regarding the conducted validations, we have formulated this supplementary table which simplifies the cohorts used for each analysis, for your reference. For instance, we have explicitly stated that all tissue cohorts were used in the Kaplan-Meier survival analysis in **Figure 2d**. We have made the necessary changes by segregating the validation data and including relevant labels in validation **Figures 1f-h and 2b**. Furthermore, we have added annotations throughout the methods to provide information about the cohorts used in each analysis.

	Figure	Analysis	Original Cohort		Validation i Cohort			Validation ii Cohort		
			Serum	Tissue	Serum	Plasma	Tissue	Serum	Plasma	Tissue
d-MeLB	1f	Validation	[x]							
	1g/h	Validation			[x]	[x]		[x]	[x]	
	S3c	Method Contrast	[x]		[x]	[x]		[x]	[x]	
	S4b	Tissue LDA		[x]						[x]
p-MeLB	2b	Validation	[x]	[x]	[x]			[x]		[x]
	2c	Model Contrast		[x]						[x]
	2d	Survival		[x]						[x]

R4.2. Second, from the ML methodology side, the authors only compared with simple baselines (dimensional reduction + RF and a linear discriminant model). It'd be interesting to see how models like XGBoost would do here.

Response: Thank you for the suggestion. We appreciate this opportunity to develop and implement XGBoost-based classification within our cohort. In brief, we developed an XGBoost algorithm which employed identical methodologies for methylation set derivation (using the XGBoost v1.7.4 package), as applied in the d-MeLB model development. We optimized the hyper-parameters of the generated algorithm using fivefold cross-validation resampling techniques (function `xgb.cv`), allowing for experimentation and isolation of optimal parameters for diagnostic purposes. However, the generated classifiers did not surpass the performance of our previous machine learning approaches (random forest), with the observed accuracy being approximately 76% across the total independent cohort vs 81.1% for the d-MeLB classifier (total liquid biopsy validation; n = 122). We have included this information in the appropriate sections and the table below in **Figure S4a**.

a.

Comparisons of diagnostic Meningioma Liquid Biopsy (d-MeLB) performance to other classification approaches.

	Machine Learning				Logistic Regression: Univariate & Multivariate		
	d-MeLB [RF]	DR [RF]	LDA	XGBoost	GW	Supervised	GW & Supervised
Signature Set (n):	25 CpGs	8,869 SMPs	21 CpGs	23 CpGs	~573k CpGs	98 DMPs	~735k CpGs & 98 DMPs
Serum-based Model Selection Set (n=23):							
AUC:	1.00	0.913	0.87	0.9	N/A	N/A	N/A
Total Independent Validation (Serum & Plasma; n=122):							
ACC (%):	81.1	72.1*	63.1*	76.2	57.4*	73.0	74.6
SE (%):	73.8	33.3*	38.1*	66.7	76.2	88.1	90.5
SP (%):	85.0	92.5	76.3*	81.3	47.5*	65.0*	66.3*
CUI (+):	0.532	0.233	0.174	0.434	0.329	0.501	0.529
CUI (-):	0.732	0.671	0.535	0.669	0.376	0.593	0.616
Immune-Based Validation (Serum & Whole Blood; n=59):							
ACC (%):	93.2	100	27.12*	100	88.14	100	100

*Notes: All statistical comparisons are between d-MeLB and the indicated classifier. RF: Random Forest; DR: Dimension Reduction; CUI(+): clinical utility index for positive cases; CUI (-): clinical utility index for negative cases; LDA: linear discriminant analysis; DMPs: differentially methylated probes; *: comparison to d-MeLB performance is statistically significant at p<0.05.*

Reviewer #5 comments:

The authors were previously asked to respond to a reviewer's **comments regarding use of serum vs plasma**.

Response: Thank you for bringing up this common concern that stems from the lack of a formal comparison between both blood elements available in the literature. As previously stated to the mentioned reviewer, to address this concern we conducted a comparative analysis between a paired serum and plasma set by profiling the cfDNA methylomes of these blood elements (n=10), allowing us to compare the observed values. Our findings revealed a positive correlation in various aspects, including DNA methylation-based immune cell estimation, genome-wide (GW) DNA methylation patterns, and concordance of diagnostic and prognostic classifications. Additionally, we observed a differential enrichment or depletion of serum immune cell proportions across k-clusters and recurrence risk groups, as classified through p-MeLB (**Supplementary Figure S6a, Figure 3b**, respectively).

While acknowledging that lysed peripheral blood mononuclear cells (PMBC) DNA might contribute to our immune findings to some extent, we present evidence supporting the hypothesis that alterations in cell proportions observed in serum specimens genuinely reflect an inflammatory or immune response to the presence of meningiomas.

Summary Comments:

My **general response** to the draft manuscript is that the **presentation is confusing**, the **analyses are not transparent**, and the **conclusions are questionable**. I strongly recommend **clarifying the text, removing extraneous information, and streamlining the entire manuscript**. Of critical importance, get to **clarity on what is being measured and what biology** it explains.

Thank you for the comments. In the revised version, we attempted to further improve the readability of the text and the detailing of our analyses by specifically addressing the concerns highlighted by the reviewer.

R5.1a. There is no mention of ctDNA in this text.

Response: Thank you for your comment. We did not directly measure ctDNA, which is why we referred to our liquid biopsy molecular element as cfDNA in concordance with other authors (Lone et al., Mol Cancer 21, 79 (2022) ; Silva, R. et al. Clin Epigenet 13, 168 (2021). Ideally, we would have liked to perform methylation profiling of cfDNA paired with assays which could detect meningioma-specific mutational burden, such as droplet digital PCR or fragmentomics. However, the limited DNA input was sufficient for a singular assay, and some of these methods presented limited sensitivity to detect somatic genomic alterations, as mentioned in a previous study (Mouliere et al. Sci Transl Med 7;10(466):eaat4921(2018).

However, to ensure the meningioma-specificity of our cfDNA-derive signatures, we implemented measures as shown in the updated d-MeLB approach through the selection of CpG probes which were differentially methylated between nontumorous epileptic brain (n=21; GSE111165) and internal MNG (n=30) collections and further filtering these signatures to those which were similarly methylated in matched meningioma serum and tissue. We tested and validated the specificity of these selected probes by overlapping them with an independent meningioma tissue cohort available in Capper et al. Nature. (22;555(7697):469-474(2018) and observing that they clustered with meningiomas (**Figure 1e**).

In order to maintain consistent and accurate nomenclature, we have decided to use the term "meningioma-specific cfDNA" to describe the molecular components identified in our analyses. This terminology aligns with previous authors who have referred to their targeted molecular components in liquid biopsy as "tumor-derived cfDNA" rather than ctDNA (Li et al., Nature Communications, 2022; Chen et al., Onco Targets Ther, 2019; Garcia-Prado et al., Br J Cancer, 2022),

R5.1b. Furthermore, there is no comparison to normal samples (neither tissue, plasma, or serum), which would be necessary to conclude they are looking at ctDNA. Therefore, all we can talk about is cfDNA.

Response: Thank you for the chance to clarify and improve our cohort upon this concern. In the previous version of our work, we did include controls such as nontumor meninges for tissue (**Figure 1d**), and CNS nontumor specimens for serum (epilepsy, abscess, necrosis, infection, and colloid cyst) (**Supplementary File 1**).

To address the concerns raised by the reviewer regarding the lack of comparison to normal samples and potential immune biases, we have made several revisions to our study. Firstly, we expanded the control cohort by profiling the methylome of additional serum samples obtained from patients with non-neoplastic conditions such as epilepsy, histiocytosis, vasculopathy, indiscriminate nontumor specimens, and necrosis. This expansion resulted in a total of 20 samples in the control cohort for more comprehensive comparisons.

We also removed potential outliers from the discovery cohort encompassing nontumor conditions which could potentially trigger local or systemic immune responses, from the training cohort. These excluded samples, including abscesses, necrosis, and sepsis (n=8), were moved to the validation cohort. This adjustment aimed to eliminate biases towards immune markers which could influence classification accuracies, as suggested by the reviewer.

Additionally, we incorporated an additional validation by retrieving publicly available methylome data from whole blood and purified immune cell types to test the specificity of our generated classifier and evaluate potential immune bias (GSE110555) presented in **Supplementary Figure S3c**.

Following the establishment of the new control group consisting of non-neoplastic diseases, we proceeded to refine our d-MeLB model. This involved incorporating two key elements:

- Comparison between the tissue methylome of meningioma (n= 31) and non-neoplastic samples (n=21; Braun et al., 2019; GSE111165) to capture DMP signatures capable of differentiating meningiomas from control samples.
- Comparison of serum methylomes between untreated meningioma (initial and recurrent) and non-meningioma samples (non-neoplastic diseases and CNS tumors, excluding glioblastoma cases as they already exhibit appreciable differential characteristics) (**Supplementary Figure S3b**) to ensure the applicability of the identified signatures to a broader range of liquid biopsy-based samples.

Through these modifications, our newly formulated MeLB model demonstrated accurate classification of whole-blood and purified immune cells as non-meningiomas, as illustrated in **Supplementary Figure S3c**. These results provide strong evidence that we have successfully captured meningioma-specific signatures.

R5.2. cfDNA represents a mixture of background contributed by peripheral blood cells and ctDNA, if there is any. a) I do not believe that the EPIC array is sensitive enough to detect ctDNA in biofluids. If cfDNA has 1/1000 ctDNA molecules, or worse 1/10,000 the average beta value at any location of a sample should not differ significantly from a control on an EPIC array.

Response: Thank you for the opportunity to address this concern. Past and other studies provide evidence that the EPIC array is sensitive and generates reproducible and reliable results in detecting methylation differences across samples that yield low and ultra-low DNA input, such as liquid biopsy samples (including cf/ctDNA, as low as 1ng) (Sabadot et al., Neuro Oncol, 2021, Herrgott et al., Neuro Oncol, 2022; Christiansen et al., Epigenetics, 2022; Whalley et al., Epigenomes, 2021; Li et al., Nature Communications, 2022).

R5.3. Nevertheless, these results show differences of -0.29 to 0.20 beta values between MNG and nonMNG samples from serum. To get numbers this high from cfDNA indicates that a mixed population of cells is being measured. The question remains whether the mixture stems from systemic changes in the peripheral blood, or contamination of the serum with DNA from white blood cells during processing. The prior reviewer is absolutely correct that serum is not used for the purposes of LB due to this issue. The

comparison of plasma and serum appears to show a high correlation rho value. That would suggest that contamination of the serum samples is not occurring. However, similar contamination can also happen with plasma, although it is less prevalent. More likely, a mixture of cells detected in the serum is also detected in the plasma at similar proportions. In this case, cfDNA could be picking up systemic immune cell differences.

Response: Thank you for the comment. We agree that our deconvolution results indicate that we are detecting methylation signatures from a mixed population of cell types, including immune cells, and possibly from other cell types which compose the bulk tumors, and were not accounted for (e.g., endothelial cells). These results are consistent to those reported in bulk tumor tissue deconvolution, which also shows methylation patterns associated with immune and other microenvironment-related cells in other CNS tumors (Sing et al., *Acta Neuropathol Commun*, 2021).

Although it is possible that the methylation signatures identified in the serum samples could be contaminated with white blood cell signatures released during processing, it does not necessarily mean we are not capturing genuine markers that are specific to the presence of meningioma. Corroborating with the authenticity of the serum results include the presence of parallel differences in immune features in plasma samples, which supposedly have no or less contamination with immune cells than serum (as shown in **Supplementary Figure S2a**); a positive correlation between the differentially methylated probes identified in serum and plasma; and the clustering of both blood elements together with meningioma tissue (**Figure 1e**). Additionally, suggesting their biological importance, through methylation-based deconvolution analysis, we found that B-cell depletion, neutrophil enrichment, and increased neutrophils over lymphocytes ratios (NLR) were associated with meningiomas with different growth behaviors (e.g., high vs. low recurrence risk) (**Figure 3, Supplementary File 1**), which is consistent with the reported prognostic significance of inflammatory markers assessed by other methodologies (e.g., neutrophilia, increased NLR) as reported by other authors (Karimi et al., *Neuro-Oncology*, 2017; Kranrmi et al., *Front Oncol*, 2020).

Another strategy we took to test the specificity of the d-MeLB model was to retrieve genome-wide data from whole-blood and purified immune cell types (obtained through fluorescence-activated cell sorting; FACS, GSE110555). Our rationale was that if the signatures used in the reformulated d-MeLB model were biased towards spurious immune contamination, it would classify them as meningiomas. However, the model correctly classified a majority of these samples as non-meningiomas, corroborating that the immune signatures detected in serum were a response to the presence of meningiomas (**Supplementary Figure S3c**).

R5.4. Line 90- The comparison of tissue and serum immune proportions is not significantly correlated. That finding would suggest that one or the other sample type is biased. Why wouldn't these be significantly correlated? The tissue should be the most reliable sample for diagnostics and prognostics as the methylation signals should be 10-1000 times stronger than blood-based signatures. If the tissue samples lack the immune signatures, then the use of these markers indicates that the classifier is not utilizing disease-specific markers. Line 126- for the reasoning stated above, it is counter intuitive that tissue would perform worse in the classifier than plasma.

Response: Thank you for your comment. Our results suggest that the lack of a significant correlation between the immune composition of serum and tumor tissue could be attributed to the possibility that local immune response in the tumor microenvironment is distinct from the

systemic immune or inflammatory response to the presence of meningioma, as reported by other studies (Xu et al, J Hematol Oncol, 2022; Suh et al. Plos One, 2021; Rumba et al. Acta Med Litu, 2018; Haslund-Vinding et al., Neurosurgery Rev, 2022). Based on this observed lack of significant correlation between tissue and serum immune landscape, we propose that these markers (enrichment with neutrophil signatures and high neutrophil-to-lymphocyte ratios) are indicative of the systemic inflammatory response triggered by the presence of meningiomas, rather than solely or not at all as reflection of the local immune response. However, this hypothesis needs to be confirmed through the comparison between local and systemic immune composition based on other methods such as flow cytometry.

Considering the reviewer's concern about the limited, although expected, performance of the d-MeLB to classify tissue methylome classification for meningiomas using random forest algorithm, we assessed the meningioma-specificity of the d-MeLB signatures as input to a simplified linear discriminant approach to classify the tissue methylome.

To train and validate the linear-based classifier, we utilized a combination of internally and externally profiled tissue methylome samples, including meningiomas (n=138), intracranial mesenchymal tumors (n=20; GSE164994), and nontumor collections (n=21; GSE111165). Additionally, for further validation, we included internally-derived pituitary tumor samples (n=14) and retrieved methylome data from publicly available datasets of meningiomas (n=110; GSE189521) and gliomas (n=16; GSE147391).

Using this algorithm tailored for tissue classification and incorporating the newly derived d-MeLB signatures, we were able to accurately classify meningioma and non-meningioma tissue samples in an independent cohort consisting of a total of 176 samples, achieving an accuracy of 94.3% (**Supplementary Figure S4b**).

This approach demonstrated the efficacy of the d-MeLB signatures in accurately distinguishing meningioma from non-meningioma using tissue samples.

R5.5. Deconvolution of the data is useful to see the contribution of blood cell types. It does not distinguish between systemic blood cell composition differences and white blood cell contamination. Therefore, there are two options: A) remove all probes that carry the immune cell bias, using a tool like ComBat, and test the classification results on the full datasets as well as each of the four k-means clusters. If the LB probes are reliable, only cluster 4 should have trouble with classification because it was enriched for neutrophils. Perhaps cluster 1 will be also worse, but clusters 2 and 3 should be reliable if they are not dependent on immune cell signatures for classification. If all four clusters fail to classify well, then the immune signature could be driving the diagnostics in ways that are unreliable/ not reproducible/ not disease specific. B) Use five or 10 -fold cross validation on the training data to assess whether there are outliers that are influencing the results and repeat cross-validation on the test set to check if it is significantly different.

Response: Thank you for your feedback and suggestions. Before any downstream analyses (e.g. deconvolution, model development, etc.), we tested all our data (liquid biopsy serum and tumor tissue) for batch effects by plate number (serum), date of extraction (serum) or institution (tumor tissue) using the ComBat R package. The results of our analysis, as shown in **Supplementary Figures S1f-i**, revealed no evidence of clustering by batch, confirming the reliability of our data. Addressing the reviewer's concern, as mentioned, we applied our original d-MeLB model to classify whole blood and FACS-purified immune cell types. We observed that 48% of samples were misclassified as meningiomas, corroborating with the reviewer's suggestion that our original signatures were potentially biased by immune infiltrates. Possibly, this result was affected because a subset of our control group comprising abscess, necrosis and sepsis specimens separated from the other controls (epilepsy, colloid cyst, indeterminate

nontumor, histiocytosis) were biasing our signatures toward immune cells as they are known to elicit systemic immune response. After exclusion of these samples from the discovery cohort and the expansion of our non-neoplastic control group as detailed in **R5.1b** the reformulated d-MeLB was able to classify whole blood, purified immune cells and the excluded control samples as non-meningiomas.

R5.6. Line 545 states a p-MeLB classifier with an optimal score cutoff based on the smallest OOB error and highest classification accuracy across the training set. Why select the highest classification accuracy? Please report the mean/median classification accuracy with SD and also repeat for the test data.

Response: Thank you for the opportunity to clarify. We utilized OOB error as our guiding measure to isolate high-performing classifiers and applied them to our independent validation datasets. In our previous approach, we did not have a testing/model selection data set due to the limited sample size at the time of classifier derivation (prior to our current cohort expansion). Classification accuracy across the training set was used as a secondary assurance, as we are aware of the inherent bias associated with the calculated classifier performance. We have opted to remove any mention of the training set classification accuracy from the text as we do not find it is crucial to our model selection and validation process.

R5.7. Ultimately the question is whether immune-specific methylation is a marker of MNG disease and is biologically relevant or is a confounding finding.

Response: Thank you for your comment. Based on the pieces of evidence outlined in **R5.3**, it is reasonable to conclude that the immune signatures are bonafide biological markers linked to the presence of meningiomas.

R5.8. Line 100 - There are no statistics here: "These DMPs were primarily annotated in intergenic regions (overlapping enhancers: 18.4%; non-enhancers: 41.8%), compared to frequencies observed across non-differential genome wide intergenic CpGs (overlapping enhancers: 15.1%; non-enhancers: 39.5%). This should have p-value or state not significant. There appear to be other problems with the statistics, as in item 8. A thorough review should be applied.

Response: Thank you for pointing this out. We removed the aforementioned section completely, in an attempt to further streamline the text and reduce the number of irrelevant conclusions, and attempted to clean up any unclear statistical descriptions/depictions throughout the text and figures.

R5.9. Comparison to normal samples should be made. This will define the range of variability for immune cell composition and determine whether these samples can be distinguished from normal plasma or serum.

Response: Thank you for the suggestion. To address this concern, please see details outlined in **R51b**.

R5.10. The comparison to other (external) disease types needs to be normalized to ensure fair comparison. I did not find what was done, and which samples were adjusted. Does preprocessing include removal of X and Y chromosomal regions?

Response: Thank you for the comment. As stated in Lines 478-479 under the preprocessing subsection, we preprocessed all samples (tissue and liquid biopsy) following the guidelines reported by Zhou et al. (**Lines 55-58, Supplementary Method file**). This included the removal of probes located within X and Y chromosomal regions, probes designed for sequences with known polymorphisms (SNP), and those with poor mapping quality. A complete list of masked probes is provided within Zhou et al. We also tested all liquid biopsy samples (internal and external) for batch effects.

We acknowledge the concern raised regarding the lack of adjustment for batch effects in the comparisons made to external disease types, such as TCGA glioma, publicly available PitNET, and meninge samples. In this case, the batch effect is confounded by tumor type, as tumor types are completely nested within individual batches. Consequently, traditional methods for batch adjustment are not applicable in this scenario without losing inherent biological differences. For instance, in our PCA series comparison of tissue-based samples (**Figure 1c**), each institution included only a singular sample type, resulting in the absence of intra-institutional tumor type variation.

However, we did validate the biological relevance of the derived signatures across independent cohorts, as demonstrated in **Figure 1d**. The separation of sample types in the independent liquid biopsy cohort suggests that these CpGs, while could be influenced by batch effects to some extent, primarily exhibit inherent biological differences

The only additional comparison to external disease types made is incorporated within the dimension reduction (**Supplementary Figure S3b**, step #1) of our newly formulated d-MeLB (e.g., MNG vs nontumor tissue-based comparisons), which given performance across both internal and external validations, we do not believe this necessitates batch correction in any form.

R5.11. These statements appear counter-intuitive and could be clarified: Line 98- mean DNA methylation levels in MNG were significantly lower compared to controls and non-MNG samples; Line 227 The prognostic DMPs displayed overall DNA hypermethylation in CR/high-risk samples; Line 250- the aggressive MNGs are hypomethylated across enhancer regions in high-risk samples.

Response: Thank you for your feedback. The mean methylation levels of the probes mentioned by the reviewer refer to separate and distinct analyses. In line 98 (now lines 99-101), the methylation levels of serum DMPs was obtained from the comparison between meningioma and non-meningiomas (**Supplementary Figure S3a**). Similarly, line 227 (removed) refers to the methylation levels of risk-specific DMPs obtained from the comparison between confirmed recurrence and non-recurrence in tissue (**Figure 4a**), whereas line 250 (now lines 280-282) refers to the methylation levels of risk-specific DMPs from the comparison of high and low risk samples in serum (**Supplementary Figure S6g**). We revised these statements and kept only the most relevant to the discussion for clarity in the manuscript.

REVIEWERS' COMMENTS

Reviewer #1 (Remarks to the Author):

I'm afraid the authors haven't entirely dealt with my small outstanding queries. With reference to my first question about the numbers used in Figure 1E this is a straightforward query. I note the new total sample size of 2268 but I can't yet tally this with either the numbers given in the reviewers response or those that are given in the table 2. If I look at table 2 using the relevant figures column as my guide for Figure 1E $19 + 149 + 57 + 1230 + 10 + 34 + 23 + 516 = 2038$ not 2268. Maybe I'm misinterpreting something but I don't see that this is clear. The point is we just need a straightforward description of which samples were used from which dataset so that someone else could follow and recreate the analysis in the future. If the whole dataset was used fine if some datapoints were excluded detail that as well.

For my third question about the google drive link. I think what has been provided is useful up to a point but if I'm reading it correctly this is simply the basic code to apply the final random forest model to the datasets. My query really was about all of the methodology that led up to creating the models in the first place all the stuff that is given in the schematic and I'm not sure if this really helps to elucidate this. It seems as if the authors are saying they have decided not to provide the code for how they derived the model? If that is the case I don't really see how this is justifiable. I don't know how this coheres with journal policy but nevertheless providing some code examples is extremely helpful for transparency and reproducibility especially for a relatively complicated procedure such as the one described in the schematics.

Reviewer #4 (Remarks to the Author):

I'm mostly satisfied with the authors' response. One clarification question is in the new Fig. S4a, why is the signature set for XGBoost ($n = 23$) different from the signature set for d-MeLB ($n = 25$)? What happens if you use the name signature set for XGBoost?

Reviewer #5 (Remarks to the Author):

The revised version of Detection of Diagnostic and Prognostic Methylation-based Signatures in Liquid Biopsy Specimens from Patients with Meningiomas is much improved and has gained clarity.

There remain a few items for response.

o Several places in the text the methylation sets are indicated as being able to differentiate risk groups- a PCA plot is shown to illustrate this point. However, the PCA plots do not show convincing separation. Maybe I'm missing the point, but I see intermixed samples on most of the PCAs. For example, Fig 3d, 4b, 4c, 4e. If there is a better explanation for the PCA illustration, please provide it to the readers.

o Figure 1f has "other" CNS disease plotted that has high methylation values in the MeLB Score, what is this sample?

o I could not find a GSEA entry for PRC, did I miss it? It looks as though many gene body and open sea locations are differentially methylated compared to promoter probe positions.

o I could not find Methods for the gene expression analysis.

We appreciate the Reviewers' comments and suggestions.

Reviewer #1

I'm afraid the authors **haven't entirely dealt with my small outstanding queries.**

Response: We appreciate the opportunity to readdress the reviewer's queries.

R1.1 With reference to my first question about the numbers used in Figure 1E this is a straightforward query. I note **the new total sample size of 2268 but I can't yet tally** this with either the numbers given in the reviewer's response or those that are given in table 2. If I look at table 2 using the relevant figures column as my guide for Figure 1E $19 + 149 + 57 + 1230 + 10 + 34 + 23 + 516 = 2038$ not 2268. Maybe I'm misinterpreting something but I don't see that this is clear. The point is we just need a straightforward description of which samples were used from which dataset so that someone else could follow and recreate the analysis in the future. If the whole dataset was used fine if some data points were excluded detail that as well.

Response: We appreciate the reviewer's thorough observation regarding the samples included in this analysis and for the opportunity to correct the information - the external samples displayed in the t-SNE constitute 2,038 out of the total of 2,267 (not 2,268 as mentioned above). The remainder 229 samples are internally collected: MNG serum (n=81), plasma (n=10) and tissue (n=138) from Institutions #1 & #2.

We have adjusted the title of the figure to include the accurate amount of samples (n=2,267). Additionally, we have attempted to provide extra clarification within the methodology, to more specifically elucidate on the sample pools being utilized, paving the way for better reproducibility (lines 580-582).

R1.2 For my third question about the **google drive link**. I think what has been provided is useful up to a point but if I'm reading it correctly this is simply the basic code to apply the final random forest model to the datasets. **My query really was about all of the methodology that led up to creating the models** in the first place, all the stuff that is given in the schematic and I'm not sure if this really helps to elucidate this. **It seems as if the authors are saying they have decided not to provide the code for how they derived the model?** If that is the case I don't really see how this is justifiable. I don't know how this coheres with journal policy but nevertheless providing some code examples is extremely **helpful for transparency and reproducibility** especially for a relatively complicated procedure such as the one described in the schematics.

Response: We appreciate the opportunity to clarify the reviewer's concerns. Our initial intent was to provide enough information without jeopardizing the novelty of our approach; however, given the favorable decision of the journal for publication, we made the requested detailed syntax, and all files required for reproduction of our diagnostic classifier (d-MeLB), as an example of our methodology, available under the Mendeley Data Accession DOI: 10.17632/zrc982rvjm.1, as well as within the supplementary data (Supplementary Data S1 - Clinical Information). Within our DOI, the reviewer and the public will find the IDAT files from tumor tissue and liquid biopsy specimens, source code to construct our random forest classifier, as well as the generated classifier to ensure reproducibility of our methodologies/results.

Reviewer #4 (Remarks to the Author):

I'm **mostly satisfied** with the authors' response.

Response: We appreciate the comment.

R4.1 One clarification question is **in the new Fig. S4a, why is the signature set for XGBoost (n = 23) different from the signature set for d-MeLB (n = 25)?** What happens if you use the name signature set for XGBoost?

Response: We appreciate the question and suggestion from the reviewer. To provide further clarification, we would like to highlight that while XGBoost replicated the methodology used in our d-MeLB model development, the parameters were re-randomized. This resulted in a final signature set size of 23 CpGs for XGBoost, whereas for d-MeLB it was 25 CpGs, as pointed out by the reviewer.

However, as an attempt to further satisfy the curiosity of the reviewer, we constructed an XGBoost using identical signatures to that of the RF, and have recorded the performance below. Overall, the XGBoost trained with 25 d-MeLB signatures performed similarly to the random forest, with minor improvements in overall accuracy and specificity in the total independent validation (serum & plasma); however, it showed a slight decrease in performance across the immune-based validation cohort.

Comparisons of diagnostic Meningioma Liquid Biopsy (d-MeLB) performance to other classification approaches.

	Machine Learning				Logistic Regression: Univariate & Multivariate		
	d-MeLB [RF]	DR [RF]	LDA	XGBoost	GW	Supervised	GW & Supervised
Signature Set (n):	25 CpGs	8,869 SMPs	21 CpGs	25 CpGs	~573k CpGs	98 DMPs	~735k CpGs & 98 DMPs
Serum-based Model Selection Set (n=23):							
AUC:	1.00	0.913	0.87	0.99	N/A	N/A	N/A
Total Independent Validation (Serum & Plasma; n=122):							
ACC (%):	81.1	72.1*	63.1*	81.9	57.4*	73.0	74.6
SE (%):	73.8	33.3*	38.1*	73.8	76.2	88.1	90.5
SP (%):	85.0	92.5	76.3*	86.3	47.5*	65.0*	66.3*
CUI (+):	0.532	0.233	0.174	0.545	0.329	0.501	0.529
CUI (-):	0.732	0.671	0.535	0.744	0.376	0.593	0.616
Immune-Based Validation (Serum & Whole Blood; n=59):							
ACC (%):	93.2	100	27.12*	89.8	88.14	100	100

*Notes: All statistical comparisons are between d-MeLB and the indicated classifier. RF: Random Forest; DR: Dimension Reduction; CUI(+): clinical utility index for positive cases; CUI (-): clinical utility index for negative cases; LDA: linear discriminant analysis; DMPs: differentially methylated probes; *: comparison to d-MeLB performance is statistically significant at $p \leq 0.05$.*

Reviewer #5 (Remarks to the Author):

The revised version of Detection of Diagnostic and Prognostic Methylation-based Signatures in Liquid Biopsy Specimens from Patients with Meningiomas **is much improved and has gained clarity.**

Response: We appreciate the encouraging remarks

There remain a few items for response.

R5.1 Several places in the text the methylation sets are indicated as being able to differentiate risk groups- a PCA plot is shown to illustrate this point. However, the **PCA plots do not show convincing separation.** Maybe I'm missing the point, but I see intermixed samples on most of the PCAs. For example, Fig 3d, 4b, 4c, 4e. **If there is a better explanation for the PCA illustration, please provide it to the readers.**

Response: We thank the reviewer for the opportunity to clarify this concern. We agree that the PCA shows moderate separation of groups within a two-dimensional space (e.g., high and low risk show a general perception of clustering, with minor intermingling) and may not be the most convincing argument to display. Thus, we provided new three-dimensional visualizations in Figures 4a-c (now Figures 4a-b). However, due to spacing constraints, we were unable to re-orient Figure 4e in the same manner. We have also adjusted the language throughout the result description of the figures to more gently describe the apparent trend of separation, addressing this particular discrepancy appropriately (lines 264-271).

R5.2 Figure 1f has "other" CNS disease plotted that has high methylation values in the MeLB Score, **what is this sample?**

Response: Thank you for your inquiry. All 'other' CNS diseases included within the paper are CNS lymphomas (primary: n=3; and secondary: n=1) and were detailed in **Supplementary Data S1**.

R5.3 I could not find a **GSEA entry for PRC**, did I miss it? It looks as though many gene body and open sea locations are differentially methylated compared to promoter probe positions.

Response: We appreciate the opportunity to clarify this important point raised by the reviewer. The CpG sites related to the PRC (Polycomb Repressive Complex)-related genes were not obtained from the analysis of our identified DMPs, nor through Gene Set Enrichment Analysis (GSEA). Instead, the selection was based on the survey of a list of 26 PRC-related genes associated with hypermethylated CpG islands and malignant transformation of meningiomas (Gao et al., 2013), across the methylome of our samples.

To elaborate, we mapped the PRC genes to their respective CpG islands using our previously described methods (lines 531-533). Then, we calculated the average methylation level across these CpG sites in both our high-risk and low-risk samples from all three sample sources (serum, plasma, and tissue). This analysis allowed us to demonstrate that the methylation patterns of these CpG sites in both serum and tissue across our risk prediction subgroups (with hypermethylation in high risk) were consistent with those reported in the malignant groups as described by Gao et al.

We have revised the text to provide a clearer explanation of this analysis, which can be found in lines 298-302.

R5.4 I could not find **Methods for the gene expression analysis**.

Response: Thank you for the opportunity to clarify. The gene expression analysis is described in the **Supplementary Text** under the subtitle: *Identification of biologically relevant and equivalent DNA methylation markers between tissue and serum* (lines 147-168). Our gene expression analysis was conducted using publicly available transcriptome data extracted from the Choudhury et al. publication. Briefly, we applied the p-MeLB model to classify the Choudhury meningioma tissue cohort (n=185) into high and low recurrence risk groups and intersected these results with equivalent prognostic groups reported in their publication (i.e. hypermitotic and merlin-intact groups, respectively), namely the high risk/hypermitotic and low risk/merlin subgroups. We analyzed the matching tissue transcriptome of these samples (available in the Choudhury cohort) and identified differentially expressed genes (DEG) using the DESeq2 (v1.36.0) package. We mapped the resulting DEGs to CpGs located in their regulatory elements (promoters and enhancers) and manually selected those CpGs which simultaneously presented differential DNA methylation levels across the subgroups as well as negative correlation with the expression of their putative target genes, to infer a potential epigenetic regulation. In a complementary manner, DMPs derived from comparisons of high and low risk groups in serum samples were mapped to tissue-derived DEGs (n=32 PGPs). These DMPs were analyzed for concordant DNA methylation across tissue groups (e.g. serum groups: hypermethylation in high risk vs low risk; tissue groups: hypermethylation in high-risk/hypermitotic vs low-risk/merlin-intact), and inverse correlation with the expression levels of their putative target genes (e.g. hypermethylation and downregulation of gene). Then, we conducted gene set enrichment analyses (GSEA) through the Ingenuity Pathway Analysis (IPA) platform, identifying relevant diseases and biological processes related to these concordantly methylated recurrence risk-related DMP-genes.